# Genome surveillance by HUSH-mediated silencing of intronless mobile elements

Marta Seczynska[1], Stuart Bloor[1], Sergio Martinez Cuesta[2] & Paul J. Lehner[1✉]

All life forms defend their genome against DNA invasion. Eukaryotic cells recognize incoming DNA and limit its transcription through repressive chromatin modifications. The human silencing hub (HUSH) complex transcriptionally represses long interspersed element-1 retrotransposons (L1s) and retroviruses through histone H3 lysine 9 trimethylation (H3K9me3)[1–3]. How HUSH recognizes and initiates silencing of these invading genetic elements is unknown. Here we show that HUSH is able to recognize and transcriptionally repress a broad range of long, intronless transgenes. Intron insertion into HUSH-repressed transgenes counteracts repression, even in the absence of intron splicing. HUSH binds transcripts from the target locus, prior to and independent of H3K9me3 deposition, and target transcription is essential for both initiation and propagation of HUSH-mediated H3K9me3. Genomic data reveal how HUSH binds and represses a subset of endogenous intronless genes generated through retrotransposition of cellular mRNAs. Thus intronless cDNA—the hallmark of reverse transcription—provides a versatile way to distinguish invading retroelements from host genes and enables HUSH to protect the genome from 'non-self' DNA, despite there being no previous exposure to the invading element. Our findings reveal the existence of a transcription-dependent genome-surveillance system and explain how it provides immediate protection against newly acquired elements while avoiding inappropriate repression of host genes.

The mammalian genome is under constant threat from invasion by mobile genetic elements including transposons and viruses. Controlling this activity is fundamental to genome integrity. These defence strategies often use repressive chromatin to silence target gene expression and major chromatin-silencing factors in mammalian cells include: (1) small RNA guides complementary to nascent transcripts and (2) sequence-specific DNA-binding proteins[4]. PIWI-interacting RNAs (piRNAs) guide PIWI proteins to transposon transcripts and promote repressive chromatin at germline transposon loci[5]. piRNAs are derived from piRNA clusters, genomic loci enriched in transposon-derived sequences[6,7]. The piRNA pathway therefore relies on the memory of transposon invasions to provide adaptive, sequence-based immunity. The large KRAB-containing zinc-finger protein (KRAB-ZFP) family of sequence-specific DNA-binding proteins recruit TRIM28 and the SETDB1 methyltransferase to deposit H3K9me3 heterochromatin at target loci[7,8]. piRNA and KRAB-ZFP pathways are mostly active in the germ line and pluripotent stem cells, whereas the HUSH complex silences mobile elements in pluripotent stem cells and differentiated cells. HUSH represses evolutionary young L1 retrotransposons[2,3], the only active autonomous mobile transposons in humans, as well as integrated lentiviruses[1] and unintegrated murine retroviral DNA via NP220[9]. The importance of HUSH in controlling lentiviral infection is emphasised by the finding that complex primate lentiviruses encode accessory proteins (Vpr and Vpx) that degrade HUSH[10–12].

To silence mobile elements, the HUSH complex of TASOR, MPP8 and periphilin, recruits two effectors: MORC2—an ATP-dependent chromatin remodeller—enables chromatin compaction[13,14], and SETDB1 deposits H3K9me3[1]. The chromodomain of MPP8 binds to H3K9me3-modified chromatin anchoring HUSH at the target locus. However, how HUSH recognizes its targets to initiate H3K9me3 deposition is unknown.

## Intronless transgenes are HUSH-repressed

Since HUSH-repressed L1s are found in diverse genomic integration sites[2,3,15], the signal for HUSH recognition must be intrinsic to the L1. To confirm that the L1 sequence confers HUSH repression independent of its integration site, we expressed a lentiviral fluorescent reporter encoding the L1 open reading frame (ORF) and a P2A-iRFP cassette. L1 expression was monitored by flow cytometry with iRFP fluorescence reflecting L1 mRNA abundance (Extended Data Fig. 1a). Inactivation of the ORF2 endonuclease[16] (D205A mutation) reduces retrotransposition; the reporter thus monitors expression from initial L1 integrations (Extended Data Fig. 1c, d). Lentiviral L1 reporter (L1$_{lenti}$) expression is repressed within the entire wild-type population (Fig. 1a), and disrupting HUSH by knockout of HUSH subunits or by TASOR degradation by lentiviral Vpx[10–12] restores L1$_{lenti}$ expression, whether the reporter is integrated before or after HUSH disruption (Fig. 1a, Extended Data Fig. 1b, e–g). As L1$_{lenti}$ is expressed from most integration sites following

[1]Cambridge Institute for Therapeutic Immunology and Infectious Disease, Jeffrey Cheah Biomedical Centre, Cambridge Biomedical Campus, University of Cambridge, Cambridge, UK. [2]Data Sciences and Quantitative Biology, Discovery Sciences, AstraZeneca, Cambridge Biomedical Campus, Cambridge, UK. ✉e-mail: pjl30@cam.ac.uk

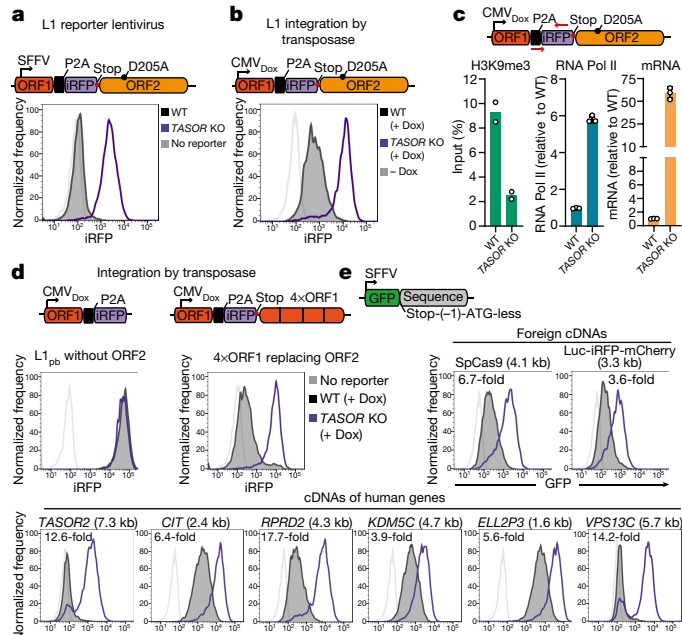

**Fig. 1 | Diverse intronless transgenes are HUSH-repressed. a**, Repression of L1 reporter lentivirus in wild-type (WT) (black) or *TASOR*-knockout (KO) (purple) HeLa cells, measured by flow cytometry. **b, c**, L1 reporter integrated by piggyBac transposase. **b**, Doxycycline (Dox)-induced expression in wild-type and *TASOR* KO HeLa cells measured by flow cytometry. CMV, cytomegalovirus promoter. **c**, Chromatin immunoprecipitation with quantitative PCR (ChIP–qPCR) assays of H3K9me3 (left; mean of *n* = 2 biological replicates ± s.d.) and RNA polymerase II (Pol II) (middle; mean of *n* = 3 biological replicates ± s.d.) in wild-type and *TASOR* KO HeLa cells at the reporter. L1 transcript levels assayed by quantitative PCR with reverse transcription (RT–qPCR) (right; mean of *n* = 3 technical replicates ± s.d.). **d**, Doxycycline-induced expression of piggyBac reporter without ORF2 sequence (left) and with ORF2 sequence (4 kb) replaced by 4×ORF1 (4×1 kb in size) (right) integrated into wild-type or *TASOR* KO HeLa cells. **e**, HUSH-mediated repression of GFP lentiviral reporters bearing different untranslated cDNA sequences measured by flow cytometry 72 h after transduction. Length of the cDNA sequence is indicated in brackets and fold change of reporter expression in *TASOR* KD and wild-type cells measured by geometric mean fluorescence is indicated on the graph. Frequency is normalized to mode (**a, b, d, e**).

HUSH depletion (Fig. 1a, Extended Data Fig. 1e, g), HUSH-mediated L1 silencing is independent of integration site. Lentiviruses predominantly integrate in transcribed gene bodies[17], whereas the piggyBac transposase directly integrates at randomly distributed TTAA sites[18]. L1 reporter expression from an inducible, piggyBac transposon vector (L1$_{pb}$) confirmed HUSH-dependent repression from most integration sites (Fig. 1b, Extended Data Fig. 1h) and HUSH-mediated H3K9me3 deposition that led to decreased RNA Pol II occupancy and reporter mRNA levels (Fig. 1c). The signal for HUSH repression is therefore intrinsic to L1 and independent of the mechanism and site of genome integration.

HUSH restriction of L1 retrotransposition depends on the native nucleotide sequence of the L1 ORF[2]. By testing the HUSH sensitivity of reporters bearing single L1 ORFs[19] (ORF1 or ORF2), we found that the ORF2 sequence alone is responsible for HUSH-mediated repression of L1 (Fig. 1d, left, Extended Data Fig. 1i). However, replacing the 4-kb ORF2 with 4 tandem repeats of the 1-kb ORF1 also caused HUSH repression (Fig. 1d, right, Extended Data Fig. 1j), suggesting that HUSH repression is not unique to ORF2.

We therefore tested the HUSH sensitivity of lentiviral transgenes with different DNA sequences (Fig. 1e, Extended Data Fig. 2a–c). To exclude effects on mRNA translation, we inserted DNA sequences lacking an ATG start codon, with a single-nucleotide frameshift at the

3′-untranslated region (3′ UTR) of the GFP reporter (Fig. 1e). Diverse, integrated transgenes containing cDNA sequences from a wide range of human genes were all HUSH-repressed (Fig. 1e), as were transgenes entirely 'foreign' to the human genome, for example, the bacterial Cas9 nuclease (Fig. 1e, Extended Data Fig. 2d). HUSH therefore silences sequence-diverse self and foreign mobile genetic elements, the latter being important as it excludes the possibility of 'genetic memory'. HUSH-mediated transgene repression was maintained over multiple cell divisions, was independent of the number of transgene integrations and showed a significant correlation with the length of inserted DNA (Extended Data Fig. 2e–g). While the L1 ORF1 (1kb) reporter is HUSH-insensitive, tandem repeats of ORF1 gradually acquire HUSH repression as their size increases (Fig. 1d, Extended Data Fig. 3h). Transgene length therefore contributes to HUSH susceptibility, with short (up to 1 kb) transgenes most likely to escape HUSH-mediated repression (for example, L1 ORF1, iRFP or a fragment of Xist long non-coding RNA (lncRNA)) (Fig. 1d, Extended Data Fig. 2h). However, lentiviral reporters encoding short 1-kb fragments of ORF2 (or 3-kb ORF2 deletion mutants) remained HUSH-repressed (Extended Data Fig. 2i–k), indicating a role for nucleotide composition in HUSH targeting.

We found no correlation between HUSH-mediated repression and adenine and thymine (AT) sequence content (Extended Data Fig. 3a), and decreasing the overall AT content of ORF2 did not alleviate HUSH-dependent silencing (Extended Data Fig. 3b, e). However, HUSH-mediated repression strongly correlates with the A nucleotide content of the sense strand (Extended Data Fig. 3c), with ORF2 showing a strong A (41%) versus T (20%) bias in the sense strand[20]. Indeed, a reverse-complement ORF2 reporter is completely HUSH-resistant, despite expressing a full-length transcript (Extended Data Fig. 3d–g). The HUSH complex therefore represses a broad range of invading DNAs, with transgene length and high A content in the sense strand acting as key determinants of HUSH targeting.

## Transcription is required for repression

Chromatinization of invading DNA precedes genome integration[21]. We therefore investigated whether HUSH initiates repression prior to transgene integration. Lentiviral ORF2 transgenes were HUSH-repressed in both the absence and presence of raltegravir, an inhibitor of viral integration (Fig. 2a, left, Extended Data Fig. 4a, b). Furthermore, transfected lentiviral plasmids encoding (1) ORF2, (2) synthetic ORF2 or (3) Cas9 cDNA sequences were HUSH-repressed (Fig. 2a, right, Extended Data Fig. 4c), as were non-viral plasmids (Extended Data Fig. 4d), indicating that HUSH can initiate silencing prior to transgene integration.

HUSH targets endogenous, full-length, young L1s that are often enriched within transcriptionally permissive euchromatin, suggesting a role for transcription in HUSH targeting[2,3,15]. To directly test whether transcription is required to initiate HUSH-mediated silencing, we transduced HeLa cells with either the standard, spleen focus forming virus (SFFV) promoter-driven L1$_{lenti}$ reporter or an otherwise identical promoterless reporter. HUSH-dependent H3K9me3 accumulated over the transcriptionally active L1 reporter, but was significantly reduced in the absence of a promoter (Fig. 2b, Extended Data Fig. 4e–h). Deletion of the promoter region from *TAF7*, an endogenous HUSH target gene also reduced transcription (Extended Data Fig. 4i, right) and locus-specific H3K9me3 deposition (Extended Data Fig. 4i, left, j) confirming that transcription is required to both initiate and maintain H3K9me3 over HUSH-sensitive loci. Furthermore, silencing cannot be conferred solely by the DNA sequence, as the sequences of HUSH-sensitive and HUSH-insensitive transgenes are identical.

A transcriptional requirement in HUSH-mediated silencing suggests that HUSH binds reporter RNA. Native RNA immunoprecipitation (RIP) showed that periphilin specifically binds RNA from a HUSH-sensitive reporter but not from a HUSH-resistant reporter (Fig. 2c, Extended Data

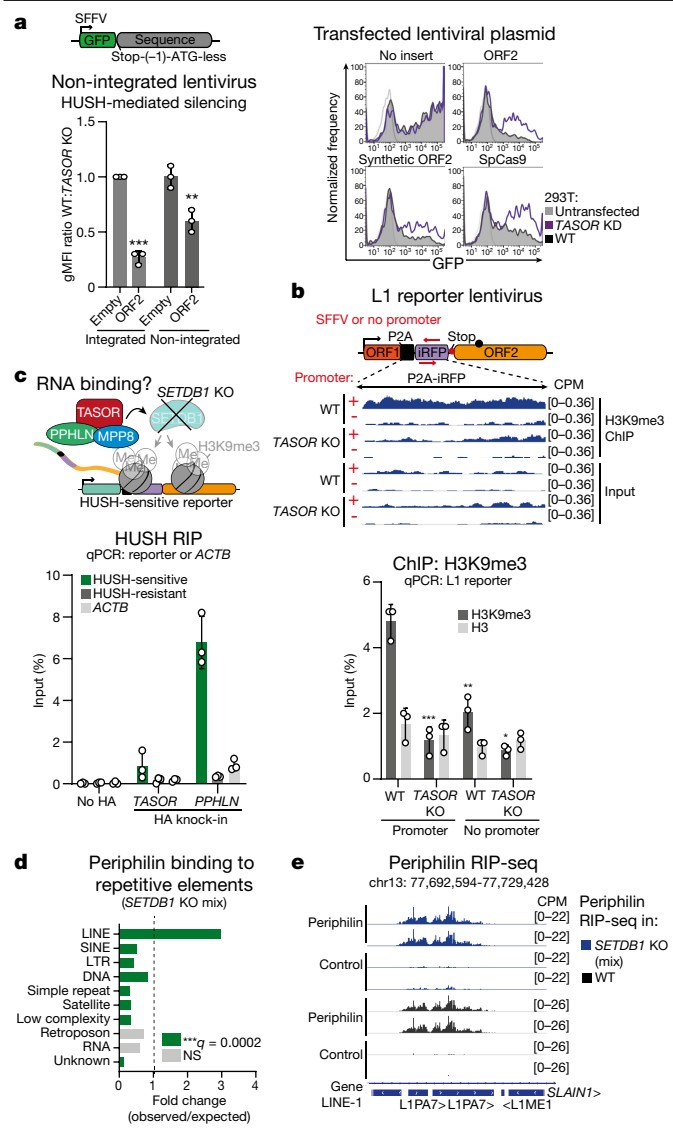

**Fig. 2 | HUSH binds target RNA and initiates silencing before DNA integration.**
**a**, HUSH-mediated repression of non-integrated reporters. Left, HUSH-mediated repression of integrated and non-integrated GFP reporter lentiviruses with no insert (empty) or with synthetic ORF2 measured by flow cytometry 24 h after transduction and calculated as the ratio of reporter expression in wild-type and *TASOR* knockdown (KD). Data are mean of $n = 3$ biological replicates ± s.d.; two-sided ***$P = 0.002$, **$P = 0.008$ versus corresponding no-insert sample, unpaired $t$-test with Welch's correction. Right, flow cytometry histograms showing expression from GFP lentiviral plasmids containing different untranslated sequences transfected into wild-type or *TASOR* KD 293T cells. gMFI, geometric mean fluorescence intensity. **b**, Top, genome browser track depicting input and H3K9me3 chromatin immunoprecipitation with sequencing (ChIP-seq) signal over the unique fragment of the SFFV-driven or promoterless L1 reporter integrated into wild-type and *TASOR* KO Hela cells. Bottom, ChIP–qPCR quantifying H3K9me3 and total histone H3 levels at a SFFV-driven or promoterless L1 lentiviral reporter integrated into wild-type and *TASOR* KO HeLa cells. Data are mean of $n = 3$ biological replicates (independent polyclonal integrations of the reporters) ± s.d.; ***$P = 0.0006$, **$P = 0.002$, *$P = 0.003$ versus wild-type promoter, paired two-tailed $t$-test. Red arrows indicate position of the primers used for subsequent quantitative PCR. **c**, RIP in *SETDB1* KO 293T cells with haemagglutinin (HA) tag knocked into *TASOR* or *PPHLN1* locus, showing periphilin and TASOR association with the indicated RNAs (see Extended Data Fig. 4k–m for more details). Data are mean ± s.d.; $n = 3$ independent experiments, normalized to input. **d**, Enrichment of periphilin RIP sequencing (RIP-seq) peaks at different repetitive elements in *SETDB1* KO (mix) cells. *SETDB1* KO (mix) is a polyclonal cell pool after *SETDB1* CRISPR–Cas9. Significant enrichment is defined as a fold change score above 1 with empirical Benjamini–Hochberg adjusted one-sided $P$-values ($q$); ***$q = 0.0002$ **e**, Genome browser tracks depicting periphilin and control RIP-seq signal over intronic L1 elements in wild-type and *SETDB1* KO (mix) cells.

cDNAs nor L1s are separated by long intragenic non-coding DNA regions (that is, introns) prompting us to investigate whether HUSH sensitivity was intron-dependent.

We compared HUSH repression of: (1) an intronless reporter in which iRFP is followed by non-coding ORF2 (iRFP-ORF2) and (2) an otherwise identical reporter with the second intron of human β-globin (*HBB* IVS2) inserted within the iRFP (Fig. 3a). Intron insertion abrogates HUSH-mediated repression (Fig. 3a, Extended Data Fig. 6b), and HUSH-mediated repression was also abolished by intron insertion at the 5′ or 3′ end of ORF2 (Fig. 3c, Extended Data Fig. 7c, d). Insertion of an antisense GFP 'stuffer' sequence had no effect (Fig. 3c, Extended Data Fig. 7a). This loss of HUSH repression was associated with decreased periphilin binding (Extended Data Fig. 7g–h) and decreased H3K9me3 deposition (Fig. 3b, Extended Data Fig. 7b). Intron-mediated HUSH protection was also observed for: (1) ORF2 reporters of different architecture expressed from an inducible, piggyBac transposon vector, (2) Cas9 reporters expressed from the piggyBac transposon vector, and (3) lentiviral reporters (Extended Data Fig. 6c–e), and was lost following Cre–*lox*-mediated deletion of an intron sequence from the integrated transgene, implying that the intron is required continuously to maintain protection (Extended Data Fig. 6f).

Four additional human introns (*EEF1A1*, *NXF1*, *SMC5* and *ACTB*) cloned into the iRFP-ORF2 reporter also provided protection from HUSH-mediated repression (Fig. 3d, f, Extended Data Fig. 7e), an effect not seen with a small artificial intron (chimeric β-globin–IgG), or reporters with similar-length control 'stuffer' sequences. The reduction in HUSH sensitivity correlated with the length of intron (Fig. 3f, Extended Data Fig. 7f). The *SMC5* intron, despite being poorly spliced, prevented HUSH-mediated repression more effectively than fully spliced *HBB* and *EEF1A1* introns (Fig. 3d, f, Extended Data Fig. 7e), suggesting that intron excision by the splicing machinery may not be required for protection against HUSH repression. To investigate whether splicing is required for intron-mediated protection, we generated a series of *HBB* IVS2 5′ and 3′ splice-site mutants (Extended Data Fig. 8a) which, despite effectively abolishing splicing (mutants no. 1 and no. 2), counteracted HUSH-mediated reporter repression as effectively as the wild-type

Fig. 4k, l). Notably, these results in SETDB1-deficient cells indicate that HUSH must bind reporter RNA prior to and independent of H3K9me3 deposition (Extended Data Fig. 4m). Transcription is therefore required for transgene repression, and periphilin binding to transgene RNA is likely to contribute to its recognition by HUSH.

To gain a global view of RNAs bound by endogenous periphilin (Extended Data Fig. 5a, b), we performed UV-cross-linked RIP and genome-wide analysis. Periphilin binding showed a significant overlap with genomic repeats, with specific enrichment over L1 elements (Fig. 2d, e, Extended Data Fig. 5c, d, f). There was no significant overlap between periphilin peaks and other repeat classes, with only transcripts of the Tigger DNA transposon family showing significant binding (Extended Data Fig. 5d, f). Periphilin preferentially bound transcripts from full-length, evolutionary young L1s (Extended Data Fig. 5e), reflecting the selective, genome-wide, HUSH-mediated H3K9me3 deposition over these L1 elements[2,3], as well as from other HUSH-targeted loci (Extended Data Fig. 5f, right). Periphilin recognition of nascent RNA therefore specifies target loci for HUSH repression.

## Introns protect against HUSH repression

We next investigated why transcribed cDNA sequences, but not their endogenous genomic loci (Fig. 1e, Extended Data Fig. 6a), are HUSH-repressed. A key difference is that coding regions of neither

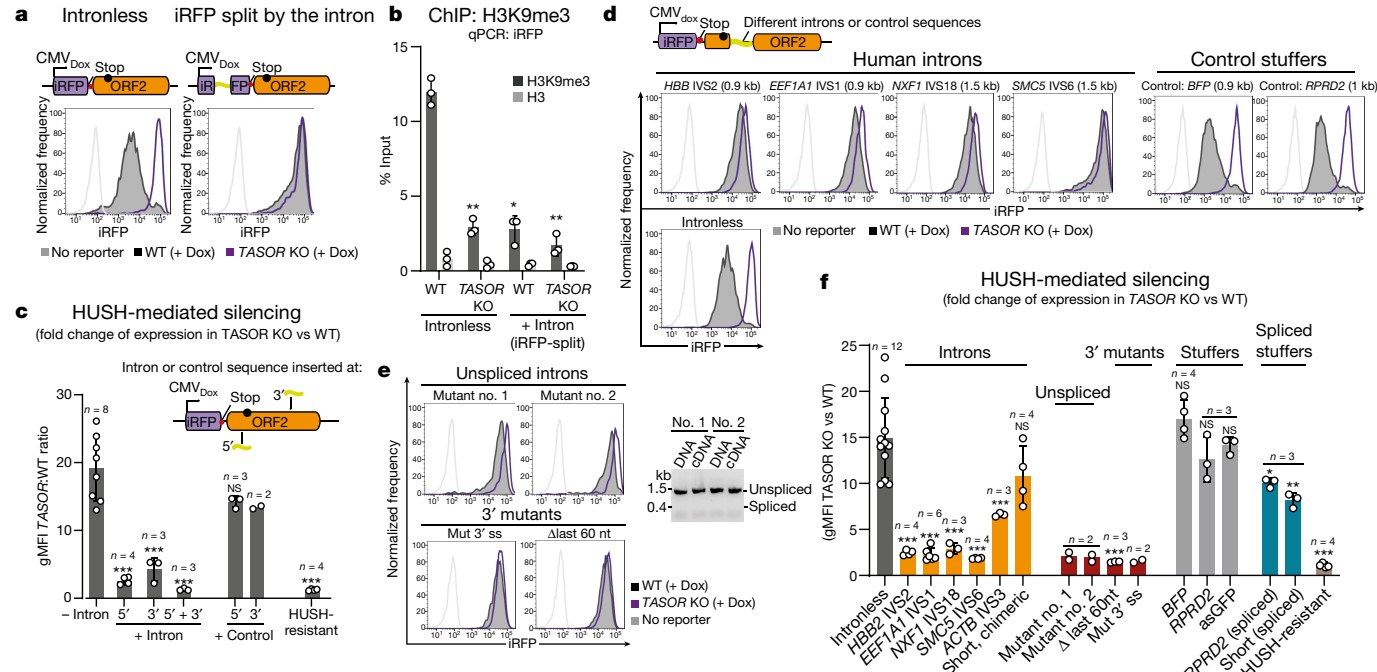

**Fig. 3 | Introns protect against HUSH, even in the absence of intron splicing.** HUSH-mediated repression of intronless and intron-containing iRFP-ORF2 piggyBac reporters. **a, b**, Second intron from the human β-globin gene (*HBB*IVS2) cloned within the iRFP gene. **a**, Flow cytometry histograms showing expression in wild-type and *TASOR* KO HeLa cells. **b**, ChIP–qPCR quantification of H3K9me3 and total histone H3 at reporters in wild-type and *TASOR* KO HeLa cells. Data are mean of *n* = 3 independent experiments ± s.d.; \*\**P* < 0.008, \**P* = 0.02 versus intronless wild type, ratio-paired two-tailed *t*-test. **c**, HUSH-mediated repression of reporter with intron(s) or control sequence cloned at the 5′ or 3′ of ORF2, measured by flow cytometry and shown as the ratio of reporter expression in *TASOR* KO and wild-type cells. Data are mean from *n* biological replicates ± s.d.; \*\*\**P* ≤ 0.0001, one-way analysis of variance (ANOVA) post hoc pairwise comparisons versus no-intron condition with Bonferroni correction.

**d**, Flow cytometry histograms showing expression from reporters with different introns from human genes or control sequences cloned at the 5′ end of ORF2. Intron size is shown in parentheses. BFP, blue fluorescent protein. **e**, Flow cytometry histograms showing expression from reporters with different *HBB* IVS2 mutant introns cloned 5′ of ORF2. Gel image (right) confirms that mutant introns are not spliced from the reporter. ss, splice site. **f**, Quantification of HUSH-mediated repression of reporters from Fig. 3d, e, Extended Data Fig. 8c by flow cytometry and calculated as the ratio of reporter expression in *TASOR* KO and wild-type HeLa cells. Data are mean of *n* biological replicates (independent polyclonal integrations of the reporters) ± s.d.; \*\*\**P* ≤ 0.0001, \**P* = 0.044, \*\**P* = 0.009, one-way ANOVA post hoc pairwise comparisons versus intronless with Bonferroni correction. asGFP data is the same as in 5′ control from **c**.

intron (Fig. 3e, f). Mutant intron no. 1 has a 5′ splice-site deletion critical for early spliceosome assembly at the transcript[22], suggesting that intron-mediated HUSH protection is independent of assembly of the core spliceosome at the transgene RNA. *HBB* IVS2 splice mutants with either a 3′ splice-site mutation or deletion of the last 60 nucleotides (including the branch-point site that pairs with the 5′ splice site to form a splicing intermediate) not only counteracted HUSH, but provided more effective protection from HUSH-mediated repression than the wild type intron (Fig. 3e, f, Extended Data Fig. 8a, b). Therefore, even in the absence of splicing, introns protect transgenes against HUSH-mediated repression, whereas effectively spliced stuffer sequences flanked by a 5′ splice site, a branch point and a 3′ splice site, did not counteract HUSH (Fig. 3f, Extended Data Fig. 8c). Thus it is the intron itself rather than the splicing process that protects against HUSH-mediated repression.

## HUSH targets endogenous intronless loci

Our data suggest that HUSH provides a genome-surveillance system to repress diverse transcribed, intronless invading DNAs, and predict that genomic loci from similar invading DNAs are bound and silenced by HUSH. Such loci include retrogenes and processed pseudogenes, created when reverse-transcribed cellular mRNA integrates into the genome, as part of a retrotransposition event[23]. We detected HUSH binding and HUSH-mediated H3K9me3 at the loci of transcribed processed pseudogenes and retrogenes, but not on their intron-containing, transcribed parent genes (Fig. 4a, Extended Data Fig. 9a–c, e). Many

HUSH-repressed pseudogenes and retrogenes are positioned within transcriptionally active genes, similar to HUSH-regulated L1s. The *MAB21L2* retrogene—a non-transcribed paralogue of the HUSH-repressed *MAB21L1* retrogene—is not HUSH-repressed, confirming the critical requirement for transcription in HUSH-mediated repression (Extended Data Fig. 9d).

Similarly, periphilin bound only retrotranscribed and not intron-containing parent genes (Extended Data Fig. 10a–c). Genomic analysis revealed that 20% of transcribed, non-L1-overlapping pseudogenes and 17% of intronless genes showed at least twofold enrichment of the periphilin RIP signal (Fig. 4b, Extended Data Fig. 10d). There was no enrichment of periphilin binding over intron-containing genes (Extended Data Fig. 10d), with the 5% of genes with bound periphilin predominantly containing HUSH-repressed long (over 2 kb) exons or zinc-finger family (ZNF) members as seen for HUSH-dependent H3K9me3[1,15] (Fig. 4b, Extended Data Figs. 9f, 10e). HUSH repression of processed pseudogenes and retrogenes—all bona fide endogenous mobile elements—emphasises the physiological role of HUSH in defending the genome against invading retroelements.

## Discussion

Our study reveals how the HUSH epigenetic repressor complex provides a versatile defence system against genome invasion. Without previous exposure to its targets, HUSH is able to recognize and transcriptionally repress a broad range of sequence-diverse, intronless

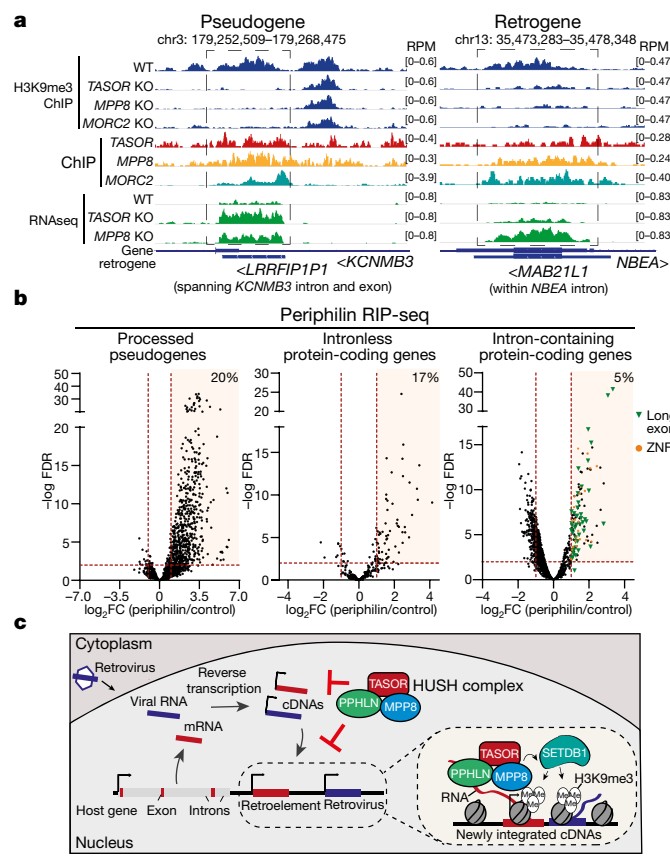

**Fig. 4 | Transcribed processed pseudogenes and retrogenes are bound and silenced by the HUSH complex. a**, Visualization of HUSH-dependent H3K9me3, HUSH–MORC2-occupancy and RNA sequencing in wild-type and *HUSH* KO K562 cells at representative loci of processed pseudogene and retrogene. Genome browser tracks are generated from publicly available BigWig files[35]. Arrowheads indicate transcriptional direction of the gene. **b**, Volcano plots showing log$_2$ fold change (log$_2$ FC) of periphilin over control RIP-seq-normalized read counts for three gene categories: processed pseudogenes (left), intronless (middle) and intron-containing protein-coding genes (right); representative data from *SETDB1* KO (mix) cells. For each data point, significance was determined after a comparative assessment of counts between conditions using negative binomial generalized linear models as implemented in edgeR. Multiple testing correction of significance was performed using the false discovery rate (FDR) method; $n$ = 4 independent experiments. Only genes with periphilin RIP-seq signal greater than 0.3 RPKM are included (>0.3 reads per kilobase of transcript, per million mapped reads (RPKM) in each RIP-seq replicate from both *SETDB1* KO (mix) and wild-type 293T cells). Genes with periphilin peaks overlapping L1 elements are excluded. Intronless protein-coding genes are defined as genes that produce only intronless isoforms. ZNFs, zinc finger family genes. **c**, Schematic of genome surveillance by the HUSH complex. HUSH recognizes long, intronless mobile DNA and targets it for transcriptional silencing. Host genes are protected against HUSH by the presence of introns (left of DNA strand). An average human protein-coding gene contains ten 6,355-bp-long introns[35]. Transcription of the target initiates HUSH-mediated repression: periphilin binds its specific target transcript, MPP8 recruits SETDB1 to deposit H3K9me3. Periphilin–RNA and MPP8–H3K9me3 interactions anchor HUSH at the target locus (area with dashed outline, right).

DNAs, whereas intron-containing DNAs are resistant to HUSH-mediated repression. The defining feature of HUSH targets is therefore the presence of long, intronless transcription units, an intrinsic feature of retroelements, including L1 retrotransposons. Non-reverse-transcribed, intronless invading DNAs are also targeted for repression, including transfected cDNA plasmids. HUSH is therefore 'programmed' to control the spread of integrating, RNA-derived mobile elements within the host genome, representing a universal, cell-autonomous genome-surveillance system (Fig. 4c). The HUSH-mediated repression of endogenous L1s[2,3,15] is a consequence of this programming rather than a recognition of unique L1 sequences. Genomic evidence for HUSH repression of sequence-diverse, retrotransposition-derived, endogenous genes supports this conclusion and validates our findings with reporter genes. HUSH specificity for target length and A-rich bias in the sense strand may reflect retroviral reliance on 'structurally poor' A-rich RNA sequences to support viral cDNA synthesis during reverse transcription[24] and may therefore allow a more selective targeting of reverse-transcribed elements. Moreover, HUSH silencing of transgenes, including most cDNAs larger than about 1.5 kb, explains why many cDNAs remain difficult to express, a practical problem in both gene therapy and in ectopic gene expression in cultured cells.

The dependence of HUSH-mediated repression on transcription is reminiscent of transcription-coupled heterochromatin formation in *Schizosaccharomyces pombe*[25], where, as with HUSH, transcription is required for both the initiation and propagation of H3K9me3. The association of periphilin with its target RNAs even in the absence of H3K9me3 deposition provides support for a critical role of RNA in HUSH-mediated repression. Binding of periphilin to nascent RNA provides specificity for target recognition by recruiting and stabilizing HUSH at target loci independent of the MPP8 chromodomain, and enables HUSH to respond to increased transcription if H3K9me3 levels decline, such as during cell division. Similar to *S. pombe*, transcription-induced recruitment of HUSH to replicated chromatin may ensure inheritance of the repressed state following DNA replication[26,27]. This requirement for active transcription explains preferential targeting of full length L1s in euchromatic environments by HUSH, and conversely, why HUSH ignores older, degenerate L1s that have lost transcriptional activity[2,3,15].

Importantly, intron-mediated protection from HUSH-mediated silencing does not require efficient intron splicing or spliceosome recruitment. Given the complex network of RNA-binding proteins involved in exon–intron definition and splicing[28], intronic sequences may counteract HUSH by recruiting proteins other than core splicing factors that compete with periphilin for transcript binding. Alternatively, HUSH may be sensitive to nucleosome distribution, with the increased occupancy over exons versus introns[29,30] correlating with reduced elongation rates[29–31]. Slow elongation through long exons may trigger HUSH recruitment, which is counteracted by the decreased nucleosome density and increased elongation in cellular introns, consistent with HUSH-mediated H3K9me3 deposition over long exons of endogenous genes. Shorter introns are much less likely to affect nucleosome positioning (with each nucleosomes occupying 147 nt) than longer introns, consistent with the limited or absent HUSH protection afforded by the short *ACTB* and very short artificial intron. The well-recognized ability of introns to enhance gene expression (intron-mediated enhancement) can, at least in part, be explained by the capacity of introns to protect transgenes from HUSH-mediated silencing[32].

To distinguish self from non-self, the host immune system recognizes conserved molecular patterns that are maintained in invading pathogens but are absent from the host. Most mammalian genes are organized such that exons comprise small islands within a sea of intronic sequences, whereas the cDNA products of reverse transcription are RNA-derived and intronless. Long, intronless cDNA, the product of reverse transcription, is therefore the molecular pattern recognized by HUSH, which provides a means to distinguish invading retroelements from host genes. Thus, HUSH comprises a component of the innate immune system. To avoid HUSH recognition, retroelements would need to maintain long, non-coding intron sequences, but are constrained by selective pressure for a compact genome. Bypassing the restriction imposed by HUSH therefore poses a major challenge. Whereas retroviral transcripts are often spliced, the intervening sequences are coding

sequences and very different from the classical long non-coding introns of cellular genes. Consequently, primate lentiviruses evade HUSH by encoding accessory proteins that degrade HUSH[10–12], whereas endogenous retroelements are unable to evade HUSH activity.

The innate immune response provides immediate defence but does not confer long-lasting immunity. HUSH-selective targeting of evolutionary young L1s[2,3] suggests a limited ability to provide long-lasting repression over evolutionary timescales. By contrast, DNA sequence-specific KRAB-ZFPs are less agile in repressing young retroelements, as it takes several million years to evolve a KRAB-ZFP with high affinity for a new DNA sequence[33,34]. By rapidly repressing transcription of novel retroelements without the need for genetic memory, HUSH buffers any potentially deleterious effects on cellular fitness. This gives the host a time window to establish sequence-specific adaptive repression to effectively restrict these retroelements and may facilitate their domestication.

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

## Methods

### Plasmids

A list and details of all plasmids used in the study are in Supplementary Table 1.

### Cell culture

HeLa cells were obtained from ECACC and HEK 293T and Jurkat cells were from ATCC. Cell morphology was assessed for authentication. All cell lines were grown in IMDM plus 10% FCS and penicillin/streptomycin (100 U ml⁻¹). Cell cultures were routinely tested and found to be negative for mycoplasma infection (MycoAlert, Lonza).

### Antibodies

Antibodies for immunoblotting: rabbit anti-TASOR (Atlas, HPA006735, 1:5,000), rabbit anti-MPP8 (Proteintech, 16796-1-AP, 1:5,000), rabbit anti-periphilin1 (Sigma-Aldrich, HPA038902, 1:5,000), rabbit anti-MORC2 (Bethyl Laboratories, A300-149A, 1:5,000), rabbit anti-SETDB1 (Proteintech, 11231-1-AP; 1:5,000), rat anti-haemagglutinin (HA) tag (3F10, Sigma-Aldrich, 11867423001, 1:10,000), mouse anti-β-actin peroxidase conjugate (Sigma-Aldrich, A3854; 1:20,000), mouse anti-p97 (Abcam, ab11433, 1:5,000), rabbit anti-α-tubulin (11H10, CST, 2125, 1:5,000). Horseradish peroxidase (HRP)-conjugated secondary antibodies for immunoblotting were obtained from Jackson ImmunoResearch: Peroxidase AffiniPure Goat Anti-Mouse IgG (H+L) (115-035-146, 1:10,000), Peroxidase AffiniPure Goat Anti-Rabbit IgG (H+L) (111-035-144, 1:10,000), Peroxidase AffiniPure Goat Anti-Rat IgG (H+L) (112-035-143, 1:10,000). Antibodies for intracellular staining for flow cytometry: mouse anti-HA tag Alexa Fluor 647 conjugate (Cell Signaling, 3444; 1:50; used only for PPHLN1–HA and HA–TASOR knockin validation). Antibodies for ChIP–qPCR: rabbit anti-H3K9me3 (Abcam, ab8898) 5 μg per immunoprecipitation, rabbit anti-histone H3 (Abcam, ab1791) 5 μg per immunoprecipitation and rabbit anti-RNA Pol II (Bethyl Laboratories, A304-405A, 7.5 μg per immunoprecipitation).

### CRISPR–Cas9 mediated gene disruption

HeLa or HEK 293T cells were transfected with a pool of sgRNAs cloned into a Cas9-containing plasmid (pSpCas9(BB)-2A-Puro) using TransIT HeLa Monster or TransIT 293T (Mirus) according to the manufacturer's protocol. Transfected cells were enriched with 24 h of puromycin selection (2 μg ml⁻¹) starting 24 h after transfection. Hela *TASOR* KD, HEK 293T *TASOR* KD and *SETDB1* KO (mix) cell lines were maintained as mixed KO populations. *HUSH*, *SETDB1* and *MORC2* KO HeLa cells were generated as described[1,13] and are polyclonal KO populations derived from a HeLa clone harbouring a repressed GFP reporter (pHRSIN-p_SFFV-GFP-WPRE-P_GK-Zeo^R) integrated at pericentromeric site on chromosome 7: 57848728 (hg19). Parental HeLa cells are GFP⁻ and HUSH, SETDB1 and *MORC2* KO cells are GFP⁺ because of de-repression of the GFP reporter.

### Lentiviral production and transduction

Lentivirus was produced by transfecting HEK 293T cells with the lentiviral vector plus the packaging plasmids pCMVΔR8.91 and pMD2.G using TransIT-293 transfection reagent (Mirus). The viral supernatant was collected 48 h later, cell debris was removed with a 0.45-μm filter and target cells transduced by spin infection at 1,800 rpm for 60 min. Transduced HeLa cells were selected with the following drug concentrations: puromycin, 2 μg ml⁻¹; hygromycin, 100 μg ml⁻¹; and blasticidin, 5 μg ml⁻¹. For experiments with non-integrated virus, cells were transduced in the presence of 1 μM raltegravir.

For the 'one-pot' establishment assay, WT HeLa cells were initially transduced with lentiviral vector encoding mCherry (pHRSIN-p_SFFV-mCherry-WPRE) at a multiplicity of infection (MOI) <1 and mCherry⁺ cells were purified by fluorescence-activated cell sorting (FACS), resulting in 98% pure mCherry⁺ populations (Supplementary Figure 2). mCherry⁺ WT and mCherry⁻ *TASOR* KD cells were mixed at a 1:1 ratio and transduced

with the lentiviral GFP reporters by spin infection. Reporter expression was typically analysed 2, 4 and 6 days after transduction by flow cytometry. Gating strategy is depicted in Extended Data Fig. 2c, Supplementary Fig. 2. Reciprocal mixing (mCherry⁺ *TASOR* KD and mCherry⁻ WT) was used to validate results.

### Transfection

WT mCherry⁺ and *TASOR* KD mCherry⁻ HEK 293T cells were mixed at a 1:1 ratio and transfected using TransIT-293T (Mirus) according to the manufacturer's protocol.

### PiggyBac-mediated integration of reporter constructs

HeLa or HEK 293T cells were co-transfected with pB-transposon plasmid and piggyBac transposase-expression plasmid at 5:1 or 2.5:1 ratio using TransIT-HeLa Monster or TransIT-293T (Mirus). Transfected cells were selected with blasticidin (5 μg ml⁻¹) for at least 3 days starting from 2 days after transfection. For flow cytometry assays, two cell lines were mixed at a 1:1 ratio prior to transfection. For assays with GFP reporters, WT mCherry⁺ cells were mixed with *TASOR* KD mCherry⁻ HeLa cells. For assays with iRFP reporters, WT GFP⁻ HeLa cells were mixed with *TASOR* KO GFP⁺ HeLa cells, which both harbour additional a HUSH-sensitive GFP reporter at chr7:57848728 (hg19). See Supplementary Figure 2 for gating strategies in flow cytometry analyses. Reporter expression was typically analysed 7 and 12 days after transfection and was induced by plating cells in media with doxycycline (1 μg ml⁻¹) 24 h prior to flow cytometry analysis or ChIP–qPCR.

### Flow cytometry

Live cells were analysed on a LSR Fortessa (BD). Data were analysed using FlowJo v10.6.1 (LCC) software. Cell sorting was carried on a FACSAria Fusion (BD).

### Immunoblotting

Cells were lysed in 100 mM Tris pH 7.4 with 1% SDS followed by boiling and vortexing to shear genomic DNA. Lysates were then boiled in SDS sample buffer, separated by SDS–PAGE and transferred to PVDF membranes (Millipore). Membranes were probed with the indicated antibodies and reactive bands visualised with ECL, Supersignal West Pico or West Dura (Thermo Scientific).

### CRISPR–Cas9 mediated knock-in of HA tag

For C-terminal periphilin tagging, the HA sequence was inserted upstream of the stop codon at the *PPHLN1* endogenous locus via CRISPR homology-directed repair. For N-terminal TASOR tagging, HA was inserted downstream of the *TASOR* start codon. Single-stranded donor oligonucleotides (ssODN) were used as donor templates and purchased from IDT. HEK 293T cells were transfected with single guide RNA (sgRNA) plasmid (pSpCas9(BB)-2A-Puro) and single-stranded donor template. Transfected cells were enriched by puromycin selection and single-cell cloned. Clonal populations were screened for the presence of HA tag by intracellular flow cytometry staining using anti-HA antibody. The genetic modifications were validated by PCR on genomic DNA followed by sequencing. sgRNA and ssODN sequences listed in Supplementary Table 1.

### CRISPR–Cas9 mediated deletion of *TAF7* promoter

Prior to the modification of the *TAF7* locus, HeLa cells were transduced with lentivirus encoding codon-optimized C-terminally HA-tagged TAF7 (TAF7_(opt)–HA) and blasticidin resistance as a selection marker. *TAF7* is an essential gene[36] and stable expression of exogenous TAF7_(opt) was used to compensate for the loss of expression from the endogenous *TAF7* locus due to promoter deletion. Sequence was codon-optimized so that exogenous *TAF7*_(opt) was not detected in RT–qPCR or ChIP–PCR.

Two sgRNAs targeting the *TAF7* promoter region were cloned into pSpCas9(BB)-2A-Puro (PX459,V2.0): one targeting within the first 80 nucleotides of the *TAF7* 5′ UTR and a second approximately 850 nt upstream of the transcription start site. Two sgRNA plasmids were

mixed at a 1:1 ratio and transfected into HeLa TAF7$_{(opt)}$–HA-expressing cells. Twenty-four hours later, cells were treated with puromycin (2 µg ml$^{-1}$) for 24 h and single-cell cloned 5 days after transfection. The genetic deletion effects were validated by PCR on genomic DNA and loss of *TAF7* expression measured by RT–qPCR. Sequences of primers and sgRNAs are detailed in Supplementary Table 1.

### Chromatin immunoprecipitation

Cells were cross-linked in 1% formaldehyde for 10 min, quenched in 0.125 M glycine for 5 min and lysed in cell lysis buffer (1 mM HEPES, 85 mM KCl and 0.5% NP-40). Nuclei were pelleted by centrifugation and then lysed in nuclear lysis buffer (5 mM Tris, 10 mM EDTA and 1% SDS) for 10 min. The chromatin was sheared with a Bioruptor (Diagenode Pico) to obtain a mean fragment size of <300 bp. Insoluble material was removed by centrifugation. The chromatin solution was diluted to a final SDS concentration of 0.1% and precleared with Pierce Protein G magnetic beads (Thermo Fisher) and then immunoprecipitated overnight with 5 µg primary antibody and Protein G–magnetic beads. Beads were washed twice with low-salt buffer (20 mM Tris pH 8.1, 2 mM EDTA, 50 Mm NaCl, 1% Triton X-100, 0.1% SDS), once with high-salt buffer (20 mM Tris pH 8.1, 2 mM EDTA, 500 mM NaCl, 1% TritonX-100, 0.1% SDS), once with LiCl buffer (10 mM Tris pH 8.1, 1 mM EDTA, 250 mM LiCl, 1% NP-40, 1% sodium deoxycholate) and twice with TE. Protein–DNA complexes were eluted in 150 mM NaHCO$_3$ and 1% SDS at 65 °C. Cross-links were reversed by overnight incubation at 65 °C with 0.3 M NaCl and RNase A. Proteinase K was then added, the samples were incubated for 2 h at 45 °C, and then the DNA was purified with a spin column (Qiagen PCR Purification Kit). Quantification by qPCR was performed on a QuantStudio 6 Flex Real-Time PCR System (Thermo Fisher Scientific) using SYBR green PCR mastermix (Thermo Fisher Scientific). qPCR primer sequences are detailed in Supplementary Table 1.

For ChIP-seq, immunoprecipitated DNA was subjected to library preparation (NEBNext Ultra II DNA Library Prep Kit, Illumina). Libraries were purified, quantified, multiplexed (with NEBNext Multiplex Oligos for Illumina kit, E7335S) and sequenced with 2× 50-bp pair-end reads on Illumina Novaseq platform (Genomics Core, Cancer Research UK Cambridge Institute).

Bioinformatics data processing and analyses were performed using Bash (v4.2.46), R (v3.6) and Python (v3.8.5) programming languages as well as the following tools: FastQC (Babraham Bioinformatics) (v0.11.7) cutadapt[37] (v1.16), HISAT2[38] (v2.1.0), SAMtools[39] (v1.9), sambamba[40] (v0.6.6) and deepTools[41] (v3.1.0). Raw fastq files were quality checked with FastQC and trimmed with cutadapt to remove adapter sequences and low-quality base calls (quality score <20). Depending on the experiment, the resulting reads were aligned using HISAT2 to either the human reference genome only (version GRCh38) or the human reference genome concatenated with the sequence of the unique fragment from reporter construct (P2A-iRFP), duplicates were marked using sambamba and alignments were formatted using SAMtools. BigWig files containing genomic signal were computed at single -base resolution and normalized to counts per million (CPM) using deepTools. Further details are available in the GitHub page of this study (https://github.com/semacu/hush).

### Native RIP–qPCR

Reporter expression was induced by doxycycline (1 µg ml$^{-1}$) for 24 h prior to the experiment. Cells were lysed in HLB-N buffer (10 mM Tris-HCl (pH 7.5), 10 mM NaCl, 2.5 mM MgCl$_2$ and 0.5% NP-40), incubated on ice for 5 min and lysate was underlaid with 1/4 volume of HLB + NS (10 mM Tris-HCl (pH 7.5), 10 mM NaCl, 2.5 mM MgCl$_2$, 0.5% NP-40 and 10% (wt/vol) sucrose). Nuclei were pelleted by centrifugation (420*g*, 5 min) and then lysed in RIP buffer (25 mM Tris pH 7.4, 150 mM KCl, 5 mM EDTA, 0.5 mM DTT, 0.5% NP-40 and 100 U ml$^{-1}$ SUPERase-IN). The nuclear fraction was sonicated (Diagenode Pico) and insoluble material was removed by centrifugation (8,000*g*, 10 min). The nuclear fraction was immunoprecipitated with Pierce anti-HA magnetic beads

(Thermo Fisher) for 2 h at 4 °C. Beads were washed four times with RIP buffer and RNA was extracted from beads (and input samples) using TRIzol and standard phenol-chloroform extraction. The aqueous phase containing the RNA was loaded onto RNeasy mini columns (QIAGEN) with 2 volumes of 100% ethanol and RNA was purified according to the manufacturer's protocol. RNA was on-column DNase I treated and reverse transcribed using random hexamers and SuperScript III Reverse Transcriptase (Thermo Fisher Scientific). Quantification by qPCR was performed on QuantStudio 6 Flex Real-Time PCR System (Thermo Fisher Scientific) using SYBR green PCR mastermix (Thermo Fisher Scientific). qPCR primers sequences are detailed in Supplementary Table 1.

### UV-crosslinked RIP-seq

Cells were UV treated (254 nM UV-C at 0.3 J cm$^{-2}$) in PBS, lysed in HLB-N buffer, incubated on ice for 5 min and lysate was then underlaid with 1/4 volume of HLB + NS. Nuclei were pelleted by centrifugation (420xg, 5 min) and lysed in RIP buffer: (25 mM Tris pH 7.4, 150 mM KCl, 5 mM EDTA, 0.5 mM DTT, 0.5% NP-40 and 100 U/ml RNasin (Promega)). The nuclear fraction was sonicated (Diagenode Pico), treated with TURBO-DNase (4U), and insoluble material was removed by centrifugation (8,000*g*, 10 min). The nuclear fraction was immunoprecipitated with Pierce anti-HA magnetic beads (Thermo Fisher) for 2 h at 4 °C. Beads were washed once with RIP buffer, once with RIP buffer + TURBO-DNase (2U), 2× RIPA buffer (50 mM Tris pH 7.4, 100 mM NaCl, 1% NP-40, 0.5% sodium deoxycholate, 0.1% SDS), 1× high-salt RIPA (50 mM Tris-HCl pH 7.4, 500 mM NaCl, 1 mM EDTA, 1% NP-40, 0.1% SDS, 0.5% sodium deoxycholate), 1× low-salt wash (15 mM Tris-HCl pH 7.4, 5 mM EDTA), for 5 min each time at room temperature with rotation. Beads were digested with proteinase K in proteinase K buffer (50 mM Tris-Cl (pH 7.5), 100 mM NaCl, and 1 mM EDTA, 0.25% SDS) and RNA was isolated by standard phenol-chloroform extraction. RNA from the first RIP-seq experiment in *SETDB1* KO (mix) was in addition rigorously treated with TURBO-DNaseI prior to library preparation. Immunoprecipitated RNA was subjected to DNA library preparation using SMARTer Stranded Total RNA-Seq Kit V3–Pico Input Mammalian (Takara Bio) according to the manufacturer's instructions with initial fragmentation at 94 °C for 3 or 4 min and ribosomal RNA depletion step included. The library quality was determined using Bioanalyzer, and sequenced on Illumina MiniSeq platform as paired-end 32-bp and 43-bp reads using MiniSeq High-Output 75 cycles kit.

Bioinformatics data processing and analyses were performed using Bash, R (v3.6.0) and Python (v3.8.5) programming languages as well as the following tools: FastQC (Babraham Bioinformatics) (v0.11.7), UMI-tools[42] (v1.1.1), cutadapt[37] (v1.16), HISAT2 (v2.1.0)[38], SAMtools (v1.9)[39], deep-Tools[41] (v3.1.0), BEDTools[43] (v2.30.0), data.table (v1.13.2), GenomicFeatures[44] (v1.38.2), edgeR[45,46] (v3.28.1), and GAT[47] (v1.0). Raw fastq files were quality checked with FastQC, unique molecular identifiers extracted using UMI-tools and resulting reads trimmed with cutadapt. Alignments to the human reference genome (version GRCh38) were performed with HISAT2, then formatted and deduplicated using SAMtools and UMI-tools respectively. Peaks were called using a customised approach involving BEDTools, deepTools, several Bash commands, datatable and edgeR. Genomic repeats were obtained from RepeatMasker and L1Base[48,49] and associations with the RIP-seq peaks were investigated using GAT and BEDTools. Tables integrating gene information, RIP-seq signal and repeats were obtained using BEDTools, data.table, GenomicsFeatures and edgeR. Finally combined bigWig files containing genomic signal were prepared with SAMTools and computed at single base resolution and normalized to CPM using deepTools. More details available in the GitHub page of this study http://github.com/semacu/hush.

### Northern blot

Sample preparation, agarose gel separation and transfer to the membrane were all performed using a NorthernMax Kit (Invitrogen) according to the manufacturer's recommendation. In brief, 1–10 µg of sample RNA or 2 µg Millennium Markers (Invitrogen) were suspended in formaldehyde

loading dye and loaded onto a 6-mm-thick 1% Agarose-LE gel and run at 5 V cm$^{-1}$ (150 V, 110 min) in 1× MOPS running buffer. The samples were transferred to a BrightStar–Plus positively charged nylon membrane (Invitrogen) over 120 min, via the described downward transfer apparatus stacked on paper towels. Following transfer, the membrane was UV (254 nm) cross-linked using 120 mJ energy (Stratagene, Stratalinker 1800) and photographed under UV to record the marker positions (Invitrogen, iBright CL1000 Imaging System). Following a 30 min, 68 °C, prehybridization in ULTRAhyb ultrasensitive hybridization buffer, the membrane was incubated overnight at 68 °C with 100 pM digoxigenin-labelled RNA probes, directed against iRFP (nucleotides 4–300) and *ACTB* (nucleotides 69–618 of mRNA, NM_001101). Membrane was washed with 1× low stringency wash solution (room temperature) and 2× NorthernMax high stringency wash buffer (68 °C), prior to blocking at room temperature with 1× casein blocking buffer (Sigma-Aldrich). The membrane was incubated for 60 min with 50 mU ml$^{-1}$ anti-digoxigenin-POD (poly), Fab fragments (Roche) in 1× blocking buffer, followed by 4 washes in 1× PBS + 0.1% Tween 20 and visualised using a SuperSignal West chemiluminescent substrate (Thermo Fisher) and the Invitrogen, iBright CL1000 Imaging System.

Primers used to generate PCR amplicons against the indicated regions of each gene are listed in Supplementary Table 1. The amplicons were used in a T7 polymerase reaction substituting the NTPs for DIG RNA labelling mix (Roche), to generate antisense digoxigenin labelled RNA probes. The reaction was digested with TURBO DNase (Invitrogen) for 15 min at 37 °C, before purification using an RNeasy MinElute cleanup kit (Qiagen).

### RT–qPCR

Total RNA was extracted using the RNeasy Plus Mini kit (Qiagen) with on-column DNase I treatment according to the manufacturer's instructions. RNA was reverse transcribed into cDNA using an equimolar mixture of random hexamers and oligo (dT)$_{16}$ primers by SuperScript III Reverse Transcriptase (Thermo Fisher Scientific). RNA quantification was performed using the $\Delta\Delta C_t$ method and normalized against *ACTB* or *GAPDH* transcript levels. Primer sequences are detailed in Supplementary Table 1.

### Analysis of splicing

Efficiency of splicing of the reporter transcripts were determined by semi-quantitative PCR using intron-flanking primers (see Supplementary Table 1) detecting both unspliced and spliced reverse-transcribed mRNA. cDNA was prepared as for RT–PCR. Corresponding plasmids served as DNA controls.

### Statistics and reproducibility

Statistical details, including the statistical test used, type (one- or two-sided), adjustments for multiple comparison and sample sizes (*n*), are reported in the figures and figure legends. The following figure panels show representative data from at least two independent experiments that showed similar results: Fig. 3e, Extended Data Figs. 1b, e, i, 2a, g, k, 3a, c, d, 4d, l, m, 5b, f, 6b, f, 7g. The following figure panels show representative data from at least three independent biological replicates that showed similar results: Figs. 1d, 2a, right, 3a, d, Extended Data Figs. 1f, 2d, h, 4b, c, k, 6c, 7c, e, h, 8b, c. The following figure panels show representative data from at least four independent biological replicates that showed similar results: Fig. 1a, b, e, Fig. 2e, Extended Data Figs. 2d, 3b, f, 10a, b, e. The experiments in Fig. 1d and Extended Data Fig. 1c were performed once, but where internally controlled for both positive and negative results. The Northern blot experiments in Extended Data Figs. 1h, j, 3g, 4f, 7d were performed once, but were internally controlled for both positive and negative results. The ChIP-seq experiments in Fig. 2b (top) and Extended Data Fig. 4h, j were performed once, but the results were independently validated by two independent ChIP–qPCR experiments.

### Reporting summary

Further information on research design is available in the Nature Research Reporting Summary linked to this paper.

## Data availability

Gels and blots source images are provided in Supplementary Fig. 1. Next-generation sequencing data have been deposited at the Gene Expression Omnibus (GEO) with accession number GSE181113. The publicly available data[2] are available at GEO under accession number GSE95374 (ChIP-seq and RNA-sequencing data on the HUSH complex). The version of the human reference genome used in this study is GRCh38 (GENCODE v35, https://www.gencodegenes.org/human/). Repeats were obtained from RepeatMasker (v UCSC hg38) and L1Base[48,49]. Source data are provided with this paper.

## Code availability

For details about the bioinformatics data analyses, check the GitHub page for this study at http://github.com/semacu/hush.

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

**Acknowledgements** We thank the Genomics Core at Cancer Research UK Cambridge Institute for next-generation sequencing; JCBC and CIMR Core Facilities, particularly R. Schulte, A. P. Harrison and their teams for assistance with FACS; J. Cohen-Gold and E. Greenwood and other members of the Lehner lab for helpful discussions; J. Wysocka for L1-GFP and (opt)-L1-GFP plasmids; and S. Menzies for generation of TASOR and PPHLN HA KI cell lines. This work was supported by the Wellcome Trust Principal Research Fellowship to P.J.L. (210688/Z/18/Z). M.S. was supported by a Boehringer Ingelheim Fonds PhD fellowship (2017-2020). The work was further supported by the NIHR Cambridge Biomedical Research Centre.

**Author contributions** M.S. designed, performed and analysed all experiments except from Northern blots and RT-PCR for GFP reporters which were conducted by S.B. S.M.C. performed all bioinformatics analyses with input from M.S. M.S. and P.J.L. conceived the study and wrote the manuscript.

**Competing interests** S.M.C. is an employee of AstraZeneca. All other authors declare no competing interests.

**Additional information**
**Correspondence and requests for materials** should be addressed to Paul J. Lehner.

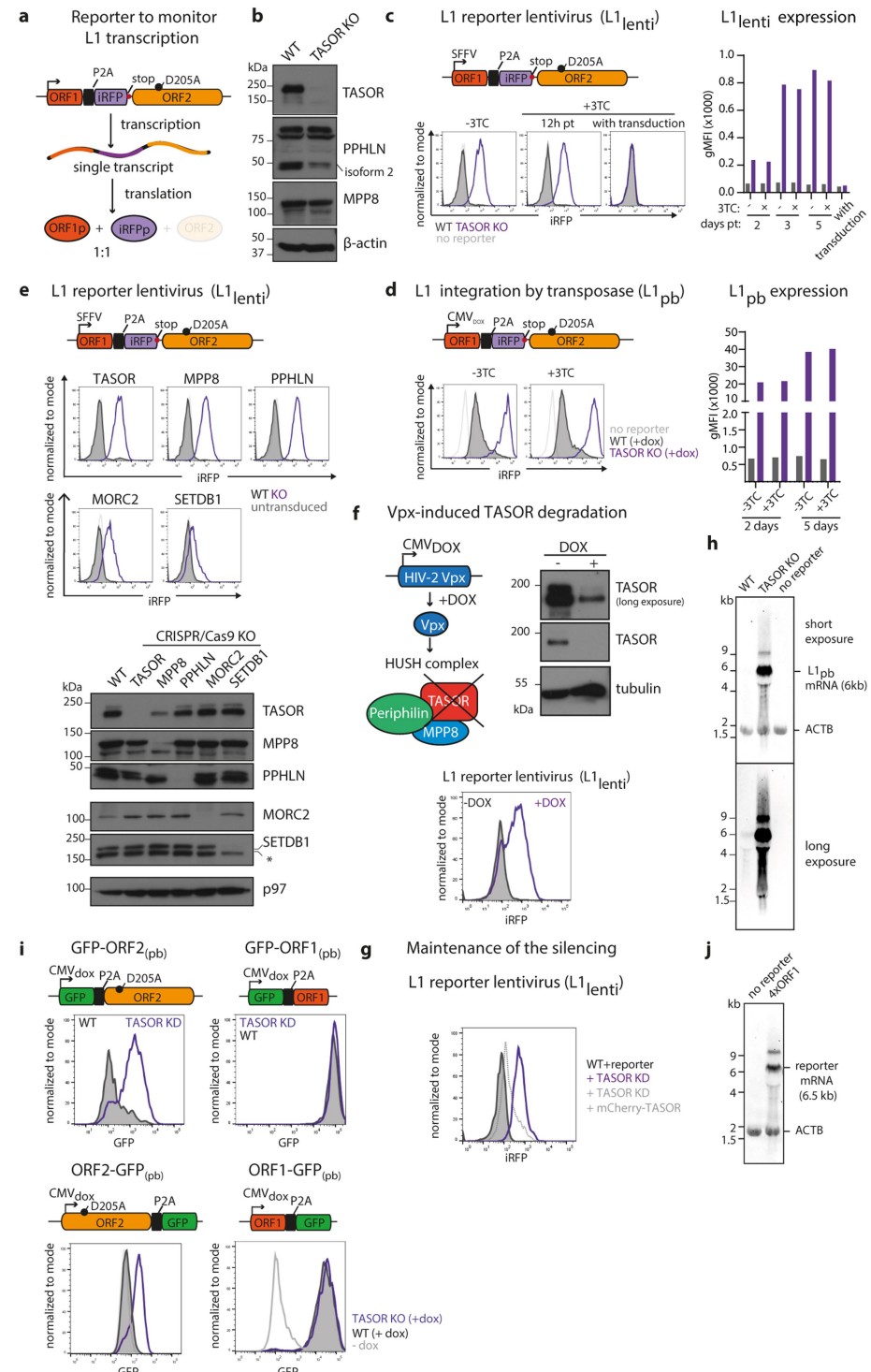

**Extended Data Fig. 1** | See next page for caption.

**Extended Data Fig. 1 | HUSH repression of the L1 transgene is independent of the integration mode and site and is due to the L1 ORF2 sequence. a**, Schematic of L1-iRFP reporter. The single mRNA transcript generates two proteins due to peptide bond skipping at the P2A sequence: the ORF1 (ORF1p) and iRFP (iRFPp). Changes in reporter transcription (e.g. due to H3K9me3-mediated silencing) affect iRFP expression. **b**, Western blot validating TASOR KO in HeLa cells with decreased levels of HUSH subunits periphilin and MPP8 due to TASOR depletion. β-actin is the loading control. **c,d**, Effect of reverse transcriptase inhibitor 3TC on the expression from L1 lentivirus (L1lenti) and L1 reporter integrated by transposase (L1pb). To validate that our reporters monitor expression only from the initial L1 integration, and no subsequent retrotransposition activity, we compared reporter expression in the presence and absence of reverse transcriptase inhibitor 3TC, which prevents retrotransposition and new L1 insertions. If expression from new insertions contributes to the iRFP signal, 3TC should decrease iRFP, in particular at time points ≥3 days. No such decrease in iRFP signal is observed with L1lenti and L1pb upon 3TC treatment, demonstrating that both reporters monitor expression from the initial L1 integration. As lentiviral integration requires reverse transcription, the 3TC was added 12h post transduction when reverse transcription will be complete. **c**, Flow cytometry histograms of expression of L1 lentivirus upon 50μM 3TC treatment in WT or TASOR KO cells (left). 3TC was added 12h post transduction and expression measured at day 5 post transduction. As a positive control, 3TC was used at the time of transduction (c – right hand panel). The absence of iRFP signal confirms inhibition of RT activity. Quantification of expression using geometric mean fluorescence intensity (gMFI) (right). **d**, Flow cytometry histograms of expression of L1pb reporter in the absence or presence of 50μM 3TC after 5 days of dox induction (left). Quantification of expression using gMFI (right). **e**, Establishment of repression of L1 reporter lentivirus in WT and HUSH KOs: TASOR, MPP8 or periphilin KO HeLa cells, and KOs of HUSH-effectors MORC2 and SETDB1. HUSH/HUSH effector KO (all GFP+) were mixed with WT cells and transduced with the L1-iRFP reporter at high multiplicity of infection (MOI). Expression of the reporter was measured by flow cytometry 72h post transduction. Western blot validating KOs with p97 as a loading control (bottom panel). * marks non-specific band. **f**, Schematic of lentiviral vector driven by dox-responsive promoter for expression of Human Immunodeficiency virus 2 (HIV-2) viral protein X (Vpx) to induce TASOR depletion (left panel). Western blot validating TASOR depletion 6 days after Vpx induction by dox with tubulin as a loading control (right panel). Flow cytometry histogram showing expression of integrated L1 lentivirus reporter before and after TASOR depletion by Vpx (bottom panel). **g**, Expression of L1 reporter lentivirus in: WT HeLa (WT+reporter) (grey histogram), HeLa cells in which TASOR was depleted after the integration of L1 lentivirus (+TASOR KD) (purple histogram) and re-expression of mCherry-TASOR (+TASOR KD +mCherry-TASOR) (grey, dotted histogram). **h**, Northern blot showing increased mRNA from L1pb reporter in TASOR KO cells 24 h post dox induction using iRFP probe. Full-length L1pb mRNA is the predominant RNA produced from the L1pb reporter. **i**, The expression of reporters with ORF2 or ORF1 sequences placed downstream of GFP in WT or TASOR-depleted (TASOR KD; see Extended Data Fig. 2a) HeLa cells measured by flow cytometry (upper panel). HUSH-mediated repression of reporters with ORF2 or ORF1 sequences placed upstream of GFP. CRISPR–Cas9 of TASOR (TASOR KO) after reporter integration (bottom panel). **j**, Northern blot analysis of mRNAs produced from L1pb reporter in which ORF2 (4kb) sequence was replaced by 4 tandem repeats of ORF1 (1kb). RNA was isolated from the mix of WT and TASOR KO cells and iRFP probe was used to detect reporter mRNA (Fig. 1d).

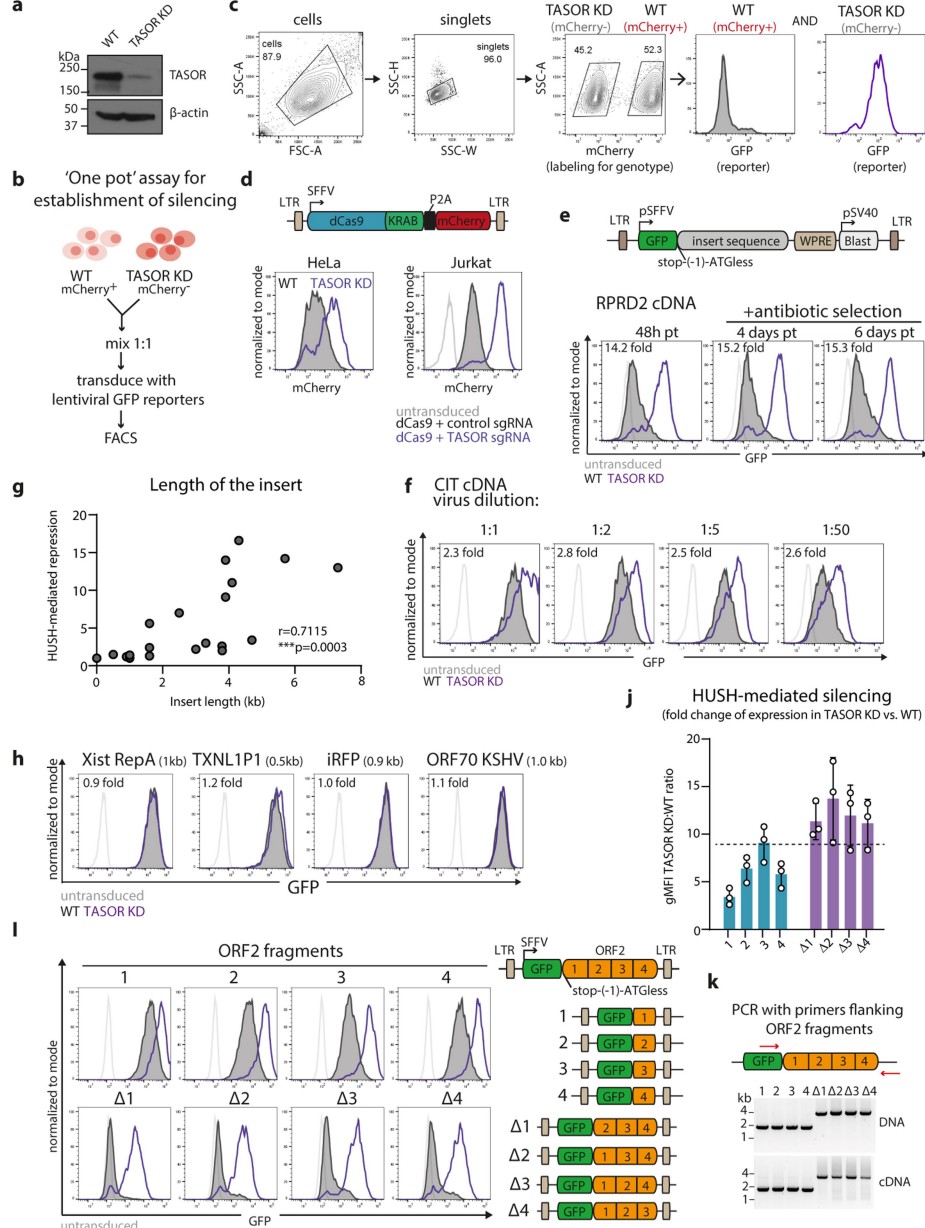

**Extended Data Fig. 2 | HUSH-mediated repression of cDNAs and ORF2 fragments correlates with length of the transgene. a**, Western blot showing TASOR depletion in TASOR KD HeLa cells. **b**, Schematic of assay for the establishment of silencing of lentiviral transgenes. **c**, Schematic of the gating strategy in 'one pot' assay for establishment of silencing. mCherry+ WT and mCherry- TASOR KO HeLa cells were defined based on the mCherry signal and the GFP signal for each of these subpopulations is subsequently plotted on the histogram. **d**, HUSH-mediated repression of the lentivirus encoding fusion of endonuclease dead Cas9 and KRAB domain (dCas9-KRAB) in HeLa (left) or Jurkat cells (right) measured by flow cytometry. mCherry fluorescence reports mRNA levels from the reporter. For Jurkat cells, a sgRNA targeting the TSS of *TASOR* was used to deplete TASOR. **e**, HUSH-mediated repression monitored at different time points post infection and after selection with the antibiotic for the transgene-delivered antibiotic resistance gene. (f) HUSH-mediated repression monitored 48h after transduction of HeLa WT and TASOR KD with

lentiviral reporter at different range of MOI. **e** and **f** were repeated with different reporters with similar results. **g**, Scatter plot illustrating a significant correlation between HUSH-mediated repression and length of the insert sequence in the GFP reporters. Each point represents a reporter with different cDNA sequence. Pearson correlation r = 0.7115, two-sided p = 0.0003; 95% CI [0.40 to 0.87] **h**, Expression of GFP non-coding lentiviral reporters bearing different short cDNA sequences in WT and TASOR KD HeLa cells measured by flow cytometry 72h post transduction. **i**, HUSH-mediated repression of GFP bearing the indicated untranslated ORF2 fragments measured by flow cytometry. **j**, Quantification of the HUSH-mediated repression of GFP untranslated reporters bearing full length ORF2 or ORF2 fragments, n = 3 biological replicates ±SD (left). **k**, RT-PCR analysis of transcripts from GFP reporters bearing ORF2 fragments with primers flanking ORF2 fragments (right). Product sizes corresponding to full length transcripts are 1.7 kb and 3.8 kb for reporters with 1-4 fragments and Δ1-4 fragments respectively.

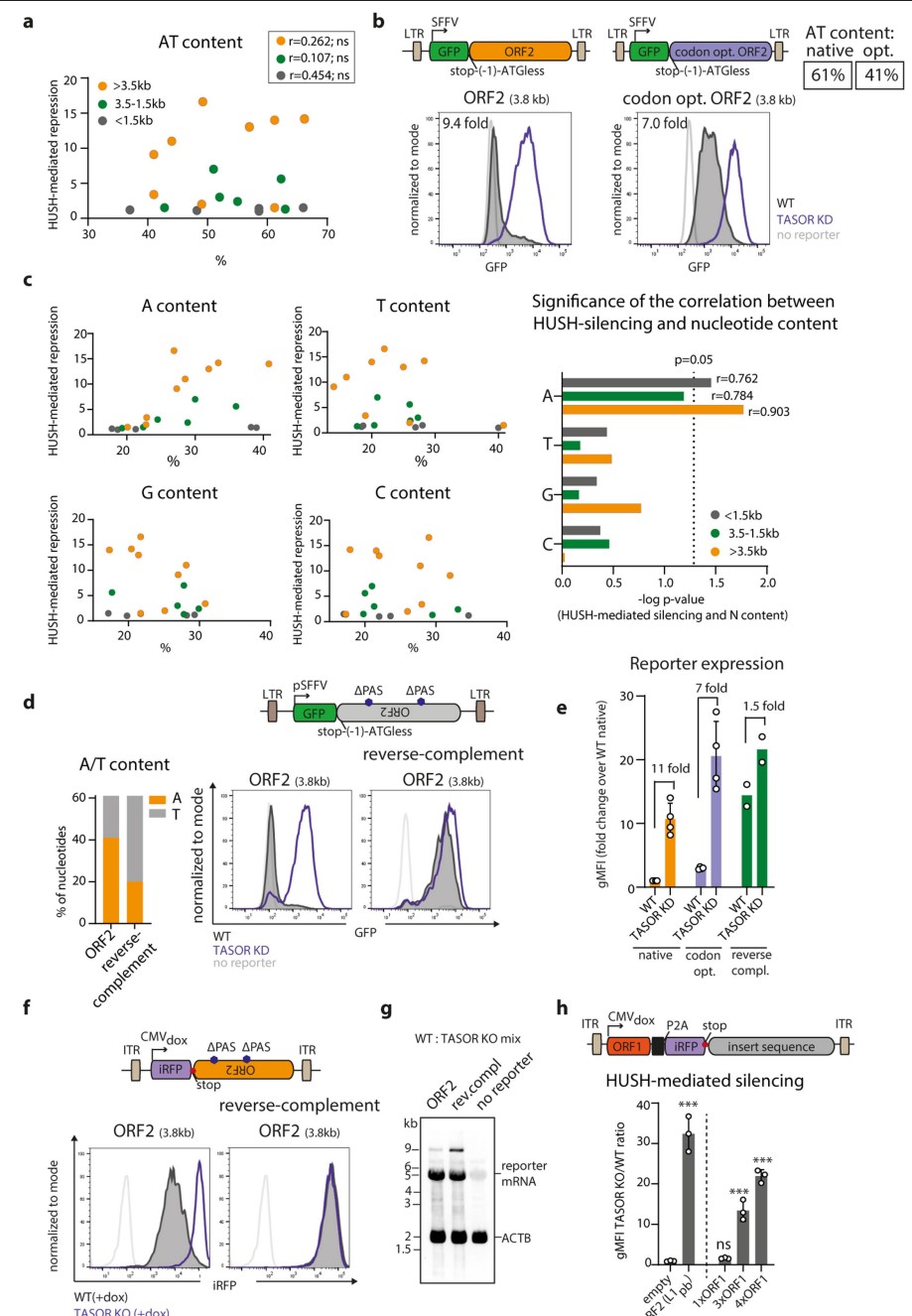

**Extended Data Fig. 3 | Susceptibility to HUSH-repression is governed by high adenine content in the sense strand and transgene length. a**, Scatter plot illustrating the relationship between HUSH-mediated repression and AT content of the insert sequence in the GFP reporter. Each point represents a reporter with different cDNA sequence. Reporters were assigned into three groups according to the length of the insert cDNA sequence (orange, green and grey) and Pearson r correlation was quantified for each group. **b**, Expression of GFP reporter bearing untranslated sequence of native or codon-optimized ORF2 (with increased GC content) in WT and TASOR KD HeLa cells measured by flow cytometry. Quantification of n = 3 independent experiments in 'e'. **c**, Scatter plots illustrating the relationship between HUSH-mediated repression and nucleotide content of the insert sequence in the GFP reporter (left) with the significance of the two-sided Pearson correlation between HUSH-mediated repression and nucleotide content (right). Dotted line on the graph corresponds to p-value = 0.05; for exact p-value see source data. **d**, HUSH-mediated repression of GFP lentiviral reporters bearing native ORF2 (A-rich) or reverse-complement ORF2 sequence (T-rich) measured 4 days post

transduction (right). To prevent premature transcription termination, two putative polyadenylation sites were deleted from ORF2 sequence (AATAAA at position 228-233 of reverse complement ORF2 and ATTAAA at 123-129). Relative contribution of A and T nucleotides to the nucleotide content of the insert (left). **e**, Quantification of the HUSH-mediated repression of GFP untranslated reporters bearing native, codon-optimized (codon opt.) or reverse complement (reverse compl.) ORF2; n = 4 (native and codon opt.) and n = 2 (for reverse compl.) biological replicates ±SD, normalized to gMFI of native ORF2 reporter in WT cells. **f**, The expression of transposase-integrated reporters bearing ORF2 or reverse complement ORF2 sequence in WT or TASOR KO HeLa cells measured by flow cytometry. **g**, Analysis of mRNA produced from reporters in **f**, by Northern blot. RNA was isolated from the mix of WT and TASOR KO cells. **h**, Quantification of HUSH-mediated repression of L1pb reporter in which ORF2 sequence was replaced by 1 ORF1, 3 ORF1 or 4 ORF1 tandem repeats. mean of n = 3 biological replicates ± SD ***p < 0.001 (for exact p value see source data); one-way ANOVA post-hoc pairwise comparisons with Bonferroni correction.

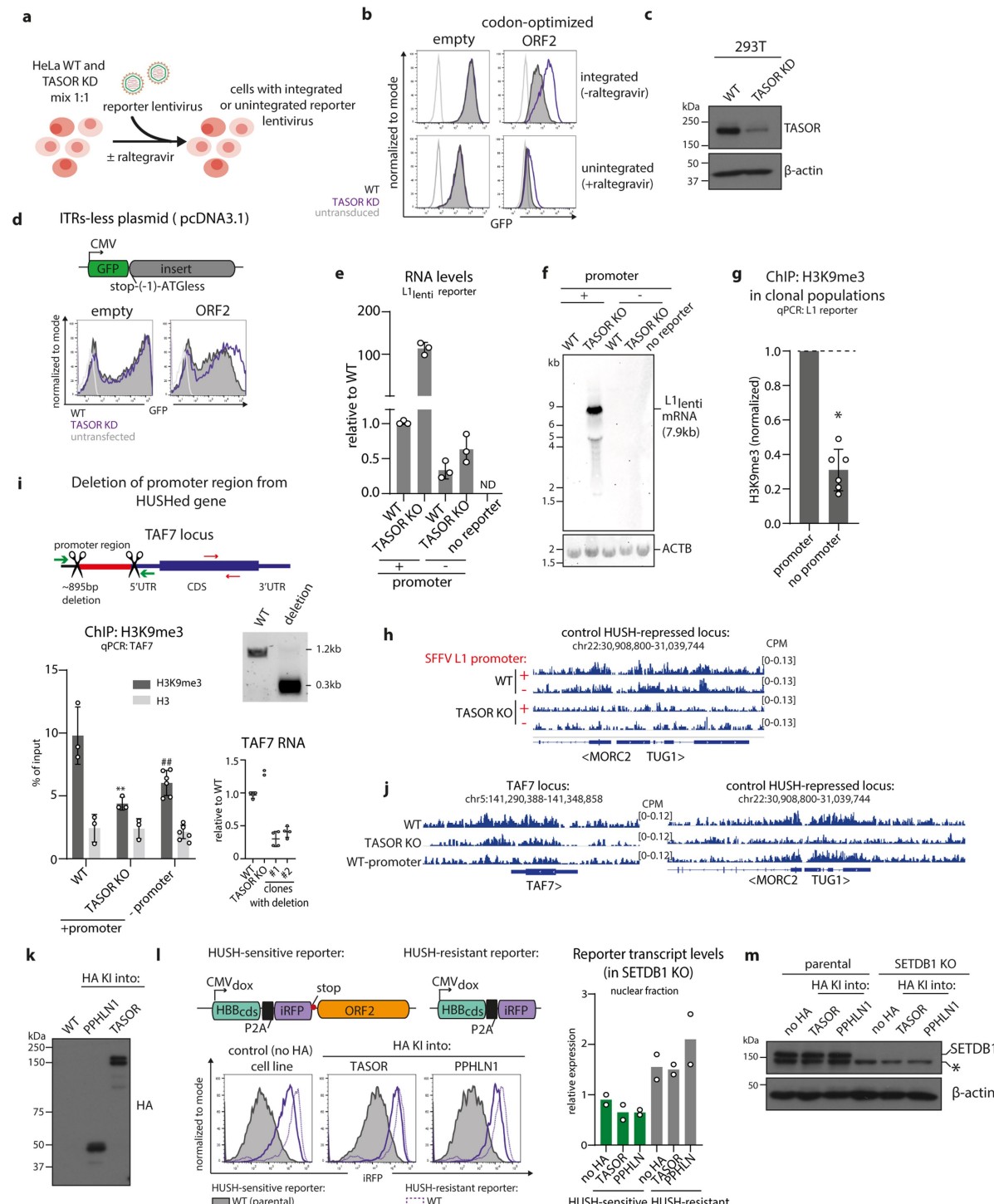

**Extended Data Fig. 4** | See next page for caption.

**Extended Data Fig. 4 | HUSH represses non-integrated DNAs and requires transcription to maintain repression. a**, Transduction of cells with lentiviral reporter in the presence or absence of the integrase inhibitor raltegravir to test the establishment of reporter silencing in the presence or absence of reporter integration. **b**, Representative flow cytometry histograms of expression from integrated or unintegrated lentiviral reporter in WT or TASOR KD HeLa cells. As unintegrated lentivirus is poorly expressed, the reporter with the synthetic ORF2 sequence was used since it provides higher expression than the native ORF2 sequence. **c**, Western blot showing CRISPR/Cas9 mediated depletion of TASOR in the population of 293T cells (TASOR KD). β-actin is a loading control. **d**, Flow cytometry histograms of expression from pcDNA3.1 plasmid transfected into WT or TASOR KD 293T cells. In contrast to lentivirus or plasmids for piggyBac-mediated integration, pcDNA3.1 lacks terminal repeats (ITRs). **e**, RT-qPCR quantifying transcript levels from SFFV-driven or promoter-less L1 lentiviral reporter integrated into WT and TASOR KO cells. Normalized to WT with SFFV-driven L1. n = 3 biological replicates (independent polyclonal integrations of the reporters) ±SD **f**, Northern blot analysis of mRNA produced from SFFV-driven or promoter-less L1 lentiviral reporter in WT and TASOR KO cells. **g**, ChIP-qPCR quantifying H3K9me3 at promoter-less L1 lentiviral reporter in clonal WT HeLa populations normalized to polyclonal WT population with SFFV-driven L1. n = 6 biological replicates ± SD, *p = 0.03 one-sample Wilcoxon test **h**, Genome browser track depicting H3K9me3 ChIPseq signal over control, HUSH-repressed locus in WT and TASOR KO HeLa cells harbouring SFFV-driven or promoter-less L1 reporter - related to Fig. 2b. **i**, CRISPR/Cas9-mediated deletion of the *TAF7* promoter region (schematic, upper left) reduces *TAF7* transcription measured by RT-qPCR and normalized to WT (bottom right). ChIP-qPCR quantifying H3K9me3 and total H3 at the locus (bottom left). n = 2 biological replicates x 3 independent experiments ± SD; **p = 0.0023, ##p = 0.009 one-way ANOVA post-hoc pairwise comparisons vs WT with Bonferroni correction. Cas9-cleavage sites indicated by scissors, green arrows indicate primers used to validate the deletion by genomic PCR and red arrows indicate position of the primers used in ChIP-qPCR. Gel image (upper right) confirms promoter deletion. **j**, Genome browser track depicting H3K9me3 ChIPseq signal over *TAF7* locus or control HUSH-repressed locus in WT, TASOR KO HeLa and HeLa with deletion of *TAF7* promoter. **k**, Western blot of HeLa cells with HA knocked into endogenous locus of *TASOR* or *PPHLN1*. **i**, Schematic of HUSH-sensitive and HUSH-resistant reporter constructs (upper schematic). Expression from the reporters is driven by dox-responsive promoter. Human beta globin coding sequence (*HBB*cds), instead of ORF1 as in the standard L1 reporter, is followed by P2A-iRFP and, for the HUSH sensitive reporter, by ORF2 sequence. HUSH-sensitive and HUSH-resistant reporters were integrated into control, HA-TASOR, PPHLN1-HA cell lines - resulting in six independent cell lines in total. In cell lines with the HUSH-sensitive reporter, SETDB1 function was then disrupted by CRISPR/Cas9-mediated knockout and mixed, polyclonal KO populations were used for RIP-qPCR. Flow cytometry histograms of expression from HUSH-sensitive or HUSH-resistant reporter in HA KI and control cell lines 48h after induction with dox (bottom). For the HUSH-sensitive reporter the expression is shown in WT and SETDB1 KO cells. Right panel: Relative levels of transcripts from reporters for RIP-qPCR (in SETDB1 KO) in nuclear fraction normalized to *ACTB*; n = 2 technical replicates **m**, Validation of SETDB1 depletion by CRISPR/Cas9 in TASOR or PPHLN1 HA-KI cells by western blot. β-actin as loading control. * marks non-specific band.

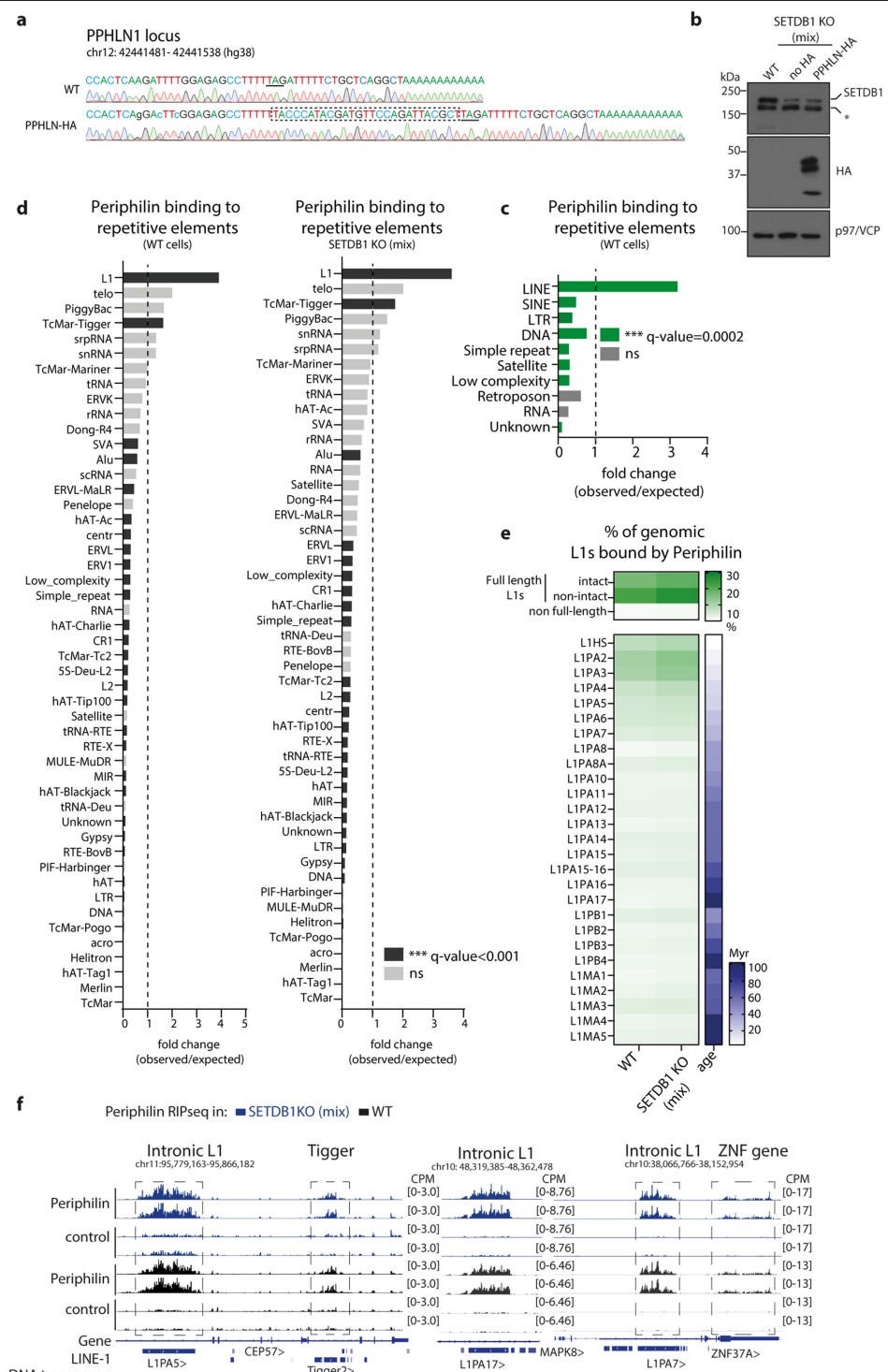

**Extended Data Fig. 5 | Periphilin specifically binds transcripts from evolutionary young, full-length L1 elements in WT and SETDB1-depleted cells. a**, Sequencing tracks showing insertion of sequence of HA-tag (marked as dashed box) into *PPHLN1* locus. Underlined is the stop codon. Nucleotide substitutions to make modified locus sgRNA-resistant are marked as small letters. **b**, Western blot showing SETDB1 depletion in SETDB1 KO (mix) Periphilin-HA HEK293Ts and control HEK293Ts. p97/VCP as a loading. **c,d**, Enrichment of periphilin RIP-seq peaks at different repetitive elements in **c**, WT cells and **d**, WT (left) and SETDB1 KO (mix) (right). Significant enrichment is defined as a fold change score greater than one with Benjamini–Hochberg empirical adjusted one-sided p-value calculated using simulations and genomic association testing[47], ***q < 0.001 (for exact p-values see source data). **e**, Fraction of full length, non-full length L1s and L1s from different families overlapping with periphilin RIPseq peaks. Full length L1s definitions are based on L1Base[48,49]. Blue heatmap indicates age of L1 families predicted from the phylogenetic analysis[50]. Periphilin-bound L1Hs may be underestimated in comparison to L1PA2-L1PA3 due to lower mappability of L1Hs as this is the least sequence-divergent L1 family. **f**, Genome browser tracks showing periphilin RIP signal over intronic L1s, Tigger DNA transposon and 3'UTR of *ZNF37A*.

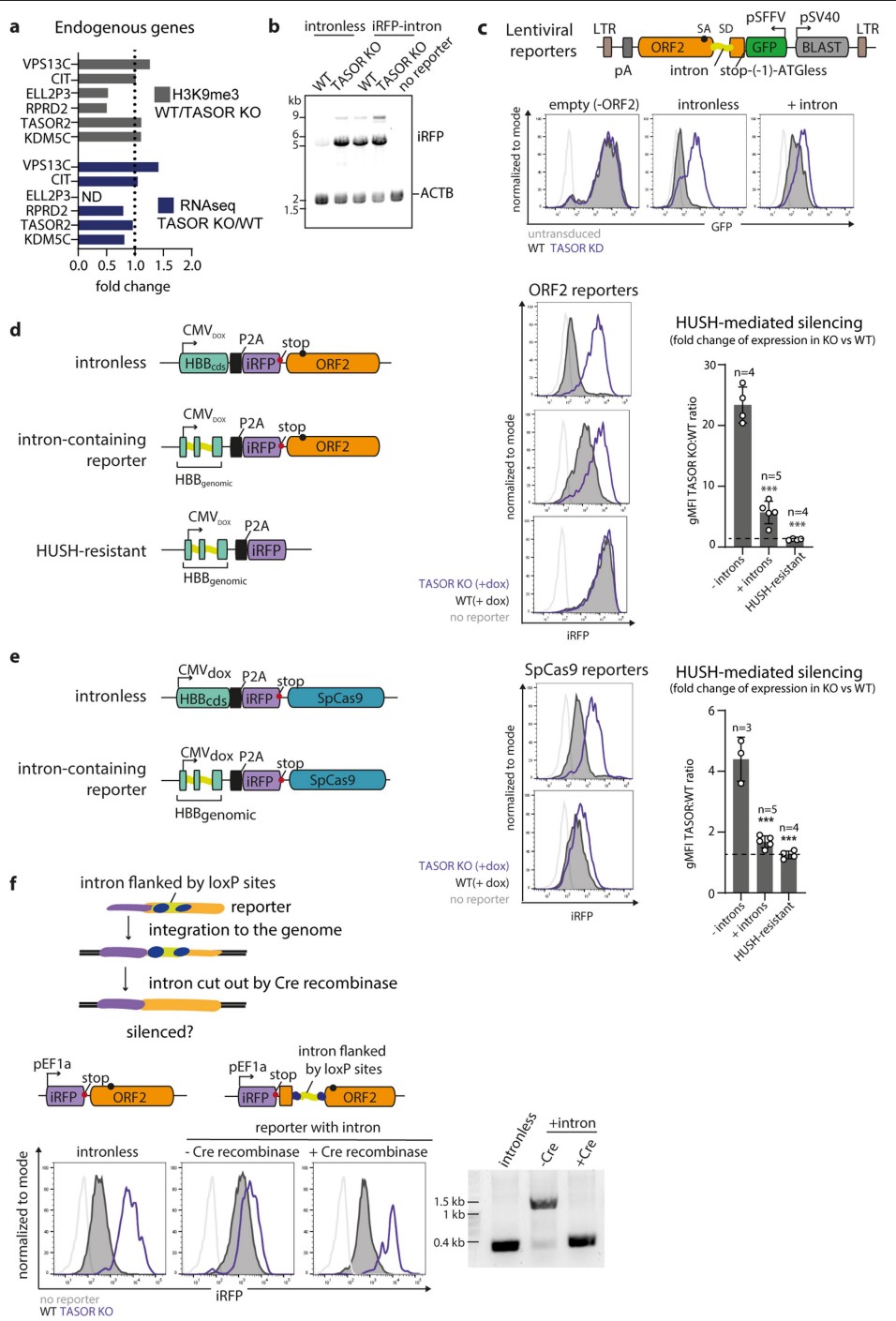

**Extended Data Fig. 6** | See next page for caption.

**Extended Data Fig. 6 | Introns protect different reporters from HUSH and are continuously required to prevent repression. a**, Quantification of H3K9me3 and RNAseq signal over endogenous genes in WT and TASOR KO K562 cells from a publicly available dataset[2]. None of these endogenous genes are HUSH-repressed, unlike lentiviral reporters containing cDNA sequences of these genes. **b**, Northern blot analysis of mRNAs produced from intronless reporter or reporter with *HBB* IVS2 cloned within the iRFP gene. ACTB is a loading control. **c**, Flow cytometry histograms showing expression from GFP and GFP-ORF2 intronless or intron-containing lentiviral reporters in WT and TASOR KD HeLa cells 72h post transduction (bottom). Schematic of the construct (top). To prevent intron splicing during transcription in the virus-producing cells, the reporter cassette driven by the SFFV promoter was cloned in reverse orientation with respect to lentiviral transcription. The polyadenylation signal (pA) in reverse orientation provides a signal for termination of transcription from the reporter cassette in transduced cells. ORF2 is untranslated and intron (*HBB* IVS2) is cloned 5' of ORF2. SA-splice acceptor, SD-splice donor. **d**, HUSH-mediated repression of integrated intronless or intron-containing ORF2 piggyBac reporters measured by flow cytometry (histograms in centre panel) and calculated as the ratio of reporter expression in TASOR KO and WT HeLa (right). Expression from the reporter is driven by a dox-responsive CMV promoter. Reporters contain either human beta globin coding sequence or genomic sequence (containing 2 introns) followed by P2A-iRFP and ORF2 sequences (schematics on the left). A HUSH-resistant reporter without ORF2 is the negative control. n biological replicates (independent polyclonal integrations of the reporters) ± SD; ***p ≤ 0.0001, one-way ANOVA post-hoc pairwise comparisons vs −introns with Bonferroni correction. **e**, HUSH-mediated repression of integrated intronless or intron-containing Cas9 piggyBac reporters measured by flow cytometry (histograms in centre panel) and calculated as the ratio of reporter expression in TASOR KO and WT HeLa (right). Expression from the reporter is driven by a dox-responsive CMV promoter. Reporters contain either *HBB* coding sequence or genomic sequence (containing 2 introns) followed by P2A-iRFP and Cas9 sequences (schematics on left panel). A HUSH-resistant reporter without Cas9 is the negative control. n biological replicates (independent polyclonal integrations of the reporters) ± SD; ***p ≤ 0.0001, one-way ANOVA post-hoc pairwise comparisons vs −introns with Bonferroni correction. **f**, HUSH-mediated repression of reporter with intron removed by Cre-loxP recombination following the reporter integration (upper schematic). Flow cytometry histograms of expression from iRFP-ORF2 reporters driven by EF1a promoter: (i) intronless or (ii) reporter-bearing intron (*HBB* IVS2) flanked by loxP sites in the absence or presence of Cre expression (left). Gel image (right) confirms intron deletion.

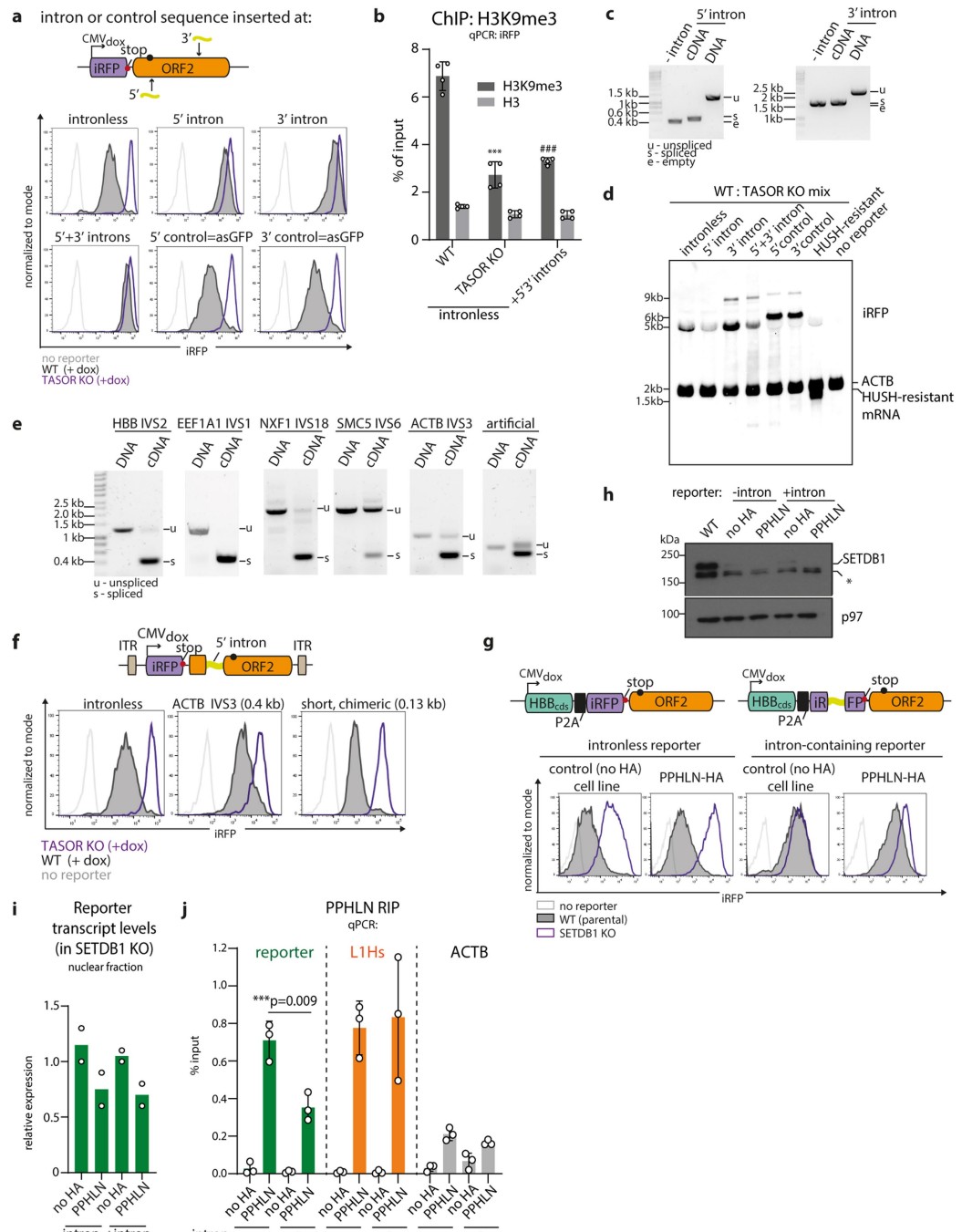

**Extended Data Fig. 7 | Intron insertion reduces HUSH-mediated repression and Periphilin binding to reporter transcripts. a**, Representative flow cytometry histograms of expression from reporters in Fig. 3c in WT and TASOR KO HeLa cells. The 5′ and 3′ control (asGFP) is the antisense GFP 'stuffer' sequence **b**, ChIP-qPCR quantifying H3K9me3 and total H3 levels at intronless or reporter with introns (*HBB* IVS2) inserted at 5′ and 3′ of ORF2 (from Fig. 3c and Extended Data Fig. 7a). n = 4 independent experiments ± SD; ***p = 0.0003 and ###p = 0.0006 vs −intron WT, ratio paired two-tailed t-test. **c**, Gel images confirming splicing of introns at 5′ and 3′ of ORF2 from iRFP-ORF2 reporter transcripts (from Fig. 3c and Extended Data Fig. 7a) by PCR. **d**, Northern blot analysis of mRNAs produced from reporters with intron or control sequence inserted 5′ and 3′ of ORF2 (~5kb), or HUSH-resistant reporter without ORF2 (~1.5kb). RNA was isolated from the mix of WT and TASOR KO cells. **e**, PCR analysis of splicing of different introns 5′ of ORF2 from iRFP-ORF2 reporter transcripts (from Fig. 3d and Extended Data Fig. 7f). **f**, Representative flow cytometry histograms of expression from iRFP-ORF2 reporter with introns from *ACTB* (0.4kb) or a short, chimeric intron (0.13kb) cloned 5′ of ORF2 in WT

and TASOR KO HeLa cells. Experiment repeated independently with similar results; quantification of n = 3-4 biological replicates in Fig. 3f. **g**, Schematic of intronless and intron-containing reporter constructs for periphilin RIP-qPCR (upper schematic). Reporters were integrated into WT 293T or periphilin-HA 293Ts - resulting in four independent cell lines. SETDB1 function was disrupted by CRISPR/Cas9-mediated knockout and mixed, polyclonal KO populations were used for RIP-qPCR. Flow cytometry histograms of expression from reporters in PPHLN-HA and control cell lines 48h after induction with dox (bottom). **h**, Validation of SETDB1 depletion by CRISPR/Cas9 in PPHLN1 HA-KI cells by western blot. β-actin as loading control. * marks non-specific band. **i**, Relative levels of transcripts from reporters for RIP-qPCR (in SETDB1 KO) in nuclear fraction normalized to *ACTB*; n = 2 technical replicates. **j**, RIP-qPCR showing decreased association of periphilin with RNA from intron-containing reporter. L1Hs and *ACTB* RNA are a positive and negative control, respectively. Data are mean ± SD; n = 3 independent experiments; and normalized to input.***p = 0.0009 vs -intron, one-way ANOVA post-hoc pairwise comparison with Bonferroni correction.

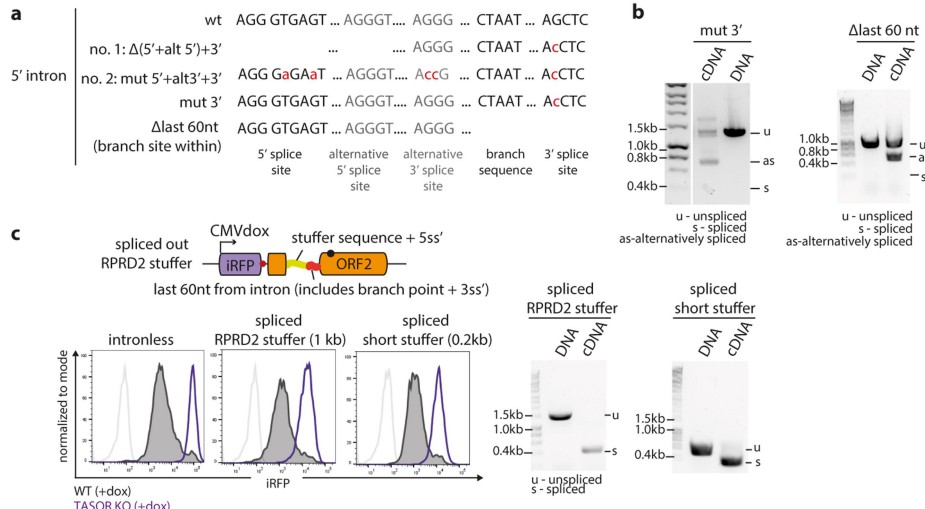

**Extended Data Fig. 8 | Sequences engineered for efficient splicing do not protect against HUSH repression. a**, Schematic of intron mutations in reporters from Fig. 3e. **b**, Analysis of splicing of mutant introns inserted 5′ of ORF2 from iRFP-ORF2 reporter transcripts by PCR (from Fig. 3e). **c**, Representative flow cytometry histograms of expression from reporters containing spliced stuffer sequences in WT and TASOR KO HeLa cells.

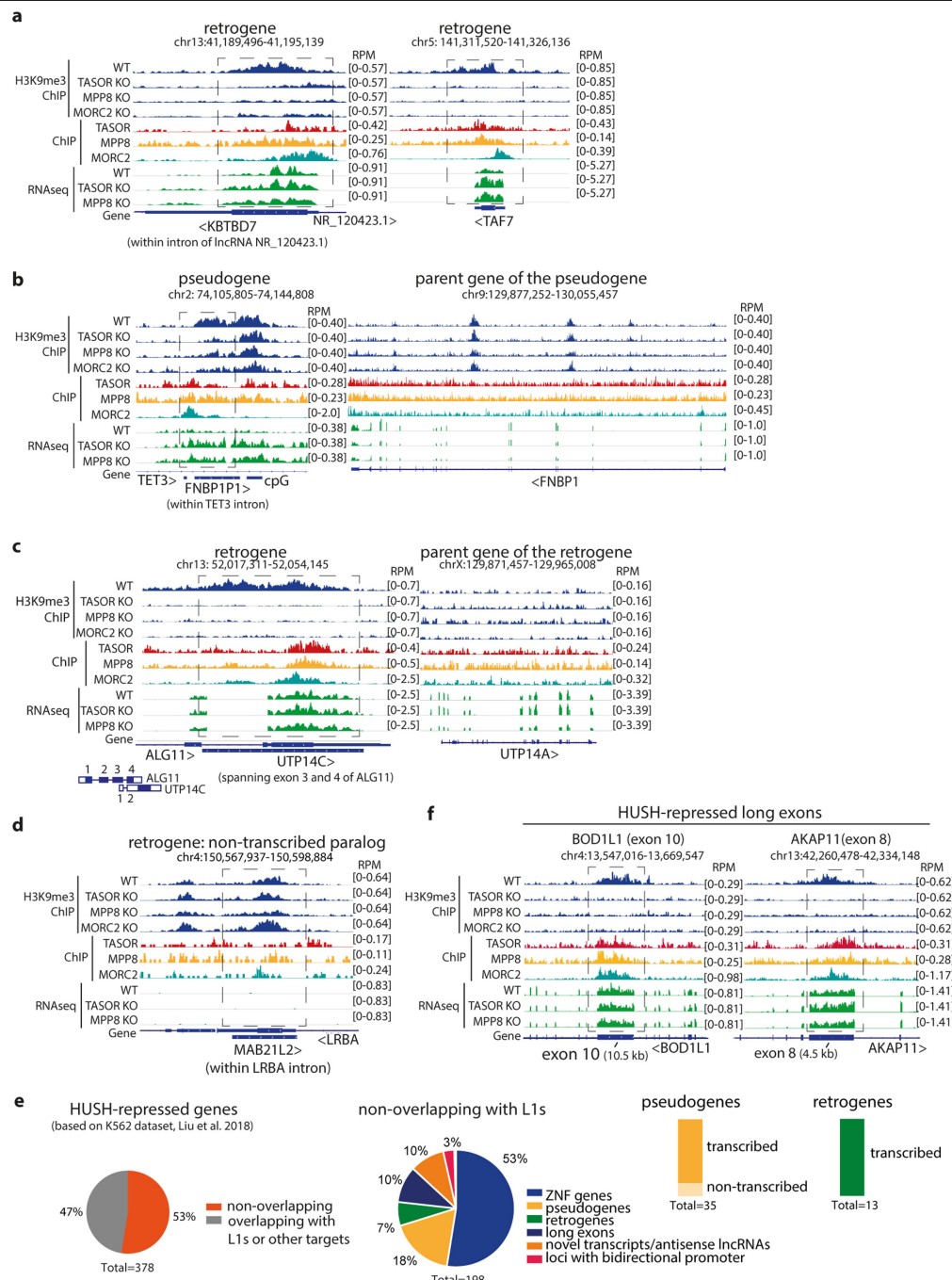

**Extended Data Fig. 9 | Transcribed processed pseudogenes and protein coding retrogenes, but not their parent genes, are bound and silenced by HUSH.** Genome browser tracks showing HUSH-dependent H3K9me3, HUSH/MORC2-occupancy and RNA-seq in WT and HUSH KO K562 cells at: **a**, additional, representative loci of retrogenes; **b**, *FNBP1P1* pseudogene (left) and its parent gene *FNBP1* (right); **c**, *UTP14C* retrogene (left) and its parent gene *UTP14A* (right) **d**, at the locus of *MAB21L2*, a non-transcribed paralog of HUSH-repressed *MAB21L1* retrogene, Data from[2]. **e**, HUSH-repressed genes obtained from the dataset in ref.[2]: 378 genes were obtained when there was at least 30% reduction of H3K9me3 signal in all 3 knockout cell lines: TASOR KO, MPP8 KO and MORC2 KO (log$_2$FC H3K9me3 TASOR KO/WT ≤ -0.5; FDR significance ≤ 0.05; determined after a comparative assessment of counts between conditions (n = 2) using negative binomial generalized linear models

as implemented in edgeR and corrected for multiple comparisons using FDR method). 104 of them were ZNF genes (including 8 ZNF pseudogenes) and the rest were inspected in IGV to determine the most probable reason for HUSH-repression e.g. overlap with L1 elements or other HUSH targets, pseudogene, retrogene, genes with signal over long exons, novel transcripts or antisense lncRNAs or loci with bidirectional promoter. The 11 resulting genes remained unannotated, either excluded because of low, background H3K9me3 over region or mapping artifacts or the reason for repression was unclear. Fraction of pseudogenes and retrogenes transcribed (average RNA-seq signal of all samples above > 0.1 RPKM or within transcriptionally active gene) (right) **f**, Genome browser tracks showing HUSH-dependent H3K9me3, HUSH/MORC2-occupancy and RNA-seq in WT and HUSH KO K562 cells at representative long exons. Data from ref.[2].

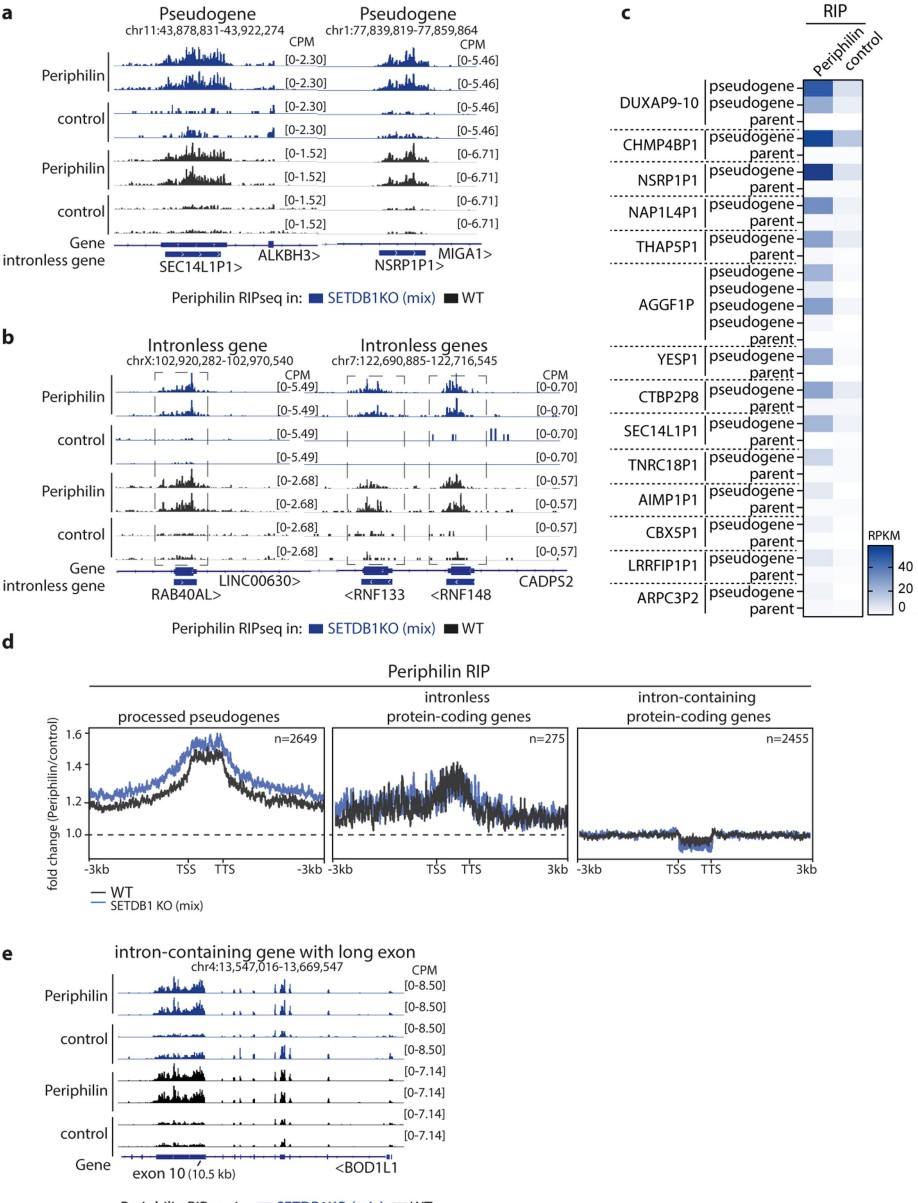

**Extended Data Fig. 10 | Periphilin specifically binds to transcripts from intronless genomic loci.** Genome browser tracks showing periphilin RIP signal over representative loci of processed pseudogenes **a**, and intronless genes **b**, in WT and SETDB1 KO (mix) HEK293T cells. **c**, Heatmap showing periphilin and control RIP signal (RPKM) over selected pseudogenes and their corresponding parent intron-containing genes. For *DUXA* and *AGGF1* parent genes, two (*DUXAP9*, *DUXAP10*) and four (*AGGF1P1*, *AGGF1P2*, *AGGF1P3*, *AGGF1P10*) pseudogenes are depicted. Data from periphilin RIPseq in SETDB1 KO (mix) cells (median of n = 4 independent experiments). **d**, Metagene profile of fold change of periphilin and control mean RIP-seq signal over three categories of genes: processed pseudogenes, intronless genes and intron-containing protein-coding genes. Only genes with periphilin RIPseq signal greater than 0.3 RPKM are considered (in each four RIP replicates in SETDB1 KO (mix) and two replicates in WT 293Ts). Genes where the periphilin signal enrichment peaks overlap with L1 elements are excluded. TSS-transcription start site, TTS-transcription termination site. Intronless protein-coding genes produce only intronless isoforms. **e**, Genome browser track showing periphilin RIPseq signal over representative locus of intron-containing gene (*BOD1L1*) with a long exon in WT and SETDB1 KO (mix) HEK293Ts.

# Reporting Summary

Nature Research wishes to improve the reproducibility of the work that we publish. This form provides structure for consistency and transparency in reporting. For further information on Nature Research policies, see our Editorial Policies and the Editorial Policy Checklist.

## Statistics

For all statistical analyses, confirm that the following items are present in the figure legend, table legend, main text, or Methods section.

| n/a | Confirmed | |
|---|---|---|
| ☐ | ☒ | The exact sample size (*n*) for each experimental group/condition, given as a discrete number and unit of measurement |
| ☐ | ☒ | A statement on whether measurements were taken from distinct samples or whether the same sample was measured repeatedly |
| ☐ | ☒ | The statistical test(s) used AND whether they are one- or two-sided<br>*Only common tests should be described solely by name; describe more complex techniques in the Methods section.* |
| ☒ | ☐ | A description of all covariates tested |
| ☐ | ☒ | A description of any assumptions or corrections, such as tests of normality and adjustment for multiple comparisons |
| ☐ | ☒ | A full description of the statistical parameters including central tendency (e.g. means) or other basic estimates (e.g. regression coefficient) AND variation (e.g. standard deviation) or associated estimates of uncertainty (e.g. confidence intervals) |
| ☐ | ☒ | For null hypothesis testing, the test statistic (e.g. $F$, $t$, $r$) with confidence intervals, effect sizes, degrees of freedom and $P$ value noted<br>*Give P values as exact values whenever suitable.* |
| ☒ | ☐ | For Bayesian analysis, information on the choice of priors and Markov chain Monte Carlo settings |
| ☒ | ☐ | For hierarchical and complex designs, identification of the appropriate level for tests and full reporting of outcomes |
| ☐ | ☒ | Estimates of effect sizes (e.g. Cohen's *d*, Pearson's *r*), indicating how they were calculated |

*Our web collection on statistics for biologists contains articles on many of the points above.*

## Software and code

Policy information about availability of computer code

| Data collection | Quant Studio Real-Time PCR S v1.7.1, BD FACS Diva, Image Lab 6.1, iBright™ Analysis Software |
|---|---|
| Data analysis | Bash (v4.2.46), R (v3.6.0), Python (v3.8.5), SRA Tools (v2.10.8), FastQC (v0.11.7), cutadapt (v1.16), UMI-tools (v1.1.1), HISAT2 (v2.1.0), SAMtools (v1.9), sambamba (v0.6.6), deepTools (v3.1.0), BEDTools (v2.30.0), HTSeq (v0.9.1), data.table (v1.13.2), GenomicFeatures (v1.38.2), edgeR (v3.28.1), GAT (v1.0), RepeatMasker (v UCSC hg38 last updated 2018-08-10), L1Base (downloaded 27th June 2021, http://l1base.charite.de/l1base.php)<br>Flowjo 10.3.0 for flow cytometry analyses,<br>IGV v2.7.0 for visualisation of ChIPseq and RNAseq data,<br>GraphPad Prism 8.4.3 for statistcs,<br>Quant Studio Real-Time PCR S v1.7.1 for qPCR analysis. |

For manuscripts utilizing custom algorithms or software that are central to the research but not yet described in published literature, software must be made available to editors and reviewers. We strongly encourage code deposition in a community repository (e.g. GitHub). See the Nature Research guidelines for submitting code & software for further information.

## Data

Policy information about availability of data

All manuscripts must include a data availability statement. This statement should provide the following information, where applicable:
- Accession codes, unique identifiers, or web links for publicly available datasets
- A list of figures that have associated raw data
- A description of any restrictions on data availability

All data supporting the findings of this study are available within the Article files. Gels and blots source images are provided in Supplementary Figure 1. In addition,

April 2020

the following figures have associated source data: Fig. 2d, Extended Data 2g, 3a, 3c, 3h, 5c, 5d, 5e, 9e, 10c. Next generation sequencing data have been deposited at the Gene Expression Omnibus with accession number: GSE181113. The accession number for the publicly available data from Liu et. al 2018 is GSE95374 (ChIP sequencing and RNA sequencing data).

# Field-specific reporting

Please select the one below that is the best fit for your research. If you are not sure, read the appropriate sections before making your selection.

☒ Life sciences ☐ Behavioural & social sciences ☐ Ecological, evolutionary & environmental sciences

For a reference copy of the document with all sections, see nature.com/documents/nr-reporting-summary-flat.pdf

# Life sciences study design

All studies must disclose on these points even when the disclosure is negative.

| | |
|---|---|
| Sample size | No sample-size calculations were performed. Rather, sample size was chosen following standard practice in the field and to balance statistical power and technical feasibility. Sample size and number of independent experiments are mentioned in Figures and Figure legends. |
| Data exclusions | No data was excluded |
| Replication | Experiments were reproduced as stated in the manuscript and appropriate positive and negative controls were used. The following figure panels show representative data from at least two independent experiments that showed similar results: Fig 3e, Extended Data Fig. 1b, 1e, 1i, 2a, 2g, 2k, 3a, 3c, 3d, 4d, 4l, 4m, 5b, 5f, 6b, 6f, 7g. The following figure panels show representative data from at least three independent biological replicates that showed similar results: Fig 1d, Fig2 2a (right), Fig 3a, Fig. 3d, Extended Data Fig. 1f, 2d, 2h, 4b, 4c, 4k, 6c, 7c, 7e, 7h, 8b, 8c. The following figure panels show representative data from at least four independent biological replicates that showed similar results: Fig 1a, Fig 1b, Fig 1e, Fig 2e, Extended Data 2d, 3b, 3f, 10a, 10b, 10e. The experiments in Extended Data Fig 1c and Fig 1d were performed once, but where internally controlled for both positive and negative results. The Northern blot experiments in Extended Data 1h, 1j, 3g, 4f, 7d were performed once, but were internally controlled for both positive and negative results. The ChIPseq experiments in Fig. 2b (upper panel) and Extended Data Fig. 4h, 4j were performed once, but the results were independently validated by two independent ChIP-qPCR experiments.

Most results were validated by different approaches and/or using alternative techniques as extensively reported in the manuscript. Once procedures were fully optimized, all attempts at replication were successful. |
| Randomization | There were no human or animal participants in this study. Random allocation did not apply because samples were not subjected to co- or multivariate analysis. |
| Blinding | The investigators were not blinded to sample allocation because samples were all analyzed using the same procedure and due to exclusive use of cell lines. Blinding was not necessary where data were generated by a digital reading or by quantitative measuremenet. |

# Reporting for specific materials, systems and methods

We require information from authors about some types of materials, experimental systems and methods used in many studies. Here, indicate whether each material, system or method listed is relevant to your study. If you are not sure if a list item applies to your research, read the appropriate section before selecting a response.

## Materials & experimental systems

| n/a | Involved in the study |
|---|---|
| ☐ | ☒ Antibodies |
| ☐ | ☒ Eukaryotic cell lines |
| ☒ | ☐ Palaeontology and archaeology |
| ☒ | ☐ Animals and other organisms |
| ☒ | ☐ Human research participants |
| ☒ | ☐ Clinical data |
| ☒ | ☐ Dual use research of concern |

## Methods

| n/a | Involved in the study |
|---|---|
| ☐ | ☒ ChIP-seq |
| ☐ | ☒ Flow cytometry |
| ☒ | ☐ MRI-based neuroimaging |

# Antibodies

| | |
|---|---|
| Antibodies used | Antibodies for immunoblotting:<br>rabbit α-TASOR (Atlas, HPA006735, 1:5000),<br>rabbit α-MPP8 (Proteintech, 16796-1-AP, 1:5000),<br>rabbit α-Periphilin1 (Sigma-Aldrich, HPA038902, 1:5000),<br>rabbit anti-MORC2 (Bethyl Laboratories, A300-149A, 1:5000),<br>rabbit α-SETDB1 (Proteintech, 11231-1-AP; 1:5000),<br>rat α-HA tag (3F10, Sigma-Aldrich, 11867423001, 1:10 000), |

mouse α-β-actin peroxidase conjugate (Sigma-Aldrich, A3854; 1:20 000),
mouse α-p97 (Abcam, ab11433, 1:5000),
rabbit α-α-tubulin (11H10, CST, #2125, 1:5000).
HRP-conjugated secondary antibodies for immunoblotting were obtained from Jackson ImmunoResearch
Peroxidase AffiniPure Goat Anti-Mouse IgG (H+L) (115-035-146, 1:10 000)
Peroxidase AffiniPure Goat Anti-Rabbit IgG (H+L) (111-035-144, 1:10 000)
Peroxidase AffiniPure Goat Anti-Rat IgG (H+L) (112-035-143, 1:10 000)
Antibody for intracellular staining for flow cytometry (only used for KI cell line pre-screening):
mouse α-HA tag Alexa Fluor® 647 conjugate (Cell Signaling, #3444; 1:50 - only used for PPHLN1-HA and HA-TASOR KI validation).
Antibodies for ChIP-qPCR:
rabbit α-H3K9me3 (Abcam, ab8898) 5ug/IP,
rabbit α-Histone H3 (Abcam, ab1791) 5 ug/IP
and rabbit α-RNA Pol II (Bethyl Laboratories, A304-405A, 7.5ug/IP)

Validation

All antibodies validated by vendor and/or used in previous literature.
Antibodies against HUSH complex subunits and MORC2 and SETDB1 validated with lysates from knockout cell lines (Extended Data Figure 1E).
rat α-HA tag (3F10, Sigma-Aldrich): validated using lysates from HA+ and HA- cell lines
mouse α-HA tag Alexa Fluor® 647 conjugate (Cell Signaling, #3444): validated using staining of HA+ and HA- cell lines
mouse α-β-actin peroxidase conjugate (Sigma-Aldrich, A3854): https://www.sigmaaldrich.com/GB/en/product/sigma/a3854
mouse α-p97 (Abcam, ab11433): https://www.citeab.com/antibodies/758977-ab11433-anti-vcp-antibody-5
rabbit α-α-tubulin (11H10, CST, #2125): https://www.cellsignal.com/products/primary-antibodies/a-tubulin-11h10-rabbit-mab/2125
rabbit α-H3K9me3 (Abcam, ab8898): https://www.abcam.com/histone-h3-tri-methyl-k9-antibody-chip-grade-ab8898.html
rabbit α-Histone H3 (Abcam, ab1791): https://www.abcam.com/histone-h3-antibody-nuclear-marker-and-chip-grade-ab1791.html
rabbit α-RNA Pol II (Bethyl Laboratories, A304-405A): https://www.bethyl.com/product/pdf/A304-405A.pdf

## Eukaryotic cell lines

Policy information about cell lines

Cell line source(s)

HeLa were obtained from ECACC and HEK293T and Jurkat cells from ATCC.

Authentication

All cells were obtained from commercial sources. Cell morphology was assessed for authentication.

Mycoplasma contamination

Cell cultures were routinely tested and found negative for mycoplasma infection
(MycoAlert, Lonza).

Commonly misidentified lines
(See ICLAC register)

None of the cell lines used in this study are in the database of commonly
misidentified cell lines.

## ChIP-seq

### Data deposition

☒ Confirm that both raw and final processed data have been deposited in a public database such as GEO.

☒ Confirm that you have deposited or provided access to graph files (e.g. BED files) for the called peaks.

Data access links
*May remain private before publication.*

Gene Expression Omnibus (GEO) with accession number GSE181113

Files in database submission

HA1_R1.fastq.gz
HA1_R2.fastq.gz
HA2_R1.fastq.gz
HA2_R2.fastq.gz
empty1_R1.fastq.gz
empty1_R2.fastq.gz
empty2_R1.fastq.gz
empty2_R2.fastq.gz
SKOHA1_R1.fastq.gz
SKOHA1_R2.fastq.gz
SKOHA2_R1.fastq.gz
SKOHA2_R2.fastq.gz
SKOempty1_R1.fastq.gz
SKOempty1_R2.fastq.gz
SKOempty2_R1.fastq.gz
SKOempty2_R2.fastq.gz
WTempty1_R1.fastq.gz
WTempty1_R2.fastq.gz
WTempty2_R1.fastq.gz
WTempty2_R2.fastq.gz
WTHA1_R1.fastq.gz
WTHA1_R2.fastq.gz

WTHA2_R1.fastq.gz
WTHA2_R2.fastq.gz
L1wtK9.SLX-19690.NEBNext31.HCT7YDRXY.s_1.r_1.fq.gz
L1wtK9.SLX-19690.NEBNext31.HCT7YDRXY.s_1.r_2.fq.gz
L1koK9.SLX-19690.NEBNext32.HCT7YDRXY.s_1.r_1.fq.gz
L1koK9.SLX-19690.NEBNext32.HCT7YDRXY.s_1.r_2.fq.gz
L1wtsffvK9.SLX-19690.NEBNext33.HCT7YDRXY.s_1.r_1.fq.gz
L1wtsffvK9.SLX-19690.NEBNext33.HCT7YDRXY.s_1.r_2.fq.gz
L1kosffvK9.SLX-19690.NEBNext34.HCT7YDRXY.s_1.r_1.fq.gz
L1kosffvK9.SLX-19690.NEBNext34.HCT7YDRXY.s_1.r_2.fq.gz
L1wtIN.SLX-19690.NEBNext36.HCT7YDRXY.s_1.r_1.fq.gz
L1wtIN.SLX-19690.NEBNext36.HCT7YDRXY.s_1.r_2.fq.gz
L1koIN.SLX-19690.NEBNext37.HCT7YDRXY.s_1.r_1.fq.gz
L1koIN.SLX-19690.NEBNext37.HCT7YDRXY.s_1.r_2.fq.gz
L1wtsffvIN.SLX-19690.NEBNext38.HCT7YDRXY.s_1.r_1.fq.gz
L1wtsffvIN.SLX-19690.NEBNext38.HCT7YDRXY.s_1.r_2.fq.gz
L1kosffvIN.SLX-19690.NEBNext39.HCT7YDRXY.s_1.r_1.fq.gz
L1kosffvIN.SLX-19690.NEBNext39.HCT7YDRXY.s_1.r_2.fq.gz
TAFwtIN.SLX-19690.NEBNext15.HCT7YDRXY.s_1.r_1.fq.gz
TAFwtIN.SLX-19690.NEBNext15.HCT7YDRXY.s_1.r_2.fq.gz
TAFkoIN.SLX-19690.NEBNext19.HCT7YDRXY.s_1.r_1.fq.gz
TAFkoIN.SLX-19690.NEBNext19.HCT7YDRXY.s_1.r_2.fq.gz
TAF5IN.SLX-19690.NEBNext20.HCT7YDRXY.s_1.r_1.fq.gz
TAF5IN.SLX-19690.NEBNext20.HCT7YDRXY.s_1.r_2.fq.gz
TAFwtK9.SLX-19690.NEBNext09.HCT7YDRXY.s_1.r_1.fq.gz
TAFwtK9.SLX-19690.NEBNext09.HCT7YDRXY.s_1.r_2.fq.gz
TAFkoK9.SLX-19690.NEBNext10.HCT7YDRXY.s_1.r_1.fq.gz
TAFkoK9.SLX-19690.NEBNext10.HCT7YDRXY.s_1.r_2.fq.gz
TAF5K9.SLX-19690.NEBNext11.HCT7YDRXY.s_1.r_1.fq.gz
TAF5K9.SLX-19690.NEBNext11.HCT7YDRXY.s_1.r_2.fq.gz
ripseq_genes.txt
HA.SKOHA.v2.bed
WTHA.v2.bed
HA.SKOHA.dedup.bw
empty.SKOempty.dedup.bw
WTempty.dedup.bw
WTHA.dedup.bw
L1wtK9.SLX-19690.NEBNext31.options1.bw
L1koK9.SLX-19690.NEBNext32.options1.bw
L1wtsffvK9.SLX-19690.NEBNext33.options1.bw
L1kosffvK9.SLX-19690.NEBNext34.options1.bw
L1wtIN.SLX-19690.NEBNext36.options1.bw
L1koIN.SLX-19690.NEBNext37.options1.bw
L1wtsffvIN.SLX-19690.NEBNext38.options1.bw
L1kosffvIN.SLX-19690.NEBNext39.options1.bw
TAFwtIN.SLX-19690.NEBNext15.options1.bw
TAFkoIN.SLX-19690.NEBNext19.options1.bw
TAF5IN.SLX-19690.NEBNext20.options1.bw
TAFwtK9.SLX-19690.NEBNext09.options1.bw
TAFkoK9.SLX-19690.NEBNext10.options1.bw
TAF5K9.SLX-19690.NEBNext11.options1.bw

| Genome browser session (e.g. UCSC) | NA |

## Methodology

| Replicates | 2 biological replicates per RIPseq experiment in WT cells, 4 biological replicates for RIPseq in SETDB1 KO cells ; 1 biological replicate per ChIPseq experiment |

| Sequencing depth | HA1_R1.fastq.gz, total:14786048, unique:10229159, 32bp, paired-end |
| | HA1_R2.fastq.gz, total:14786048, unique:10322919, 43bp, paired-end |
| | HA2_R1.fastq.gz, total:10722549, unique:7439348, 32bp, paired-end |
| | HA2_R2.fastq.gz, total:10722549, unique:7505001, 43bp, paired-end |
| | empty1_R1.fastq.gz, total:4729885, unique:1528746, 32bp, paired-end |
| | empty1_R2.fastq.gz, total:4729885, unique:1543107, 43bp, paired-end |
| | empty2_R1.fastq.gz, total:8282694, unique:2598283, 32bp, paired-end |
| | empty2_R2.fastq.gz, total:8282694, unique:2624044, 43bp, paired-end |
| | SKOHA1_R1.fastq.gz, total:15609117, unique:10990995, 32bp, paired-end |
| | SKOHA1_R2.fastq.gz, total:15609117, unique:11101686, 43bp, paired-end |
| | SKOHA2_R1.fastq.gz, total:9550120, unique:9540930, 32bp, paired-end |
| | SKOHA2_R2.fastq.gz, total:9550120, unique:9540930, 43bp, paired-end |
| | SKOempty1_R1.fastq.gz, total:7323898, unique:5175408, 32bp, paired-end |
| | SKOempty1_R2.fastq.gz, total:7323898, unique:5200168, 43bp, paired-end |
| | SKOempty2_R1.fastq.gz, total:7979937, unique:5541629, 32bp, paired-end |
| | SKOempty2_R2.fastq.gz, total:7979937, unique:5570615, 43bp, paired-end |

WTempty1_R1.fastq.gz, total:6866855, unique:4969526, 32bp, paired-end
WTempty1_R2.fastq.gz, total:6866855, unique:4998030, 43bp, paired-end
WTempty2_R1.fastq.gz, total:6616513, unique:4781251, 32bp, paired-end
WTempty2_R2.fastq.gz, total:6616513, unique:4810245, 43bp, paired-end
WTHA1_R1.fastq.gz, total:16267305, unique:11585128, 32bp, paired-end
WTHA1_R2.fastq.gz, total:16267305, unique:11718466, 43bp, paired-end
WTHA2_R1.fastq.gz, total:11772624, unique:8425527, 32bp, paired-end
WTHA2_R2.fastq.gz, total:11772624, unique:8525050, 43bp, paired-end
L1wtK9.SLX-19690.NEBNext31.HCT7YDRXY.s_1.r_1.fq.gz, total:38420810, unique:38404212, 50bp, paired-end
L1wtK9.SLX-19690.NEBNext31.HCT7YDRXY.s_1.r_2.fq.gz, total:38420810, unique:38404212, 50bp, paired-end
L1koK9.SLX-19690.NEBNext32.HCT7YDRXY.s_1.r_1.fq.gz, total:33340238, unique:33318987, 50bp, paired-end
L1koK9.SLX-19690.NEBNext32.HCT7YDRXY.s_1.r_2.fq.gz, total:33340238, unique:33318987, 50bp, paired-end
L1wtsffvK9.SLX-19690.NEBNext33.HCT7YDRXY.s_1.r_1.fq.gz, total:33998289, unique:33975659, 50bp, paired-end
L1wtsffvK9.SLX-19690.NEBNext33.HCT7YDRXY.s_1.r_2.fq.gz, total:33998289, unique:33975659, 50bp, paired-end
L1kosffvK9.SLX-19690.NEBNext34.HCT7YDRXY.s_1.r_1.fq.gz, total:25911961, unique:25893781, 50bp, paired-end
L1kosffvK9.SLX-19690.NEBNext34.HCT7YDRXY.s_1.r_2.fq.gz, total:25911961, unique:25893781, 50bp, paired-end
L1wtIN.SLX-19690.NEBNext36.HCT7YDRXY.s_1.r_1.fq.gz, total:31476780, unique:31461111, 50bp, paired-end
L1wtIN.SLX-19690.NEBNext36.HCT7YDRXY.s_1.r_2.fq.gz, total:31476780, unique:31461111, 50bp, paired-end
L1koIN.SLX-19690.NEBNext37.HCT7YDRXY.s_1.r_1.fq.gz, total:36089855, unique:36067616, 50bp, paired-end
L1koIN.SLX-19690.NEBNext37.HCT7YDRXY.s_1.r_2.fq.gz, total:36089855, unique:36067616, 50bp, paired-end
L1wtsffvIN.SLX-19690.NEBNext38.HCT7YDRXY.s_1.r_1.fq.gz, total:32113099, unique:32089220, 50bp, paired-end
L1wtsffvIN.SLX-19690.NEBNext38.HCT7YDRXY.s_1.r_2.fq.gz, total:32113099, unique:32089220, 50bp, paired-end
L1kosffvIN.SLX-19690.NEBNext39.HCT7YDRXY.s_1.r_1.fq.gz, total:31415661, unique:31397086, 50bp, paired-end
L1kosffvIN.SLX-19690.NEBNext39.HCT7YDRXY.s_1.r_2.fq.gz, total:31415661, unique:31397086, 50bp, paired-end
TAFwtIN.SLX-19690.NEBNext15.HCT7YDRXY.s_1.r_1.fq.gz, total:35643250, unique:35610088, 50bp, paired-end
TAFwtIN.SLX-19690.NEBNext15.HCT7YDRXY.s_1.r_2.fq.gz, total:35643250, unique:35610088, 50bp, paired-end
TAFkoIN.SLX-19690.NEBNext19.HCT7YDRXY.s_1.r_1.fq.gz, total:28226413, unique:28213047, 50bp, paired-end
TAFkoIN.SLX-19690.NEBNext19.HCT7YDRXY.s_1.r_2.fq.gz, total:28226413, unique:28213047, 50bp, paired-end
TAF5IN.SLX-19690.NEBNext20.HCT7YDRXY.s_1.r_1.fq.gz, total:39157181, unique:39140446, 50bp, paired-end
TAF5IN.SLX-19690.NEBNext20.HCT7YDRXY.s_1.r_2.fq.gz, total:39157181, unique:39140446, 50bp, paired-end
TAFwtK9.SLX-19690.NEBNext09.HCT7YDRXY.s_1.r_1.fq.gz, total:33411826, unique:33396420, 50bp, paired-end
TAFwtK9.SLX-19690.NEBNext09.HCT7YDRXY.s_1.r_2.fq.gz, total:33411826, unique:33396420, 50bp, paired-end
TAFkoK9.SLX-19690.NEBNext10.HCT7YDRXY.s_1.r_1.fq.gz, total:40739392, unique:40723988, 50bp, paired-end
TAFkoK9.SLX-19690.NEBNext10.HCT7YDRXY.s_1.r_2.fq.gz, total:40739392, unique:40723988, 50bp, paired-end
TAF5K9.SLX-19690.NEBNext11.HCT7YDRXY.s_1.r_1.fq.gz, total:40605513, unique:40587539, 50bp, paired-end
TAF5K9.SLX-19690.NEBNext11.HCT7YDRXY.s_1.r_2.fq.gz, total:40605513, unique:40587539, 50bp, paired-end

| | |
|---|---|
| Antibodies | ChIPseq: rabbit α-H3K9me3 (Abcam, ab8898);<br>RIPseq: Pierce™ anti-HA magnetic beads (Thermo Fisher, 88837): anti-HA monoclonal antibody (clone 2-2.2.14) |
| Peak calling parameters | Peaks were called using a customised approach. For details about the bioinformatics data analyses, check https://github.com/semacu/hush |
| Data quality | FastQC was used for sequencing QC. Signal enrichment was investigated with deepTools. For details about the bioinformatics data analyses, check https://github.com/semacu/hush |
| Software | ChIPseq:<br>Raw fastq files were quality checked with FastQC and trimmed with cutadapt to remove adapter sequences and low-quality base calls (quality score < 20). Depending on the experiment, the resulting reads were aligned using HISAT2 to either the human reference genome only (version GRCh38) or the human reference genome concatenated with the sequence of the reporter construct unique fragment (P2A-iRFP), duplicates were marked using sambamba and alignments formatted using SAMtools. BigWig files containing genomic signal were computed at single base resolution and normalized to Counts Per Million (CPM) using deepTools. For details about the bioinformatics data analyses, check https://github.com/semacu/hush<br><br>RIPseq:<br>Raw fastq files were quality checked with FastQC, unique molecular identifiers extracted using UMI-tools and resulting reads trimmed with cutadapt. Alignments to the human reference genome (version GRCh38) were performed with HISAT2, then formatted and deduplicated using SAMtools and UMI-tools respectively. Peaks were called using a customised approach involving BEDTools, deepTools, several Bash commands, data.table and edgeR, and peak overlaps later visualised using Intervene. Genomic repeats were obtained from RepeatMasker (https://www.repeatmasker.org/) and L1Base (http://l1base.charite.de/l1base.php), and associations with the RIPseq peaks were investigated using GAT and BEDTools. Tables integrating gene information, RIPseq signal and repeats were obtained using BEDTools, data.table, GenomicsFeatures and edgeR. Finally combined bigWig files containing genomic signal were prepared with SAMTools and computed at single base resolution and normalized to Counts Per Million (CPM) using deepTools. For details about the bioinformatics data analyses, check https://github.com/semacu/hush |

# Flow Cytometry

## Plots

Confirm that:

☒ The axis labels state the marker and fluorochrome used (e.g. CD4-FITC).

☒ The axis scales are clearly visible. Include numbers along axes only for bottom left plot of group (a 'group' is an analysis of identical markers).

☐ All plots are contour plots with outliers or pseudocolor plots.

☐ A numerical value for number of cells or percentage (with statistics) is provided.

## Methodology

| | |
|---|---|
| Sample preparation | Cells were trypsinized, resuspended in culture media and washed and resuspended in PBS and acquired on a BD LSR Fortessa or sorted on BD FACSAria Fusion. Live cells were analysed. No staining involved. |
| Instrument | BD LSR Fortessa; BD FACSAria Fusion (for sorting) |
| Software | BD Diva for collection and FlowJo 10.3.0 for analysis |
| Cell population abundance | For 'one pot establishment assay' WT cells were transduced with mCherry-encoding lentiviral vectors and resulting cell population of 85% mCherry+ cells was FACS purified to ~98% mCherry+ cells. |
| Gating strategy | Cells were gated for live/dead and doublet exclusion using FSC and SSC channels. For 'one pot establishment assay' cells were gated for presence of mCherry signal (reporting on the genotype) and GFP signal for each of these subpopulations subsequently plotted on the histogram. In the assays with iRFP reporters, cells were gated for presence of GFP signal (reporting on the genotype) and iRFP signal for each of these subpopulations plotted on the histogram. See Extended Data Fig.2C and Supplementary Figure 2 for more details. |

☒ Tick this box to confirm that a figure exemplifying the gating strategy is provided in the Supplementary Information.

