## [Peer Review File · Nature]

Manuscript Title: Genome surveillance by HUSH-mediated silencing of intronless mobile elements.

Redactions – unpublished data

Redaction due to referees' comments about authors' unpublished data

Reviewer Comments & Author Rebuttals

Reviewer Reports on the Initial Version:

Referee #1 (Remarks to the Author):

In this manuscript by M. Seczynska and P. Lehner, the authors perform an elegant set of experiments to examine the mechanisms by which the HUSH complex silences invading DNA. Using fluorescent reporter genes, flow cytometry, ChIP, and genetic manipulation of HUSH components in cultured cells, the authors determine that HUSH is capable of silencing a number of invading DNA elements, when carried on retroviral vectors or DNA transposon based integrating vectors. Furthermore, the authors confirm the finding (recently published by Goff and colleagues) that HUSH can also silence vectors prior to transgene integration. HUSH dependent silencing is largely sequence independent, although some constructs (LINE orf1 P alone, GFP vector alone, smaller transgenes) are not silenced. The authors show that HUSH mediated silencing (measured by H3K9me3 ChIP-qPCR) is reduced upon mutation of promoter sequences, which blocks transcription, and is strikingly lost if the same transgene has an intron (whether spliced or not), suggesting a mechanism by which HUSH may specifically distinguish integrated cDNAs (retroviral and LINE1-derived elements) from intron containing genes. The authors provide evidence supporting this model by showing HUSH mediated silencing of intronless pseudogenes. How HUSH specifically recognizes intron-less sequences is not explored, nor precisely how introns might disrupt such silencing. The authors provide some evidence that HUSH interacts with RNA, which could explain how transcription helps to recruit HUSH components.

This is really a beautiful paper, and will be of interest to a wide audience . I have a few concerns that should be addressed.

1. In figures 3C and D, The authors use CHIP-qPCR to show that H3K9me3 levels are reduced on transgenes or pseudogenes upon blocking transcription (by mutations of the promoter), but the authors do not examine any control regions. This experiment would be dramatically improved by using ChIP-seq to demonstrate both the relative levels of H3K9me3 at the pseudogene/transgene compared with the rest of the genome, as well as how much H3K9me3 is lost upon reduction of transcription. In other words, does promoter loss completely prevent H3K9me3, or just lead to a partial reduction?
2. The RIP experiment on its own is pretty preliminary. This point could be greatly strengthened by additional experiments looking more globally at RNA binding and determining specificity. For example, does HUSH (periphilin) bind only to intron-less sequences? Does it have specificity for ERVs, LINEs, pseudogenes, or is it much less specific? Is it sequence dependent? This is important because it will clarify whether RNA association is really providing any of the specificity for HUSH activity. IF the authors cannot demonstrate this conclusively, it should probably be removed from the model.
3. Only a few retrogenes or pseudogenes targeted by HUSH are used as representative examples.

The authors should continue their analysis genome wide with strong statistics. How many genes are targeted by HUSH (not overlapping with retrotransposons)? How many retrogenes or pseudogenes are targeted by HUSH? How many are transcribed or not transcribed.

Minor issues.

- Figures annotations in the text needs to be verified. For example, l. 193.
- Line 86 needs a citation.

Referee #2 (Remarks to the Author):

Seczynska and Lehner (MS ID#:2021-02-01916)

In this study, the authors described how the human silencing hub (HUSH) complex targets and initiates silencing of invading genetic elements. It has been reported that the HUSH complex silences long interspersed element-1 retrotransposons (LINE-1s) and exogenous retroviruses, but how this process is conducted (how this target specificity is created) was unknown. By using elegant experimental approaches, the authors showed that HUSH is able to target and transcriptionally repress a broad range of long, intronless transgenes. Intronic DNA sequence insertion into HUSH-repressed transgenes counteracts repression, even in the absence of intron splicing. Furthermore, they showed that HUSH binds transcripts from the target locus, prior to and independent of H3K9me3 deposition, and the target transcription is crucial for the HUSH-mediated silencing. Genomic data also showed that HUSH binds and represses a subset of endogenous intronless genes generated through retrotransposition of cellular mRNAs. Collectively, the authors proposed a novel model of the immediate protection or specific selection mechanism against newly invading retroelements in mammals.

The manuscript is well written. Their findings are quite interesting, especially the proposed HUSH targeting mechanism of transposons (non-self) vs self-genes based on the presence/absence of introns. However, the reviewer also feels that current data is not sufficient for accepting the authors entire conclusion or proposed idea. Therefore, the reviewer requests the authors to respond to following issues for improving this interesting manuscript.

Major comments,

1. target size vs/and DNA sequence specificity issue

In this manuscript, the issue of sequence specificity and size limitation in HUSH-mediated silencing of its target should be more clarified. Because this study addressed these two factors for HUSH targeting and silencing, but two issues were mostly investigated not separately. For example, L1's ORF1 (relatively shorter DNA element than other silenced cDNA elements) was not targeted for HUSH-mediated silencing, but it is not clear whether this is due to the DNA sequence specificity or the size limitation. Therefore, the reporter silencing assay +/- HUSH using the untargeted/unsilenced cDNA element such as ORF1 but multiplied may clarify more about this issue. If multiplied ORF1 (for instance 3xORF1) becomes the HUSH-targeted/silenced element, this data strongly suggests that specific DNA sequence information is most likely not important (is not essential) for HUSH function, but target size is more critical for being the target of HUSH-mediated silencing. Also, the authors should be careful to use the term "recognize" if there is no clear sequence specificity in HUSH-mediated target silencing.

2. RNA binding and intronless, how generally critical for HUSH-mediated silencing?

How much RNA binding of the HUSH complex (PPHLN1) correlates with HUSH-mediated transcriptional silencing? Especially, how much intron counteracts PPHLN1 RNA binding of target elements? HUSH RNA binding is one of most important findings of this report, but how this finding is functionally crucial for the HUSH function is not described much. To address this question, 1) the authors should perform same expt as shown in Fig. 3E using any HUSH-sensitive reporter construct +/- intron. Also, the authors should provide the level of each reporter transcript. If a

huge difference between HUSH-sensitive and -resistant reporter exists, results of RIP-ChIP should be carefully evaluated. In addition, 2) RNA-seq analysis of RIP-ChIP of HA-tagged PPHLN1 in SETDB1 KO 293T cells is also recommended. This analysis clarifies not only how much intronless gene transcripts are enriched but also any RNA binding specificities (binding motif or GC contents) or nature of target/bound RNA (length of target transcript or expression amount) of the HUSH complex (PPHLN1), thus proving a more general role or nature of intronless genetic elements for being silenced by the HUSH complex.

3. HUSH targets intronless genes?

Associated with comment 2, the reviewer recommends performing bioinformatics analysis of HUSH components ChIP-seq data to clarify how intronless genes or elements are enriched at genome wide level which further validates the authors' proposal. For this analysis, the authors should focus on the genes or elements which are not targeted by TRIM28 because the HUSH complex also could be recruited to the TRIM28 target loci by the KAP1/SETDB1/ATF7IP complex and SETDB1-mediated H3K9me3.

4. splicing defective intron

It is clear that all mutant introns used in this study are defective for splicing. Although the authors argue that intron-mediated protection from HUSH is independent of assembly of the core spliceosome at the transgene RNA based on the ref 23, the biochemical evidence for absence of spliceosome components on the mutant intron elements still needs to be shown. If not spliced, what is different between unspliced intron and exon? They are the same DNA sequences. In this manuscript, at least the authors show potential mechanistic (positive) evidence of intron-mediated protection from HUSH. Discussion of possible mechanism is not satisfying.

5. HUSH target retrogene/pseudogene

a) The authors stated that "HUSH-repressed retrogenes and pseudogenes are positioned within transcriptionally active genes, in a manner similar to HUSH-regulated L1 elements." line 220-221. How can these elements be targeted and silenced by HUSH within the regular gene context and under the intron containing transcripts? The authors should address this problem.

b) The authors stated that "HUSH binding and HUSH-mediated H3K9me3 are also observed over some transcribed retrogenes (e.g TAF7, MAB21L1, KBTBD7, MAP10, UTP14C) and in most cases, lead to downregulation of their expression (Fig. 5B, Extended data Fig. 8A-B)". Indeed, 2 (KBTBD7 and UTP14C) out of 5 are not silenced by HUSH, thus so far not most cases. Should revise the sentence to fit their entire findings.

Minor points,

6. "HUSH-mediated restriction of L1 retrotransposition was reported to depend on the native nucleotide sequence of the L1 open reading frames (ORF)." line 85-86. The authors should cite original paper.

7. typo. line 201: Fig.4E → Fig. 4F

Referee #3 (Remarks to the Author):

In the manuscript entitled "Genome surveillance..." Seczynska and Lehner describe how silencing foreign DNA sequences by HUSH, a fundamental unknown step within the Epigenetics field, can be rescued with the inclusion of introns.

This study builds from seminal studies by the Lehner lab, and the findings included in this study are ground-breaking are worth to be published in Nature.

While conclusions are sound and experiments are well controlled, there are a number of points that deserve further experimentation as described below:

Major points

1) To inspect whether HUSH might recognise L1 sequences during silencing, a new lentiviral reporter was developed (Fig1, S1A). This lentiviral-based reporter allows to follow changes in L1 RNA expression by FACS. While this is a clever strategy, there are numerous controls missing regarding the validation of such reporter.

-Why authors assume L1-ORF2p is not translated from the inserted L1 reporter? Human L1-ORF2p is translated by an unconventional termination/reinitiation mechanism, AND ALL THE SEQUENCES REQUIRED FOR ITS TRANSLATION ARE LOCATED WITHIN ORF2 SEQUENCES (see Alisch et al., Genes Dev 2006). ORF2p translation is highly unusual, and don't even require an AUG to generate enough ORF2p to support retrotransposition (of engineered L1 vectors). Thus, authors need to demonstrate that ORF2p is not translated from the inserted L1 reporter lentivirus. This is not trivial, as L1 is known to retrotranspose to high levels in HeLa cells, and it is likely that new insertions might be generated from the inserted L1 reporter, making impossible to distinguish whether iRFP is translated from the inserted L1 reporter or from genomic de novo L1 integrations inserted by retrotransposition. The same applies for the Transposase-based construct. Authors could use epitope-tagging of ORF1 and ORF2 sequences, as described (Doucet et al., 2010; Taylor et al., 2013). A straightforward way to test this could be the use of the L1pb transgene in the presence of RT inhibitors (AZT, 3TC, 4dT, etc).

-Similarly, can authors exclude that splicing of the inserted L1 reporter could artifactually generate iRFP? There are commercial antibodies to ORF1p and ORF2p, and these should be used to control that iRFP and ORF1p are translated at an equimolar rate, and to further explore ORF2p expression. -iRFP expression should be also explored by Northern-blot, not by RTqPCR, which will also would control for putative changes in splicing of the L1 reporter.

- While populations of transduced cells were used in Fig1, authors should also explore clonal HUSH KO lines, to really demonstrate that indeed the inserted L1 reporter is expressed from all (presumably) lentiviral insertions.

The above controls are critical, as KO HUSH cells are known to support elevated levels of L1 retrotransposition, and there is not a single piece of data to demonstrate that iRFP might not be produced from bona fide L1 integrations in HeLa.

2) While presented data seem to suggest that ORF2p sequences might be responsible for HUSH-mediated silencing, critical controls are missing.

-On one hand, lack of silencing detected in TASOR depleted (or KO) cells is very different depending if only ORF2 or ORF1 and ORF2 sequences are included in the L1pb construct. Cells transduced with the only-ORF2 construct (Fig 1D, left side) are not as homogenous as when ORF1 and ORF2 are present within the construct (Fig 1B). However, it is unclear why authors used different constructs and cell lines in these experiments, and clearly, there is not an easy way to compare among these experiments (the equivalent of comparing apple and oranges). Authors should use the same construct backbone and cell type to conduct these experiments. This is important, as at difference with constructs containing ORF1 and ORF2 sequences, ORF1-only and ORF2-only constructs cannot retrotranspose in cultured cells, which could explain why (even if apple and oranges were compared) the profile of reporter expression is different. Note that trans-complementation of L1s is not effective.

-More importantly, in 2004 Han et al demonstrated that poor L1 expression is due to inadequate transcriptional elongation by RNAPolII, and that the A/T richness of the L1 sequences is responsible for the poor elongation. In a back-to-back study, Han and Boeke further demonstrated that poor L1 expression could be alleviated by increasing the GC content of the L1 sequence (i.e., codon optimization), in a length-dependent manner. Authors should explore whether the AT richness of sequences inserted by viral infection and their length (Fig 2), rather than their sequence, are the main feature recognised by HUSH. Indeed, many viruses are known to be ATrich, in part to avoid restriction by ZAP.

-Related to above: to rule-out that other L1 sequences might be implicated in HUSH-silencing, authors should also explore the role of 5' and 3' UTR sequences, as both are fast evolving during evolution to presumably escape host control.

- Authors tried to identify regions in L1-ORF2p responsible for HUSH-silencing (Fig. 2D). While data suggest that there are no regions in ORF2 responsible for their silencing, no controls for splicing were included. These controls are critical, as all constructs could generate the same (or very similar) mRNA due to splicing with L1 sequences. Consistently, Belancio and colleagues identified a major splicing site in ORF2.

3) It is unclear why some inserted DNAs are silenced and others are not, but their length seem to be a major determinant in their silencing (and perhaps their nucleotide composition). This should be further explored. For example, would HUSH silence a construct containing 4 tandem copies of L1-ORF1 (the size of ORF2)?

4) While presented data seem to indicate that transgene integration is not strictly required for HUSH silencing, it seem to have a clear effect in reporter expression (Fig 3A). The same applies for transcription. However, if transcription is required for transgene repression, it is unclear why the promoterless L1 reporter used in Fig3C is enriched in H3K9me3. The same applies for TAF7 (Fig3D). As above, authors should explore RNA levels by Northern-blot, to exclude that splicing of L1 sequences might interfere with reporter expression.

5) Role of splicing. Even if L1 (and other cellular cDNAs) don't contain canonical introns, there are numerous reports documenting splicing of L1 sequences (from the Deininger and Moran labs). Indeed, it appears that splicing is an effective way to attenuate retrotransposition. Furthermore, the full-length L1 mRNA is the less prevalent RNA isoform detected in cells. As authors conclude, these data suggest that canonical introns, rather than splicing per se might be involved in avoiding HUSH-mediated repression. As stated above, Northern-blot should be conducted to confirm that the expected RNAs are generated upon transfecting constructs used in Fig 4A&C. In addition, authors could explore the use of self-splicing introns in constructs to further strength their data. Similarly, authors could also explore whether AT-AC introns can also prevent HUSH-silencing.

6) Related to above: while data seem to strongly suggest that the presence of introns, spliced or not, might be ultimately related with avoiding HUSH-silencing, their length also seem to be important. Following the same rationale as in point 2, could authors explore whether intron nucleotide composition is associated with the avoidance of HUSH-silencing?

7) Pseudogene datasets. While intriguing, only a handful of processed pseudogenes have been explored here. Authors should explore all processed pseudogenes annotated in the human genome, to gain more robust conclusions, and to further establish the main role of transcription in HUSH-mediated silencing. This is critical, as the examples shown here could be the outliers, as there thousands of processed pseudogenes in the human genome.

8) A conclusion of the current study seems to suggest that introns, and not splicing, can avoid HUSH-mediated repression of L1 sequences. While intriguing, an open question is how HUSH can distinguish between intronless and intron-containing sequences. Or how HUSH can selectively bind young L1s and not older L1s. Several studies from the Ule lab have demonstrated that several RNA-binding-proteins (RBP) bind young L1s and insulate LINE sequences from RNA processing (i.e., splicing). As RBPs such as MATR3 and PTB1P bind young L1s, could these proteins be involved in the HUSH-mediating silencing described here?

9) Could authors also explore the behaviour of intron-less genes (Sox and others)? According to their model, some could be targeted by HUSH. Indeed, finding intron-less genes regulated by HUSH might allow to further solidify the proposed model, by inserting functional and mutated introns in these genes using CRISPR/Cas9.

Minor points:

a) There are some retroelements that have introns within their genomes, such as Penelope retrotransposons (Arkipoova, Systematic Biology, Volume 55, Issue 6, December 2006, Pages 875–885). While how introns are preserved during retrotransposition remains to be uncovered, strong statements should be corrected in the manuscript.

b) In results, add a reference to the following statement: "HUSH-mediated restriction of L1 retrotransposition was reported to depend on the native nucleotide sequence of the L1 open reading frames (ORF)"

c) page 7, lane 194: authors refer to Fig4E, but it should refer to 4F.

Author Rebuttals to Initial Comments:

Referee #1 (Remarks to the Author):

In this manuscript by M. Seczynska and P. Lehner, the authors perform an elegant set of experiments to examine the mechanisms by which the HUSH complex silences invading DNA. Using fluorescent reporter genes, flow cytometry, ChIP, and genetic manipulation of HUSH components in cultured cells, the authors determine that HUSH is capable of silencing a number of invading DNA elements, when carried on retroviral vectors or DNA transposon based integrating vectors. Furthermore, the authors confirm the finding (recently published by Goff and colleagues) that HUSH can also silence vectors prior to transgene integration. HUSH dependent silencing is largely sequence independent, although some constructs (LINE orf1 P alone, GFP vector alone, smaller transgenes) are not silenced. The authors show that HUSH mediated silencing (measured by H3K9me3 ChIP-qPCR) is reduced upon mutation of promoter sequences, which blocks transcription, and is strikingly lost if the same transgene has an intron (whether spliced or not), suggesting a mechanism by which HUSH may specifically distinguish integrated cDNAs (retroviral and LINE1-derived elements) from intron containing genes. The authors provide evidence supporting this model by showing HUSH mediated silencing of intronless pseudogenes. How HUSH specifically recognizes intron-less sequences is not explored, nor precisely how introns might disrupt such silencing. The authors provide some evidence that HUSH interacts with RNA, which could explain how transcription helps to recruit HUSH components.

This is really a beautiful paper, and will be of interest to a wide audience. I have a few concerns that should be addressed.

We thank the reviewer for their helpful and complimentary comments and for concisely summarizing our findings.

1. In figures 3C and D, The authors use CHIP-qPCR to show that H3K9me3 levels are reduced on transgenes or pseudogenes upon blocking transcription (by mutations of the promoter), but the authors do not examine any control regions. This experiment would be dramatically improved by using ChIP-seq to demonstrate both the relative levels of H3K9me3 at the pseudogene/transgene compared with the rest of the genome, as well as how much H3K9me3 is lost upon reduction of transcription. In other words, does promoter loss completely prevent H3K9me3, or just lead to a partial reduction?

Thank you for this helpful suggestion and we agree with your comments. In response to the suggestions, we now provide ChIPseq tracks over:

- (i) The uniquely mappable fragment of the promoter-less L1 transgene, as well as a control locus (**Fig. 2B, Extended Data 4H**)
- (ii) TAF7 locus with promoter deletion alongside with the control locus (**Extended Data Fig. 4J**)

This experiment clarifies that in the absence of the promoter there is no H3K9me3 deposition over the target gene, and the H3K9me3 ChIPseq signal dropping to the background level, as seen in TASOR KO.

2. The RIP experiment on its own is pretty preliminary. This point could be greatly strengthened by additional experiments looking more globally at RNA binding and determining specificity. For example, does HUSH (periphilin) bind only to intron-less sequences? Does it have specificity for ERVs, LINEs, pseudogenes, or is it much less specific? Is it sequence dependent? This is important because it will clarify whether RNA association is really providing any of the specificity for HUSH activity. IF the authors cannot demonstrate this conclusively, it should probably be removed from the model.

We agree this is a critical issue and were aware of the importance of clarifying the specificity of Periphilin binding. We now present additional data showing (i) specific binding of Periphilin to RNA and (ii) analysis of the Periphilin-RNA interactome.

We have performed UV-crosslinked Periphilin RIPseq in 293T cells (both WT and SETDB1 KO) and analysed the Periphilin-RNA interactome in the context of repeated elements and different gene classes.

These analyses reveal:

1. Periphilin binds to L1 transcripts with high specificity. Only L1s (and to a lesser extent TcMar-Tigger), showed significant enrichment for Periphilin signal (**Fig. 2D-E, Extended Data Fig. 5C-D & F**)
2. Periphilin binding to L1 transcripts reflects selective, genome-wide HUSH-mediated H3K9me3 deposition over L1s, as seen by preferential Periphilin binding to full length, evolutionary young L1 elements (**Extended Data Fig. 5E**)
3. Transcripts from intronless genes and processed pseudogenes are enriched for Periphilin binding (**Extended Data Fig 10D**). 20% transcribed pseudogenes and 17% intronless genes show at least 2-fold enrichment of Periphilin signal (with no overlap to L1s) (**Fig. 4B**).
4. Periphilin binding to intronless genes and pseudogenes is specific as no enrichment of signal is seen for (i) intron-containing pseudogene parent genes as well as other (ii) intron-containing genes (**Extended Data Fig. 10C-D**)
5. Transcripts from a small number of intron-containing genes are Periphilin-bound and these are predominantly (i) ZNF genes (known HUSH-targets) and interestingly (ii)

genes containing HUSH-repressed, unusually long exons >2kb (**Fig. 4B, right, Extended data Fig. 5F, right, Extended Data Fig. 10E, Extended Data Fig. 9F**)

These data conclusively demonstrate that Periphilin-RNA binding is specific.

We would like to stress that our experiments were performed in both the presence and absence of SETDB1, and in each case, the findings are the same. Our finding that Periphilin also binds RNA in the absence of SETDB1, and therefore H3K9me3, provides important mechanistic insight into HUSH function: (i) such conditions mimic what happens prior to H3K9me3 deposition over the HUSH-repressed locus, allowing us to infer that Periphilin-binding to RNA specifies target loci for repression. Whether this is the sole source of specificity in target recognition is unclear, and does not exclude a critical upstream event which triggers Periphilin-RNA association (e.g. related to transcriptional kinetics or structural flexibility of nascent transcript). (ii) Since MPP8 is absent from the HUSH-repressed locus in the absence of SETDB1 (Müller et al., 2021), the Periphilin-RNA binding is likely to specifically anchor HUSH at the target locus, independent of the MPP8 chromodomain-H3K9me3 interaction. We highlight both these critical points in our model, which is now strongly supported by additional data.

3. Only a few retrogenes or pseudogenes targeted by HUSH are used as representative examples. The authors should continue their analysis genome wide with strong statistics. How many genes are targeted by HUSH (not overlapping with retrotransposons)? How many retrogenes or pseudogenes are targeted by HUSH? How many are transcribed or not transcribed.

We have addressed the reviewer's questions by performing the following two analyses:

- (i) Analysis of genes with HUSH-mediated H3K9me3 (non-overlapping with L1s) using publicly available data (Liu et al., 2018) (**Extended Data Fig. 9E**)

We found 198 genes targeted by HUSH (as defined by >30% loss in signal in all HUSH core component knockout lines: TASOR KO, MPP8 KO and MORC2 KO): 104 were ZNFs, 35 were pseudogenes, 20 were protein-coding genes with H3K9me3 over long exons, 19 were lncRNAs/novel transcripts, 13 were retrogenes and 7 others.

All HUSH-repressed retrogenes were transcribed and 29 out of 35 pseudogenes were transcribed (either 'non-background' RNAseq or located within transcriptionally active genes).

In addition, we took advantage of our newly-generated Periphilin RIPseq dataset and analysed:

- (ii) Periphilin binding to transcripts from different gene classes: processed pseudogenes, intronless and intron-containing protein-coding genes (including only those genes which show no overlap with Periphilin peaks at the L1 elements) (**Fig. 4B, Extended Data Fig. 10D**)

We found 685 genes with at least 2-fold enrichment of Periphilin signal including:

- 519 pseudogenes (20% of all analysed)
- 48 intronless protein-coding genes (17%)
- 118 intron-containing protein-coding (5%), including 43 with long exons, 16 ZNFs, 6 whose only intronless isoform is Periphilin-bound, 2 that overlap pseudogenes and 51 others.

These analyses establish intronless genes as specific HUSH-targets.

Minor issues.

- Figures annotations in the text needs to be verified. For example, l. 193.
- Line 86 needs a citation.

We thank the reviewer for pointing out these oversights. The citation was added and new figure annotations have been verified.

Referee #2 (Remarks to the Author):

Seczynska and Lehner (MS ID#:2021-02-01916)

In this study, the authors described how the human silencing hub (HUSH) complex targets and initiates silencing of invading genetic elements. It has been reported that the HUSH complex silences long interspersed element-1 retrotransposons (LINE-1s) and exogenous retroviruses, but how this process is conducted (how this target specificity is created) was unknown. By using elegant experimental approaches, the authors showed that HUSH is able to target and transcriptionally repress a broad range of long, intronless transgenes. Intronic DNA sequence insertion into HUSH-repressed transgenes counteracts repression, even in the absence of intron splicing. Furthermore, they showed that HUSH binds transcripts from the target locus, prior to and independent of H3K9me3 deposition, and the target transcription is crucial for the HUSH-mediated silencing. Genomic data also showed that HUSH binds and represses a subset of endogenous intronless genes generated through retrotransposition of cellular mRNAs. Collectively, the authors proposed a novel model of the immediate protection or specific selection mechanism against newly invading retroelements in mammals. The manuscript is well written. Their findings are quite interesting, especially the proposed HUSH targeting mechanism of transposons (non-self) vs self-genes based on the presence/absence of introns. However, the reviewer also feels that current data is not sufficient for accepting the authors' entire conclusion or proposed idea. Therefore, the reviewer requests the authors to respond to following issues for improving this interesting manuscript.

We thank the reviewer for their helpful and inciteful comments and very much appreciate their positive feedback. Our responses are detailed below:

Major comments,

1. target size vs/and DNA sequence specificity issue

In this manuscript, the issue of sequence specificity and size limitation in HUSH-mediated silencing of its target should be more clarified. Because this study addressed these two factors for HUSH targeting and silencing, but two issues were mostly investigated not separately. For example, L1's ORF1 (relatively shorter DNA element than other silenced cDNA elements) was not targeted for HUSH-mediated silencing, but it is not clear whether this is due to the DNA sequence specificity or the size limitation. Therefore, the reporter silencing assay +/- HUSH using the untargeted/unsilenced cDNA element such as ORF1 but multiplied may clarify more about this issue. If multiplied ORF1 (for instance 3xORF1) becomes the HUSH-targeted/silenced element, this data strongly suggests that specific DNA sequence information is most likely not important (is not essential) for HUSH function, but target size is more critical for being the target of HUSH-mediated silencing. Also, the authors should be careful to use the term "recognize" if there is no clear sequence specificity in HUSH-mediated target silencing.

We thank the reviewer for their suggestion and have tested HUSH-repression of reporters in which the ORF2 sequence is replaced by an increasing number of tandem repeats of the ORF1 sequence. Interestingly, tandem repeats of ORF1 gradually became HUSH-repressed as their size increases (**Extended Data Fig. 3H, Fig. 1D**). On its own, this result could indicate that target length is the sole determinant of HUSH-susceptibility. However, our additional data suggests this is not the case. We show that reporters with the reverse-complement sequence of ORF2 (length-matched to ORF2) are completely HUSH-insensitive, despite producing full-length transcript as confirmed by Northern blot (**Extended Data Fig. 3D-G**). We attribute this effect to differences in the Adenine content (A-content), which is unusually high in the sense strand of ORF2 (41%) and only 20% in the sense strand of reverse-complement ORF2. Consistent with this assertion, we identify a strong correlation between the A-content in the sense strand of the different insert sequences (but not the overall AT content) and HUSH-repression (**Extended Data Fig. 3A&C**). Again it is important to note that this A-rich bias does not, by itself, confer HUSH-sensitivity, as illustrated by the HUSH-insensitive 1kb-ORF1, which has similar A-rich bias in the sense strand as ORF2 (~40%). We conclude that the HUSH-repression of reporters with multiple ORF1 repeats, and our findings on the role of nucleotide composition strongly suggest that HUSH susceptibility is governed by both A-rich bias in the sense strand and target length.

Having identified these determinants of HUSH-specificity and specific binding of Periphilin to target RNA (see later), we now feel more confident to use term 'recognize' in regard to HUSH and its targets.

2. RNA binding and intronless, how generally critical for HUSH-mediated silencing?

How much RNA binding of the HUSH complex (PPHLN1) correlates with HUSH-mediated transcriptional silencing? Especially, how much intron counteracts PPHLN1 RNA binding of target elements? HUSH RNA binding is one of most important findings of this report, but how this finding is functionally crucial for the HUSH function is not described much. To address this question, 1) the authors should perform same expt as shown in Fig. 3E using any HUSH-sensitive reporter construct +/- intron. Also, the authors should provide the level of each reporter transcript. If a huge difference between HUSH-sensitive and -resistant reporter exists, results of RIP-ChIP should be carefully evaluated.

As suggested by the reviewer, we have performed RIP-qPCR, comparing intronless and intron-containing reporters in SETDB1 KO cells. We find reduced Periphilin binding to intron-containing RNA (**Extended Data Fig. 7J**), with no change in binding to control L1Hs RNA. Intronless and intron-containing reporters produce similar RNA levels (in SETDB1 KO cells), allowing a reliable comparison of Periphilin binding to these two targets (**Extended Data Fig. 7I**).

We also provide levels of each reporter transcripts for the previous experiment (ex Fig. 3E, now Fig. 2C) (**Extended Data Fig. 4L, right graph**). The HUSH-insensitive reporter from Fig. 2C produces slightly more RNA. This means that RNA-binding to HUSH-insensitive reporter might be overestimated, which does not affect our conclusion that Periphilin specifically binds to HUSH-sensitive reporter.

In addition, 2) RNA-seq analysis of RIP-ChIP of HA-tagged PPHLN1 in SETDB1 KO 293T cells is also recommended. This analysis clarifies not only how much intronless gene transcripts are enriched but also any RNA binding specificities (binding motif or GC contents) or nature of target/bound RNA (length of target transcript or expression amount) of the HUSH complex (PPHLN1), thus proving a more general role or nature of intronless genetic elements for being silenced by the HUSH complex.

We agree that this is an important experiment and now present additional data showing (i) specific binding of Periphilin to RNA and (ii) analysis of the Periphilin-RNA interactome.

To better understand the specificity of Periphilin-RNA binding, we performed UV-crosslinked Periphilin RIPseq in 293T cells (both WT and SETDB1 KO) and analysed the Periphilin-RNA interactome in the context of repeated elements and different gene classes.

These analyses reveal:

1. Periphilin binds to L1 transcripts with high specificity. Only L1s and (to a lesser extent TcMar-Tigger), showed significant enrichment for Periphilin signal (**Fig. 2D, Extended Data Fig. 5C-D**)
2. Periphilin binding to L1 transcripts reflects selective, genome-wide HUSH-mediated H3K9me3 deposition over L1s as seen by preferential Periphilin binding to full length, evolutionary young L1 elements (**Extended Data Fig. 5E**)
3. Intronless genes and processed pseudogenes are enriched for Periphilin binding (**Extended Data Fig. 10D**)
4. Periphilin binding to intronless genes and pseudogenes is specific as no enrichment of signal was seen for (i) intron-containing parent genes of pseudogenes as well as other (ii) intron-containing genes (**Extended Data Fig. 10C-D**)

These data demonstrate that Periphilin-RNA binding is specific and recognition of the target transcript by Periphilin specifies target loci for HUSH-mediated H3K9me3 deposition. We feel this new data now justifies our use of the term 'recognize' with respect to HUSH and its targets (as mentioned in the previous comment).

3. HUSH targets intronless genes?

Associated with comment 2, the reviewer recommends performing bioinformatics analysis of HUSH components ChIP-seq data to clarify how intronless genes or elements are enriched at genome wide level which further validates the authors' proposal. For this analysis, the authors should focus on the genes or elements which are not targeted by TRIM28 because the HUSH complex also could be recruited to the TRIM28 target loci by the KAP1/SETDB1/ATF7IP complex and SETDB1-mediated H3K9me3.

As suggested by the reviewer, we have performed bioinformatics analysis and now present additional data in the revised manuscript:

- (i) An analysis of genes enriched in HUSH-mediated H3K9me3 (non-overlapping with L1s) based on publicly available data from (Liu et al., 2018)

We find that retrogenes and pseudogenes constitute 25% of HUSH-repressed genes (**Extended Data Fig. 9E**).

In addition, we took advantage of our newly-generated Periphilin RIPseq dataset and performed:

- (ii) Genome-wide analysis of Periphilin binding to transcripts from different gene classes: processed pseudogenes, intronless genes and intron-containing (including only those genes which don't show any overlap with Periphilin peaks over L1 elements)

We reasoned that since target recognition by the HUSH-complex relies on Periphilin-RNA binding, Periphilin-bound RNAs represent *bona fide* HUSH-targets. This analysis therefore cannot be confounded by the crosstalk between HUSH and other silencing pathways, including KAP1/SETDB1/ATF7IP, a potential problem rightly pointed out by the reviewer.

To summarise, we find:

- (i) Periphilin-binding is enriched over transcripts from pseudogenes and intronless genes but not intron-containing genes (metagene profiles in **Extended Data Fig. 10D**).
- (ii) 20% of processed pseudogenes and 17% intronless genes show at least 2-fold enrichment of Periphilin signal (**Fig. 4B, representative tracks in Extended Data Fig. 10B**). In contrast, only 5% intron-containing genes show Periphilin signal and these are predominantly: (i) ZNF genes, known HUSH-targets, and interestingly, (ii) genes with HUSH-repressed long exons (>2kb) (**Fig. 4B, Extended Data Fig. 9F&10E**).

We feel this data validates our model.

4. splicing defective intron

It is clear that all mutant introns used in this study are defective for splicing. Although the authors argue that intron-mediated protection from HUSH is independent of assembly of the

core spliceosome at the transgene RNA based on the ref 23, the biochemical evidence for absence of spliceosome components on the mutant intron elements still needs to be shown. If not spliced, what is different between unspliced intron and exon? They are the same DNA sequences. In this manuscript, at least the authors show potential mechanistic (positive) evidence of intron-mediated protection from HUSH. Discussion of possible mechanism is not satisfying.

The referee is correct that we did not show that the spliceosome is absent from the reporter mRNA with the mutant introns. As we are sure they recognise, it would be extremely challenging to unambiguously demonstrate the 'absence' of an RNA-protein interaction, particularly for such a highly abundant and complex cellular machinery as the spliceosome.

To further support our conclusion that the spliceosome machinery is not required for intron-mediated protection, we have taken a complementary, functional genetic approach. We tested the HUSH-repression of two independent reporters with non-intron (stuffer) sequences flanked by 5'ss, branch sequence and 3'ss. These reporters **remain HUSH-repressed**, despite the sequences being fully spliced from the reporter mRNA (i.e. the reporters must have effectively recruited the spliceosome machinery) (**Extended Data Fig. 8C**). These complementary and orthogonal data demonstrate that the protective effect of the intron must be independent of spliceosome recruitment to the target transcript.

With regard to the potential differences between introns and exons, it is well recognised that intron and exon sequences are not the same. The best recognized differences are the overall GC content of exons (51% GC) vs introns (46%) (Zhu et al., 2009). We have investigated the nucleotide composition of introns and we have some encouraging results. However, further work is required to solidify and mechanistically understand our observations, which is beyond the scope of this paper.

Redacted text and figure

5. HUSH target retrogene/pseudogene

a) The authors stated that "HUSH-repressed retrogenes and pseudogenes are positioned within transcriptionally active genes, in a manner similar to HUSH-regulated L1 elements." line 220-221. How can these elements be targeted and silenced by HUSH within the regular gene context and under the intron containing transcripts? The authors should address this problem.

Thank you for raising this issue. It is indeed remarkable that even within the regular gene context, HUSH specifically targets intronless elements, with the H3K9me3-deposition limited to these elements and without targeting the entire gene (as illustrated in representative tracks throughout the manuscript: e.g. Fig.4A, Extended Data Fig. 5F, Extended Data Fig. 9A-D & 10A-B). We initially thought that because many intronless HUSH-targets have their own promoter, they represent independent transcription units of the gene transcription unit they are inserted into, explaining their targeting within the regular gene context. However, the intriguing observation that HUSH also represses long exons indicates that independent transcriptional start sites might not be absolutely required for HUSH-targeting and H3K9me3-deposition. We

can only speculate here, but this observation suggests HUSH may be recruited to target loci for other reasons e.g. as a consequence of slow transcriptional elongation through long, intronless coding regions. Such a mechanism would explain HUSH-targeting of intronless elements within the regular gene context. Further detailed mechanistic studies are required to fully understand the events leading to HUSH-recruitment, which are beyond the scope of the present manuscript.

b) The authors stated that "HUSH binding and HUSH-mediated H3K9me3 are also observed over some transcribed retrogenes (e.g TAF7, MAB21L1, KBTBD7, MAP10, UTP14C) and in most cases, lead to downregulation of their expression (Fig. 5B, Extended data Fig. 8A-B)". Indeed, 2 (KBTBD7 and UTP14C) out of 5 are not silenced by HUSH, thus so far not most cases. Should revise the sentence to fit their entire findings.

This sentence has been removed from the text.

Minor points,

6. "HUSH-mediated restriction of L1 retrotransposition was reported to depend on the native nucleotide sequence of the L1 open reading frames (ORF)." line 85-86. The authors should cite original paper.

7. typo. line 201: Fig.4E → Fig. 4F

We thank the reviewer for pointing out these oversights. The citation has been added and new figure annotations verified.

Referee #3 (Remarks to the Author):

In the manuscript entitled "Genome surveillance...." Seczynska and Lehner describe how silencing foreign DNA sequences by HUSH, a fundamental unknown step within the Epigenetics field, can be rescued with the inclusion of introns.

This study builds from seminal studies by the Lehner lab, and the findings included in this study are ground-breaking are worth to be published in Nature.

We thank the reviewer for their very positive assessment and recognizing the impact of our findings. We provide a point-by-point response to their helpful comments below.

While conclusions are sound and experiments are well controlled, there are a number of points that deserve further experimentation as described below:

Major points

1) To inspect whether HUSH might recognise L1 sequences during silencing, a new lentiviral reporter was developed (Fig1, S1A). This lentiviral-based reporter allows to follow changes in L1 RNA expression by FACS. While this is a clever strategy, there are numerous controls missing regarding the validation of such reporter.

-Why authors assume L1-ORF2p is not translated from the inserted L1 reporter? Human L1-ORF2p is translated by an unconventional termination/reinitiation mechanism, AND ALL THE SEQUENCES REQUIRED FOR ITS TRANSLATION ARE LOCATED WITHIN ORF2 SEQUENCES (see Alisch et al., *Genes Dev* 2006). ORF2p translation is highly unusual, and don't even require an AUG to generate enough ORF2p to support retrotransposition (of engineered L1 vectors). Thus, authors need to demonstrate that ORF2p is not translated from the inserted L1 reporter lentivirus. This is not trivial, as L1 is known to retrotranspose to high levels in HeLa cells, and it is likely that new insertions might be generated from the inserted L1 reporter, making impossible to distinguish whether iRFP is translated from the inserted L1 reporter or from genomic de novo L1 integrations inserted by retrotransposition. The same applies for the Transposase-based construct.

We thank the reviewer for their helpful comments and for raising this important issue.

We have two explanations to suggest that our reporter monitors HUSH-mediated repression of initial L1 insertion:

(i) Using a standard L1-GFP reporter, we have shown (please see **Peer Review Figure 1** below) that depletion of HUSH leads to the accumulation of ORF1p, even in the presence of the reverse transcriptase (RT) inhibitor 3TC, which clearly inhibits L1 retrotransposition (50.9% vs 1.9%). This data provided our rationale to simplify the L1-GFP reporter to monitor L1 expression from the initial L1 insertion.

Peer Review Figure 1: Accumulation of ORF1p in TASOR-depleted cells in the absence of L1 reporter retrotransposition. (A) Fraction of GFP⁺ cells in WT and TASOR KO cells after 3 days of dox or dox plus RTi. **(B)** Western blot showing the accumulation of ORF1p in TASOR KO cells, which is independent of new L1 copies.

(ii) In all the reporters used in this study ORF2 contains the D205A mutation, which renders its endonuclease domain inactive (Feng et al. 1996) and thus cannot create new L1 insertions. We apologize that we failed to make this clear in the initial version of the manuscript. We now modify schemes of reporters to clearly indicate the D205A mutation.

Authors could use epitope-tagging of ORF1 and ORF2 sequences, as described (Doucet et al., 2010; Taylor et al., 2013).

As the reviewer has suggested, we have used ORF2 HA-tagging to demonstrate that ORF2p is not translated from the inserted L1_{pb} reporter (**Peer review Figure 2**). This is in contrast to the ORF2_{pb} reporter which supports ORF2p translation, a critical positive control.

Peer Review Figure 2: ORF2 is not translated from L1_{pb} reporter. Schematics of L1_{pb} and ORF2_{pb} reporters containing ORF2 sequence followed by HA tag (left). Expression of ORF2p-HA and ORF1p from L1_{pb} and ORF2_{pb} reporters by western blot. ORF2p is translated from a positive control ORF2_{pb} reporter but not from L1_{pb}. B-actin is a loading control.

A straightforward way to test this could be the use of the L1_{pb} transgene in the presence of RT inhibitors (AZT, 3TC, 4dT, etc).

As suggested by the reviewer, we further confirm that our reporters monitor expression from initial L1 integrations by demonstrating that the RT inhibitor 3TC does not affect iRFP levels from either L1_{lenti} and L1_{pb} (**Extended Data Fig. 1C-D**).

Similarly, can authors exclude that splicing of the inserted L1 reporter could artifactually generate iRFP?

As helpfully suggested by the reviewer, we have examined L1 expression by Northern blot and demonstrate that iRFP is not artifactually generated due the splicing of inserted L1 reporter (**Extended Data Fig. 1H**).

There are commercial antibodies to ORF1p and ORF2p, and these should be used to control that iRFP and ORF1p are translated at an equimolar rate, and to further explore ORF2p expression.

We note that different antibodies have different binding capacities and thus the relative amounts of ORF1p and iRFP cannot be directly compared with different antibodies. To directly and accurately compare relative amounts of ORF1p and iRFP produced from L1_{pb} reporter, we translated the reporter *in vitro* in the presence of ³⁵S-labelled methionine. After 45min, ORF1p and iRFP both constitute 46% of total signal, demonstrating that ORF1p and iRFP are translated at an equimolar rate.

Peer Review Figure 3: Time course of L1_{pb} *in vitro* translation. Relative contribution of reaction products are quantified for 45min time point. Signal was normalized to the methionine content.

-iRFP expression should be also explored by Northern-blot, not by RTqPCR, which will also would control for putative changes in splicing of the L1 reporter.

We have examined L1 reporter expression by Northern blot as requested (**Extended Data Fig. 1H**). It shows (i) increased reporter transcripts levels in TASOR KO and (ii) full length transcripts (6kb) are produced in both WT and TASOR KO (see long exposure).

- While populations of transduced cells were used in Fig1, authors should also explore clonal HUSH KO lines, to really demonstrate that indeed the inserted L1 reporter is expressed from all (presumably) lentiviral insertions.

We note that iRFP histograms for WT and TASOR KO cells are almost completely non-overlapping (**Fig 1A** and **Extended Data Fig. 1G**), demonstrating that the inserted L1 reporter must be expressed from all lentiviral insertions.

The above controls are critical, as KO HUSH cells are known to support elevated levels of L1 retrotransposition, and there is not a single piece of data to demonstrate that iRFP might not be produced from bona fide L1 integrations in HeLa.

In the revised manuscript, we clarify that our L1 reporter is retrotransposition-incompetent due to the ORF2p mutation (see above) and show that iRFP expression arises only from the initial integrations (**Extended Data Fig. 1C-D**).

2) While presented data seem to suggest that ORF2p sequences might be responsible for HUSH-mediated silencing, critical controls are missing.

-On one hand, lack of silencing detected in TASOR depleted (or KO) cells is very different depending if only ORF2 or ORF1 and ORF2 sequences are included in the L1pb construct. Cells transduced with the only-ORF2 construct (Fig 1D, left side) are not as homogenous as when ORF1 and ORF2 are present within the construct (Fig 1B). However, it is unclear why authors used different constructs and cell lines in these experiments, and clearly, there is not

an easy way to compare among these experiments (the equivalent of comparing apple and oranges). Authors should use the same construct backbone and cell type to conduct these experiments. This is important, as at difference with constructs containing ORF1 and ORF2 sequences, ORF1-only and ORF2-only constructs cannot retrotranspose in cultured cells, which could explain why (even if apple and oranges were compared) the profile of reporter expression is different. Note that trans-complementation of L1s is not effective.

We agree with the reviewer that it is difficult to quantitatively compare HUSH-mediated repression of full length L1 and ORF2-only reporters (**Fig. 1B vs Extended Data Fig. 1I, left plot**). However, we did not intend to compare repression of full length L1 and ORF2-only reporters but rather to qualitatively assess **HUSH-mediated repression** of reporters with and without ORF2. Our conclusion that ORF2 is responsible for repression of L1 transgene by HUSH is therefore fully supported by the data because ORF2-only (**Extended Data Fig. 1I**) but not ORF1-only (**Fig. 1D and Extended Data Fig. 1I**) is clearly HUSH-repressed.

Reporters of different architecture have intrinsically different expression profiles (e.g. reporters with ORF2 sequence within 3'UTR versus reporter with translated ORF2). To accurately compare HUSH-mediated repression of ORF1 and ORF2-only reporters, we decided to place ORF sequences downstream of GFP so that both transcription and translation initiate in the GFP. We note that a similar strategy was used in Han et al. 2004. This is relevant as we observed that the reporter with ORF2 placed upstream of GFP is intrinsically poorly expressed (**Peer Review Fig. 4**). Importantly, ORF2 also elicits HUSH-mediated repression in this configuration, which further supports our conclusions.

Without wishing to confuse the reader, we would argue that the use of different constructs and cells lines is important as it ensures that our findings are robust and not limited to specific experimental setups.

Peer Review Fig. 4: HUSH-mediated repression of reporters with ORF2 and ORF1 sequences placed upstream of GFP.

-More importantly, in 2004 Han et al demonstrated that poor L1 expression is due to inadequate transcriptional elongation by RNAPolIII, and that the A/T richness of the L1 sequences is responsible for the poor elongation. In a back-to-back study, Han and Boeke

further demonstrated that poor L1 expression could be alleviated by increasing the GC content of the L1 sequence (i.e., codon optimization), in a length-dependent manner. Authors should explore whether the AT richness of sequences inserted by viral infection and their length (Fig 2), rather than their sequence, are the main feature recognised by HUSH. Indeed, many viruses are known to be A-rich, in part to avoid restriction by ZAP.

Thank you for making these helpful points and we are indeed familiar with these studies. We apologize if this was not made clear, but the ORF2 reporter in the previous Fig. 2C (**now Extended Data Fig. 3B&E**) contains ORF2 sequence with increased GC content (from 39% to 59%), which we previously referred to as ORF2 'synthetic'. This nomenclature is not our own, but derived in the Deininger lab who generated this L1 (Gasior et al., 2006). In the current submission we have tried to avoid further confusion and called it 'codon optimized' and clearly indicate its increased GC content.

High GC% did not abolish HUSH-repression - although notably, and consistent with the observation of Han and Boeke, it was better expressed than native ORF2, but in a HUSH-independent manner (please compare expression of both reporters in TASOR KO).

This finding is consistent with a lack of correlation between the overall AT content of insert sequences and reporter HUSH-repression (**Extended Data Fig.3A**). Instead, when we analysed the individual base content of different cDNAs from our reporters, we found a strong correlation between HUSH-repression and A content in the sense strand (**Extended Data Fig. 3C**).

ORF2 itself has a strong A-rich bias in the sense strand. To experimentally validate the observed correlation between HUSH-repression and A content in the sense strand, we compared HUSH-repression of reporters with ORF2 and reverse-complement ORF2 (T-rich bias in the sense strand). Remarkably, we found reverse-complement ORF2 to be completely HUSH-insensitive (**Extended Data Fig. 3D-F**), despite producing full length transcript as confirmed by Northern blot (**Extended Data Fig. 3G**). Such a neat experimental setup using reporters with the same overall AT content and length but different sense-strand nucleotide content confirms the A-rich bias in the sense strand as an important determinant of HUSH-repression. Interestingly, and as helpfully pointed by the reviewer, retroviruses are known to have this A-rich bias (Kypr and Mrázek, 1987; Berkout and Hemert, 1994), which we briefly discuss in the text (lane 259-261).

We note that the A-rich bias does not by itself confer HUSH-sensitivity as illustrated by the HUSH-insensitive ORF1 (1kb) reporter, which has similar A-rich bias as ORF2 (~40%). However, the reporter with ORF1 repeats (3x or 4x ORF1) becomes increasingly HUSH-sensitive (**Fig. 1D, Extended Data Fig. 3H**). HUSH susceptibility is therefore governed by both A-rich bias in the sense strand and target length.

We thank the reviewer for emphasising the work from the Boeke lab as it is helpful to consider our findings in the context of their study (Han et al., 2004). We would argue that the inadequate elongation over L1 and poor L1 expression they report, can now be explained by HUSH-mediated H3K9me3 deposition and is consistent with the role of H3K9me3 in decreasing Pol II elongation. This was first proposed by the Wysocka lab (Liu et al., 2018), and our observations further support these findings.

(Han et al., 2004) reported that:

- (i) ORF2 sequence is responsible for poor L1 expression. Our data shows that ORF2 and not ORF1 elicits HUSH-repression.
- (ii) This inhibitory effect of ORF2 on expression does not map to discrete sequence and rather is length dependent. Our observations are that non-overlapping L1 fragments or deletion of these fragments are all HUSH-repressed
- (iii) ORF2 only causes inadequate elongation when expressed in the 'sense orientation', and poor expression of reporters with ORF2 in the antisense orientation is associated with premature transcription termination. We find that HUSH-repression is seen with sense ORF2, while reverse complement ORF2 is completely HUSH-insensitive (please note our reporters have deletions of two potential polyadenylation sites)
- (iv) ORF1 itself is efficiently expressed but insertion of 4 tandem repeats of ORF1 into reporter leads to poor reporter expression, similar to that observed for ORF2-containing reporter, an effect not seen with tandem repeats of 5'UTR which lacks A-rich bias. We observe ORF1 itself is HUSH-resistant but the reporter with ORF1 tandem repeats becomes increasingly HUSH-repressed and we show a strong correlation between A-content and HUSH-repression

In conclusion, our data explains many of the previously unexplained data reported from the Boeke lab, and strongly supports the notion that HUSH-repression is responsible for the inhibitory effect of L1 on gene expression.

-Related to above: to rule-out that other L1 sequences might be implicated in HUSH-silencing, authors should also explore the role of 5' and 3' UTR sequences, as both are fast evolving during evolution to presumably escape host control.

In regard to L1 UTRs, we thought that it would be interesting to compare 5' and 3' UTRs from evolutionary old L1 with the most recent L1s. However, our attempts to amplify these sequences from genomic DNA or obtain synthesized DNA from commercial sources (due to high-complexity sequence) have so far been unsuccessful and we have not therefore pursued this route of investigation.

- Authors tried to identify regions in L1-ORF2p responsible for HUSH-silencing (Fig. 2D). While data suggest that there are no regions in ORF2 responsible for their silencing, no controls for splicing were included. These controls are critical, as all constructs could generate the same (or very similar) mRNA due to splicing with L1 sequences. Consistently, Belancio and colleagues identified a major splicing site in ORF2.

To control for putative splicing within reporters with different ORF2 fragments, we analysed reporter transcripts by RT-PCRs with primers flanking ORF2 fragments. For all reporters, we observed PCR products of the predicted sizes (**Extended Data Fig. 2K**).

3) It is unclear why some inserted DNAs are silenced and others are not, but their length seem to be a major determinant in their silencing (and perhaps their nucleotide composition). This

should be further explored. For example, would HUSH silence a construct containing 4 tandem copies of L1-ORF1 (the size of ORF2)?

As discussed above, we have further explored the contribution of length and nucleotide composition of transgenes to their HUSH-sensitivity. We now provide additional data related to this aspect (**Extended Data Fig. 2G, Extended Data Fig. 3A-G**).

As suggested, we tested HUSH-repression of a reporter in which the 4kb-ORF2 sequence was replaced by tandem repeats of 1kb-ORF1 sequence. The reporter with 4xORF1 is repressed to a similar extent as ORF2. This result is now shown as **Fig. 1D (left)** along with a corresponding Northern blot in **Extended Data Fig. 1J**, confirming full-length transcript is produced from the reporter. This effect was length-dependent as HUSH-repression gradually increased with the number of ORF1 repeats (**Extended Data Fig. 3H**).

4) While presented data seem to indicate that transgene integration is not strictly required for HUSH silencing, it seem to have a clear effect in reporter expression (Fig 3A). The same applies for transcription. However, if transcription is required for transgene repression, it is unclear why the promoterless L1 reporter used in Fig3C is enriched in H3K9me3. The same applies for TAF7 (Fig3D).

The non-integrated lentivirus is intrinsically poorly expressed (Sakai et al., 1993) and this effect is independent of HUSH – illustrated by HUSH-insensitive control reporter (empty) which is poorly expressed in the absence of integration (previous Fig3A - now Fig.2A). Reduced expression may affect the extent of silencing, for instance due to the decrease in transcripts available for Periphilin binding. However, the reporter clearly remains HUSH-repressed, which supports the conclusion that integration is not strictly required for HUSH-repression. These data have been confirmed using an orthogonal approach with both viral and non-viral transfected plasmids.

The referee is also asking why we detect any H3K9me3 signal over the promoterless L1 reporter (now Fig. 2B) or the TAF7 locus with the promoter deletion (now Extended Data Fig. 4I). This residual H3K9me3 signal is comparable to the H3K9me3 signal obtained in the TASOR KO and represents the background of the ChIP-qPCR assay. We have added a genome browser track with the ChIPseq signal mapped to the unique part of the transgene (ORF1-ORF2 mapping is not possible due to multiple endogenous L1 elements). We find no H3K9me3 over promoter-less reporter (**Fig. 2B**).

As above, authors should explore RNA levels by Northern-blot, to exclude that splicing of L1 sequences might interfere with reporter expression.

We have performed Northern blots and confirmed that reduced expression from the reporter lacking the promoter is not due to the splicing of L1 sequences (**Extended Data Fig. 4F**).

5) Role of splicing. Even if L1 (and other cellular cDNAs) don't contain canonical introns, there are numerous reports documenting splicing of L1 sequences (from the Deininger and Moran labs). Indeed, it appears that splicing is an effective way to attenuate retrotransposition.

Furthermore, the full-length L1 mRNA is the less prevalent RNA isoform detected in cells. As authors conclude, these data suggest that canonical introns, rather than splicing *per se* might be involved in avoiding HUSH-mediated repression. As stated above, Northern-blot should be conducted to confirm that the expected RNAs are generated upon transfecting constructs used in Fig 4A&C.

We performed Northern blots and confirmed that the expected RNAs are generated from reporters used in previous Fig. 4A&C; now Fig. 3A&C (**Extended Data Fig. 6B and Extended Data Fig. 7D**).

In addition, authors could explore the use of self-splicing introns in constructs to further strength their data. Similarly, authors could also explore whether AT-AC introns can also prevent HUSH-silencing.

As suggested, we used a self-splicing intron (a model group I intron from *Tetrahymena*). The self-splicing intron did not abolish HUSH-mediated repression (**Peer Review Figure 5**), an observation consistent with the conclusion that splicing *per se* does not protect against HUSH-mediated repression. However, low efficiency of self-splicing makes this experiment difficult to interpret and we have therefore not included this data in the manuscript.

Peer Review Fig. 5: HUSH-mediated repression of reporters with self-splicing intron and self-splicing intron with mutated splice sites.

6) Related to above: while data seem to strongly suggest that the presence of introns, spliced or not, might be ultimately related with avoiding HUSH-silencing, their length also seem to be important. Following the same rationale as in point 2, could authors explore whether intron nucleotide composition is associated with the avoidance of HUSH-silencing?

We have investigated the nucleotide composition of introns. While we have some encouraging results, further work is required to solidify and mechanistically understand our observations, which is beyond the scope of this paper.

Redacted text and figure

7) Pseudogene datasets. While intriguing, only a handful of processed pseudogenes have been explored here. Authors should explore all processed pseudogenes annotated in the human genome, to gain more robust conclusions, and to further establish the main role of

transcription in HUSH-mediated silencing. This is critical, as the examples shown here could be the outliers, as there thousands of processed pseudogenes in the human genome.

We performed systematic analysis and present additional data in the revised manuscript:

- (i) Analysis of genes regulated by HUSH-mediated H3K9me3 (non-overlapping L1s) (**Extended Data Fig. 9E**) showing that pseudogenes constitute 18% of HUSH-targeted genes with 83% being transcriptionally active by criteria: (i) either showing detectable RNAseq signal or being located within transcribed genes.
- (ii) Genome-wide analysis of Periphilin-RNA binding over transcribed pseudogenes (**Fig. 4B; Extended Data Fig. 10D**)

In summary, we analysed 2649 processed pseudogenes (excluding pseudogenes with Periphilin peaks over L1s) and found that 567 (20%) show at least 2-fold enrichment of Periphilin (**representative tracks in Extended Data Fig. 10A**). Periphilin binding to transcripts from these elements is specific because transcripts from intron-containing parent genes are not Periphilin bound (**Extended Data Fig. 10C**). Similarly, there is an overall enrichment of Periphilin signal over processed pseudogenes, unlike over intron-containing protein genes (**Extended Data Fig. 10D**).

We feel this data strengthens our claim that processed pseudogenes are specifically recognized by the HUSH-complex.

8) A conclusion of the current study seems to suggest that introns, and not splicing, can avoid HUSH-mediated repression of L1 sequences. While intriguing, an open question is how HUSH can distinguish between intronless and intron-containing sequences. Or how HUSH can selectively bind young L1s and not older L1s. Several studies from the Ule lab have demonstrated that several RNA-binding-proteins (RBP) bind young L1s and insulate LINE sequences from RNA processing (i.e., splicing). As RBPs such as MATR3 and PTB1P bind young L1s, could these proteins be involved in the HUSH-mediating silencing described here?

We thank the reviewer for this interesting point. We observed no effect of MATR3 depletion on retrotransposition of the standard L1-GFP reporter. In addition, neither MATR3 nor PTB1P gene were hits in our unpublished genome-wide CRISPR/Cas9 screen for negative regulators of L1_{pb} reporter expression (unlike HUSH components and other transcriptional regulators). This suggests that MATR3/PTB1P are not involved in HUSH-mediated silencing, or there is a redundancy between RBPs which precludes detection of the loss-of-function phenotype of a single RBP.

9) Could authors also explore the behaviour of intron-less genes (Sox and others)? According to their model, some could be targeted by HUSH. Indeed, finding intron-less genes regulated by HUSH might allow to further solidify the proposed model, by inserting functional and mutated introns in these genes using CRISPR/Cas9.

We agree with the reviewer that our model predicts some intronless genes (retrogenes) should be targeted by HUSH and some representative examples were presented in our previous Fig. 5 and accompanying extended data.

We have now performed a more systematic analyses and present additional data in the revised manuscript:

- (i) Analysis of genes regulated by HUSH-mediated H3K9me3 (non-overlapping with L1s), among which we indeed find intronless protein-coding genes **(Extended Data Fig. 9E)**
- (ii) Genome-wide analysis of Periphilin-RNA (RIP-seq) binding over transcribed genes (non-overlapping with L1s) **(Fig. 4B; Extended Data Fig. 10D)**

To summarise our findings:

We find Periphilin binding is enriched over transcripts from intronless genes **(Extended Data Fig. 10D, middle plot)** and 17% of intronless genes show at least 2-fold enrichment of Periphilin **(Fig. 4B, right, representative tracks in Extended Data Fig. 10B)**. This is in striking contrast to intron-containing genes that show no enrichment of Periphilin binding **(Extended Data Fig. 10D, right plot)**, with only a few specific gene classes Periphilin bound **(Fig. 4B, left)**: (i) ZNF genes, known HUSH-targets, and interestingly, (ii) genes with unusually long exons (>2kb).

We agree the experiment suggested by the reviewer (to insert functional and mutated introns in HUSH regulated genes) would be very elegant and we strongly considered this option. However, it is technically challenging. Knock-in of the ~1kb-long intron sequence into all alleles is expected at only very low efficiency and its validation only possible by genomic PCR, which with preferential amplification of short amplicons (no insertion) is insufficiently sensitive for high-throughput clone screening. We have therefore not been successful here due to technical challenges.

Minor points:

a) There are some retroelements that have introns within their genomes, such as Penelope retrotransposons (Arkipoova, Systematic Biology, Volume 55, Issue 6, December 2006, Pages 875–885). While how introns are preserved during retrotransposition remains to be uncovered, strong statements should be corrected in the manuscript.

We have corrected these strong statements.

b) In results, add a reference to the following statement: "HUSH-mediated restriction of L1 retrotransposition was reported to depend on the native nucleotide sequence of the L1 open reading frames (ORF)"

We thank the reviewer for pointing out that oversight. The reference has been added.

c) page 7, line 194: authors refer to Fig4E, but it should refer to 4F.

New figure references are included.

Mrázek, J., and Kypr, J. (1994). Biased distribution of adenine and thymine in gene nucleotide sequences. *J. Mol. Evol.* 39, 439–447.

B, B., and FJ, van H. (1994). The unusual nucleotide content of the HIV RNA genome results in a biased amino acid composition of HIV proteins. *Nucleic Acids Res.* 22, 1705–1711.

Gasior, S.L., Palmisano, M., and Deininger, P.L. (2006). Alu-linked hairpins efficiently mediate RNA interference with less toxicity than do H1-expressed short hairpin RNAs. *Anal. Biochem.*

Han, J.S., Szak, S.T., and Boeke, J.D. (2004). Transcriptional disruption by the L1 retrotransposon and implications for mammalian transcriptomes. *Nature* 429, 268–274.

Kypr, J., and Mrázek, J. (1987). Unusual codon usage of HIV. *Nature* 327, 20.

Li, H., Chen, D., and Zhang, J. (2012). Analysis of Intron Sequence Features Associated with Transcriptional Regulation in Human Genes. *PLoS One* 7, 46784.

Liu, N., Lee, C.H., Swigut, T., Grow, E., Gu, B., Bassik, M.C., and Wysocka, J. (2018). Selective silencing of euchromatic L1s revealed by genome-wide screens for L1 regulators. *Nature* 553, 228–232.

Sakai, H., Kawamura, M., Sakuragi, J., Sakuragi, S., Shibata, R., Ishimoto, A., Ono, N., Ueda, S., and Adachi, A. (1993). Integration is essential for efficient gene expression of human immunodeficiency virus type 1. *J. Virol.* 67, 1169–1174.

Zhu, L., Zhang, Y., Zhang, W., Yang, S., Chen, J.-Q., and Tian, D. (2009). Patterns of exon-intron architecture variation of genes in eukaryotic genomes. *BMC Genomics* 2009 101 10, 1–12.

Reviewer Reports on the First Revision:

Referee #1:

The authors have answered all of my questions and included significant new data that support their hypothesis. I recommend publication.

Todd Macfarlan

Referee #2:

The authors responded to the reviewer's comments properly. The new data of HUSH RIP-seq analysis and further dissection of HUSH-target specificity are very nice. No further comments. The revised version should be published in *Nature*.

Referee #3:

In the revised manuscript entitled "Genome surveillance...." Seczynska,...., and Lehner includes a substantial amount of new data to solidify their findings and model, which in essence uncovered how HUSH repress foreign DNA sequences, in a A-rich and length dependent manner. I congratulate the authors for their effort in solidifying their study, especially in these atypical COVID times that have clearly impacted our capability to conduct wet-bench research. In sum, and while some aspects of HUSH-repression warrant additional research, I think that the findings reported here merit publication in Nature.

Below I include comments to the responses provided by authors. All minor points were adequately addressed, and for the rest:

- In response to Major point 1:

-- The fact that authors used a mutant L1 clearly address this major concern. However, the authors should be aware that the EN mutant used, D205A, is not completely dead for retrotransposition in HeLa or other cultured cells, retrotransposing at 2-5% the level of an allelic wild-type L1. In fact, L1 can retrotranspose by an endonuclease-independent mechanism, targeting DNA lesions/breaks. Therefore, the statements "...and thus cannot create new L1 insertions" (rebuttal) and "...prevents retrotransposition and the reporter therefore monitors expression from initial L1 integrations..." (revised manuscript) are not accurate and should be edited. Indeed, if the authors were concerned about retrotransposition from the site of viral integration/s, a much better control in these experiments would be using a RT mutant L1, which is truly compromised for retrotransposition. Or even better, an L1 carrying a mutation in the RNA binding domain of L1-ORF1p (RR261/62AA), which cannot form a RNP and thus cannot retrotranspose in cells (most of these mutations were original described in Moran et al., 1996; see also Kulpa and Moran 2005).

Finally, I don't think that the statement in the rebuttal "We apologize that we failed to make this clear in the initial version of the manuscript" is fair, as there was NOT A SINGLE mention to the use of this mutant L1 in the submitted manuscript, not in Methods, Figures or Supplemental Table with PLASMID LIST.

-- On a related point, the data included in "Peer Review Figure 2: ORF2 is not translated from L1pb reporter" is misleading and contradicts many studies in the field of L1 biology. You can't and shouldn't compare ORF2p translation from a bicistronic context (L1pb) with a monocistronic context (ORF2pb). This is comparing apple and oranges. There are numerous reports demonstrating that ORF2p is translated from bicistronic constructs, analogous to those used in this study. Thus, I don't understand what the authors are trying to demonstrate here. If the intention of authors is to exclude ORF2p translation from L1lenti or L1pb, confocal microscopy would be a fairer comparison (see Doucet et al., PLoS Genet., 2010). Also, I don't know which cell type was used in experiments shown in Peer Review Fig. 2; unless HEK293T and SV40-containing constructs were used, trying to detect ORF2p on a western from a bicistronic RNA is simply naïve (Taylor et al., Cell, 2013).

-- In full agreement with authors' conclusions, the use of RTis, Ext. Data Fig. 1c,d, indicates that iRFP is generated from the site of lentiviral integration, and not from subsequent retrotransposition integration sites. The authors should indicate the concentration of 3TC used. Together with the use of D205A-mutated L1s, I believe this concern has been satisfactorily addressed.

-- Northern blot experiments: As with the RTi experiments, the new Northern blot data adds to the validation of the strategy/system used by authors to follow silencing of L1 sequences, which is an important control added to the manuscript. Do the authors know what the 9Kb hybridizing band corresponds to? Also, it could be convenient to add that an iRFP probed was used (in the figure or

legends).

-- L1-ORF expression by western blot: The authors used in vitro translation (S35 experiments) to explore the synthesis of iRFP and ORF1p as independent proteins. While the fusion protein could be detected in these experiments, it is also clear that indeed both proteins are translated in an equimolar ratio. Again, these data add to the validation of the strategy/system used by authors to follow silencing of L1 sequences, and I congratulate the authors. In fact, their strategy of using in vitro translation is less biased than what I proposed using antibodies.

-- With respect to my original suggestion: "While populations of transduced cells were used in Fig. 1, authors should also explore clonal HUSH KO lines, to really demonstrate that indeed the inserted L1 reporter is expressed from all (presumably) lentiviral insertions" As the authors responded, "histograms for WT and TASOR KO cells are ALMOST completely non-overlapping...", which indicates that not all cells would express the inserted transgene at the same level, which was the point I was trying to reach. In other words, while the system created is very robust and allow to study silencing, it is very (very) unlikely that all integration sites would express the same amount of L1 RNA, as sites of integration clearly impact their expression. It is uncommon for biological systems to work at 100% efficiency. My intention here was for the authors to solidify that their findings were independent of the site of integration. However, and while I still think that exploring clonal lines would be informative, with the addition of new data I found that the conclusions of the study are solid.

- In response to Major point 2:

-- ORF1 and ORF2 only vs ORF1-ORF2 constructs: I am indeed satisfied with the response provided by authors. Comparing among cell lines and constructs is rather difficult. That said, the addition of new data (Peer review Fig. 4) solidifies ORF2-dependency of silencing, and further support conclusions of authors. My suggestion would be to also include Peer review Fig. 4 to further support ORF2-dependency in a completely different experimental setting.

-- A-T richness: I congratulate the authors. The new experiment with the A vs T enrichment is very very informative, and I think it adds to the mechanism of HUSH silencing. I agree with the interpretation of authors and agree the A vs T findings are solid and unexpected but quite informative, helping also to consolidate previous findings by Boeke and Wysocka labs.

For what it is worth, the authors may want to consider discussing a somehow unexplained finding in L1 biology: genic endogenous full-length L1s are enriched in the antisense transcriptional orientation of genes, at a 1.8 antisense/sense ratio, which is not trivial. These findings were originally reported by Arian Smit in 1999 (Smit, A.F. (1999). Interspersed repeats and other mementos of transposable elements in mammalian genomes. *Curr. Opin. Genet. Dev.* 9:657–663), but were confirmed in the human genome sequence project and others. The current working hypothesis suggests that evolutionary forces and the T-richness of coding strands (the L1 EN recognises a consensus 5' TTTTT/AA for integration) might favour enrichment on antisense strands. However, I think that the findings reported here add to this model and further suggest why L1s are "excluded" from the sense strand, as HUSH-silencing could interfere with gene expression.

-- Role of 5' UTR and 3' UTR L1 sequences. It is somehow surprising that old LINEs could not be amplified from genomic DNA, or that commercial sources failed to obtain them by DNA synthesis. In 2013, the Engel lab reported the successful synthesis of the consensus ORFs from two evolutionary human L1s with success (L1PA4 and L1PA8 elements; see Molecular reconstruction of extinct LINE-1 elements and their interaction with nonautonomous elements. Wagstaff BJ, Krutter EN, Derbes RS, Belancio VP, Roy-Engel AM. *Mol. Biol. Evol.* 2013 30:88-99). While with the addition of new data these experiments might not be that relevant (I found the A-bias findings quite strong), the authors may want to consider contacting Astrid Engel for advice/support if this is

a route of research further consider.

-- Splicing controls: As above, the new data supporting A-bias silencing make this control less relevant. I also acknowledge the use of RT-qPCR to control for splicing differences, but unfortunately, only Northern blotting using unbiased probes can truly rule out splicing differences.

- In response to Major point 3: To address the influence of A-T richness and length of inserted foreign DNA on HUSH silencing, the authors now demonstrate that a construct containing four tandem copies of ORF1 (i.e., with a similar size to ORF2) is silenced in a HUSH-dependent manner, further solidifying their model. The northern-blot controls are very nice and further rule out splicing. As above, I am curious about the 9-kb band... do the authors know the nature of this sequence?

- In response to Major point 4: I am completely satisfied with the response provided and agree that the new data indicate that integration is not strictly required for HUSH-repression. Additionally, the addition of browser tracks and northern blot data helps to clarify lack of expression from promoterless constructs, as expected.

- In response to Major point 5:

-- Northern blot data: Again, I congratulate the authors as these important controls exclude splicing.

-- Use of self-splicing intron: I acknowledge the effort of authors in testing a self-splicing intron to further demonstrate that splicing doesn't prevent HUSH-repression of L1 ORF2. I also agree that the efficiency of splicing in this context is not high enough to interpret these data without biases and agree that these data should not be incorporated in the manuscript. I am completely satisfied with the response provided.

- In response to Major point 6 (length and composition of introns): Again, I acknowledge the effort of the authors in addressing this point with the addition of new data.

[Redaction due to referee
comments about authors'
unpublished data]

□

- In response to Major point 7 (exploring additional pseudogenes): The analysis of new pseudogenes (>2500) and findings reported clearly solidify this part of the study. I congratulate the authors for their efforts and findings.

- In response to Major point 8 (role of RNA-binding proteins (RBP) on HUSH-repression): First, and for clarification, I never suggested that MATR3 (or PTB1P) would influence L1 retrotransposition on an engineered assay that is known to NOT recapitulate L1 transcription from genomic loci (L1-GFP reporter assay). Thus, I don't understand the point made by authors. That said, and although I don't have access to these data, the fact that neither gene was found in the CRISPR-Cas9 screen for negative regulators of L1pb expression might suggest that these genes are likely not influencing HUSH-repression.

However, and as originally stated in my review, a fundamental question that remains to be addressed is how HUSH can distinguish between intronless and intron-containing sequences, and how young L1s are preferentially targeted by HUSH. I understand that these are not trivial questions.

- In response to Major point 9 (intron-less genes): The addition of new genome-wide analyses and RIP-seq data solidify that HUSH preferentially regulate intronless genes, which was the point I was making. Thus, and while not functionally tested, these data, together with the remaining data included in the study, is enough to support the model proposed by authors.

Regarding the newly added data, below I add several minor points that should be explored by authors:

A) New RIP experiments: These experiments provide a nice link between transcription and HUSH repression and are highly relevant to support conclusions of the study. However, there are some aspects that deserve further analyses.

A.1. On one hand, Ext. Data Fig. 5 clearly shows an enrichment on L1, and in an age-dependent manner: the younger the L1, the more RNAs bound by periphilin. However, L1Hs elements, the younger and only active class in the human genome, are not the elements bound by periphilin at the highest level. This is surprising considering that what it is reported here is a new restriction mechanism to active TEs. In fact, L1PA2 and L1PA3 elements, currently inactive, are the ones showing the highest level of periphilin binding. However, data in cultured cells have further demonstrated that histone modifications, and not DNA methylation, are the main factor influencing expression of young L1s. As a result, only a handful of full-length L1 copies are expressed in a given cell line (see Philipe et al., eLife, 2016 for a recent study). By exploring periphilin binding to expressed L1s using subfamilies, the authors might be masking preferential binding to expressed young L1s, which are the likely target of HUSH. Thus, the authors should use recent pipelines that allow the identification of specific expressed L1s copies in the periphilin dataset. L1ME, from the Fenyo lab, is a recent tool that allows one to explore expression at a loci-specific manner. By reanalysing the periphilin dataset using L1ME, the authors would have a better resolution of which L1 copies are bound by periphilin, which could influence mechanistic aspect of HUSH repression.

A.2. TcMar-Tigger DNA Transposon: Are these sequences more A-T-rich than other TEs? What is the average length of TcMar-Tigger DNA transposons explored? Clearly, a closer look at these transposons can provide important clues to definitively address how HUSH might recognise TEs that would be repressed.

Author Rebuttals to First Revision:

Referee #1:

The authors have answered all of my questions and included significant new data that support their hypothesis. I recommend publication.

Todd Macfarlan

Referee #2:

The authors responded to the reviewer's comments properly. The new data of HUSH RIP-seq analysis and further dissection of HUSH-target specificity are very nice. No further comments. The revised version should be published in Nature.

We thank reviewers #1 and #2 for their recommendation.

Referee #3:

In the revised manuscript entitled “Genome surveillance....” Seczynska,....., and Lehner includes a substantial amount of new data to solidify their findings and model, which in essence uncovered how HUSH repress foreign DNA sequences, in a A-rich and length dependent manner. I congratulate the authors for their effort in solidifying their study, especially in these atypical COVID times that have clearly impacted our capability to conduct wet-bench research. In sum, and while some aspects of HUSH-repression warrant additional research, I think that the findings reported here merit publication in Nature.

We thank the reviewer for the recommendation and appreciate their positive feedback.

Below I include comments to the responses provided by authors. All minor points were adequately addressed, and for the rest:

- In response to Major point 1:

-- The fact that authors used a mutant L1 clearly address this major concern. However, the authors should be aware that the EN mutant used, D205A, is not completely dead for retrotransposition in HeLa or other cultured cells, retrotransposing at 2-5% the level of an allelic wild-type L1. In fact, L1 can retrotranspose by an endonuclease-independent mechanism, targeting DNA lesions/breaks. Therefore, the statements “...and thus cannot create new L1 insertions” (rebuttal) and “...prevents retrotransposition and the reporter therefore monitors expression from initial L1 integrations...” (revised manuscript) are not accurate and should be edited.

We have edited the statement in the manuscript.

Indeed, if the authors were concerned about retrotransposition from the site of viral integration/s, a much better control in these experiments would be using a RT mutant L1, which is truly compromised for retrotransposition. Or even better, an L1 carrying a mutation in the RNA binding domain of L1-ORF1p (RR261/62AA), which cannot form a RNP and thus cannot retrotranspose in cells (most of these mutations were original described in Moran et al., 1996; see also Kulpa and Moran 2005).

We acknowledge the reviewer’s concern that EN D205A mutant displays residual 2-5% retrotransposition ability and the suggestion that perhaps RT ORF2 or ORF1 mutants could have been used instead. However, we did not expect that, even with functional ORF2, retrotransposition from the site of transgene integration/s would significantly contribute to transgene expression because of the time point used in our assays. Indeed, we measured expression at time points that do not support high levels of retrotransposition i.e. 24 post dox induction (for piggyBac reporters) or 48 h post transduction (for lentiviral reporters). Our timecourse experiment with the retrotransposition competent L1-GFP piggyBac reporter suggests that it takes \$\geq 3\$ days to complete retrotransposition and to detect expression from new sites of integration. We initially were actually more concerned about the potential toxicity of ORF2 expression. We therefore reasoned that by introducing the D205A mutation (rather than RT or ORF1p mutations), we could not only further reduce the possibility of retrotransposition but also reduce potential toxicity of ORF2 EN (Gasior et al., 2006; Wallace et al., 2008).

Finally, I don't think that the statement in the rebuttal “We apologize that we failed to make this clear in the initial version of the manuscript” is fair, as there was NOT A SINGLE mention to

the use of this mutant L1 in the submitted manuscript, not in Methods, Figures or Supplemental Table with PLASMID LIST.

Again, we can only apologise for the confusion. We mistakenly omitted this information in the initial submission.

-- On a related point, the data included in "Peer Review Figure 2: ORF2 is not translated from L1pb reporter" is misleading and contradicts many studies in the field of L1 biology. You can't and shouldn't compare ORF2p translation from a bicistronic context (L1pb) with a monocistronic context (ORF2pb). This is comparing apple and oranges. There are numerous reports demonstrating that ORF2p is translated from bicistronic constructs, analogous to those used in this study. Thus, I don't understand what the authors are trying to demonstrate here. If the intention of authors is to exclude ORF2p translation from L1lenti or L1pb, confocal microscopy would be a fairer comparison (see Doucet et al., PLoS Genet., 2010). Also, I don't know which cell type was used in experiments shown in Peer Review Fig. 2; unless HEK293T and SV40-containing constructs were used, trying to detect ORF2p on a western from a bicistronic RNA is simply naïve (Taylor et al., Cell, 2013).

We were previously asked by the reviewer to use epitope-tagging as referenced in (Doucet et al., 2010; Taylor et al., 2018) to demonstrate that ORF2p is not translated from L1pb. Both studies utilized biochemical approaches (western blots, affinity purification) as well as confocal microscopy imaging. so we took the biochemical approach. We simply used ORF2pb (also a bicistronic vector as it contains GFP-P2A-ORF2) as a positive control to prove that our inability to detect ORF2p produced from L1pb is not due to the technical experimental issues. In fact we were unable to detect ORF2p expression from the L1pb reporter even when after HA-affinity purification and subsequent blotting for HA, further suggesting that ORF2 is not expressed from our L1pb. In these experiments we used HeLa cells with stable integration of L1pb driven by dox-responsive CMV promoter (induced for 3 days to allow the accumulation of L1 proteins). Perhaps the presence of the ~1kb iRFP sequence between ORF1 and ORF2 inhibits the unconventional translation of the ORF2 sequence.

-- In full agreement with authors' conclusions, the use of RTis, Ext. Data Fig. 1c,d, indicates that iRFP is generated from the site of lentiviral integration, and not from subsequent retrotransposition integration sites. The authors should indicate the concentration of 3TC used. Together with the use of D205A-mutated L1s, I believe this concern has been satisfactorily addressed.

We have specified that 3TC was used at 50 μ M concentration.

-- Northern blot experiments: As with the RTi experiments, the new Northern blot data adds to the validation of the strategy/system used by authors to follow silencing of L1 sequences, which is an important control added to the manuscript. Do the authors know what the 9Kb hybridizing band corresponds to?

We think that the 9kb hybridizing band likely corresponds to the mRNA product from read-through transcription terminating at the second polyadenylation site (PAS) within the piggyBac constructs. This second PAS is to terminate transcription from the cassette containing antibiotic-resistance cassette plus tet-repressor which is driven by an ubiquitin promoter (see scheme below).

Also, it could be convenient to add that an iRFP probed was used (in the figure or legends).

We have indicated that iRFP was used to detect reporter transcript.

-- L1-ORF expression by western blot: The authors used in vitro translation (S35 experiments) to explore the synthesis of iRFP and ORF1p as independent proteins. While the fusion protein could be detected in these experiments, it is also clear that indeed both proteins are translated in an equimolar ratio. Again, these data add to the validation of the strategy/system used by authors to follow silencing of L1 sequences, and I congratulate the authors. In fact, their strategy of using in vitro translation is less biased than what I proposed using antibodies.

We are pleased the reviewer liked this experiment.

-- With respect to my original suggestion: "While populations of transduced cells were used in Fig. 1, authors should also explore clonal HUSH KO lines, to really demonstrate that indeed the inserted L1 reporter is expressed from all (presumably) lentiviral insertions". As the authors responded, "histograms for WT and TASOR KO cells are ALMOST completely non-overlapping...", which indicates that not all cells would express the inserted transgene at the same level, which was the point I was trying to reach. In other words, while the system created is very robust and allow to study silencing, it is very (very) unlikely that all integration sites would express the same amount of L1 RNA, as sites of integration clearly impact their expression. It is uncommon for biological systems to work at 100% efficiency. My intention here was for the authors to solidify that their findings were independent of the site of integration. However, and while I still think that exploring clonal lines would be informative, with the addition of new data I found that the conclusions of the study are solid.

- In response to Major point 2:

-- ORF1 and ORF2 only vs ORF1-ORF2 constructs: I am indeed satisfied with the response provided by authors. Comparing among cell lines and constructs is rather difficult. That said, the addition of new data (Peer review Fig. 4) solidifies ORF2-dependency of silencing, and further support conclusions of authors. My suggestion would be to also include Peer review Fig. 4 to further support ORF2-dependency in a completely different experimental setting.

Data from Peer review Fig. 4 has been now incorporated into Extended Data Fig. 11 (bottom panel). -- A-T richness: I congratulate the authors. The new experiment with the A vs T enrichment is very very informative, and I think it adds to the mechanism of HUSH silencing. I agree with the interpretation of authors and agree the A vs T findings are solid and unexpected but quite informative, helping also to consolidate previous findings by Boeke and Wysocka labs.

We are pleased the reviewer appreciates these findings are indeed a significant addition to our study.

For what it is worth, the authors may want to consider discussing a somehow unexplained finding in L1 biology: genic endogenous full-length L1s are enriched in the antisense transcriptional orientation of genes, at a 1.8 antisense/sense ratio, which is not trivial. These findings were originally reported by Arian Smit in 1999 (Smit, A.F. (1999). Interspersed repeats and other mementos of transposable elements in mammalian genomes. *Curr. Opin. Genet. Dev.* 9:657–663), but were confirmed in the human genome sequence project and others. The current working hypothesis suggests that evolutionary forces and the T-richness of coding strands (the L1 EN recognises a consensus 5′ TTTT/AA for integration) might favour enrichment on antisense strands. However, I think that the findings reported here add to this model and further suggest why L1s are “excluded” from the sense strand, as HUSH-silencing could interfere with gene expression.

This is a very helpful and important point and we agree with the reviewer that HUSH-silencing is likely to contribute to the asymmetric distribution of the L1 elements. We considered to mention this in our manuscript but are constrained by space limitations. It would be beneficial for future discussions.

-- Role of 5′ UTR and 3′ UTR L1 sequences. It is somehow surprising that old LINEs could not be amplified from genomic DNA, or that commercial sources failed to obtain them by DNA synthesis. In 2013, the Engel lab reported the successful synthesis of the consensus ORFs from two evolutionary human L1s with success (L1PA4 and L1PA8 elements; see Molecular reconstruction of extinct LINE-1 elements and their interaction with nonautonomous elements. Wagstaff BJ, Kroutter EN, Derbes RS, Belancio VP, Roy-Engel AM. *Mol. Biol. Evol.* 2013 30:88-99). While with the addition of new data these experiments might not be that relevant (I found the A-bias findings quite strong), the authors may want to consider contacting Astrid Engel for advice/support if this is a route of research further consider.

We thank the reviewer for pointing this out and referring us to the work of Astrid Engel. While our revised manuscript was under reviewers’ evaluation, we have in fact successfully amplified the 5′UTR and 3′UTR sequences from evolutionary young L1 and tested their effects on HUSH-repression (please see figure below). While the L1 5′UTR enhances expression of the L1pb reporter, it still remained HUSH-sensitive as seen in the absence of the 5′UTR. The L1 3′UTR does not affect HUSH-repression; either alone or in combination with the 5′UTR.

Peer Review 2 Figure 1: The effect of 5′ and 3′UTR sequences on HUSH-repression of L1pb reporter.

-- Splicing controls: As above, the new data supporting A-bias silencing make this control less relevant. I also acknowledge the use of RT-qPCR to control for splicing differences, but unfortunately, only Northern blotting using unbiased probes can truly rule out splicing differences.

- In response to Major point 3: To address the influence of A-T richness and length of inserted foreign DNA on HUSH silencing, the authors now demonstrate that a construct containing four tandem copies of ORF1 (i.e., with a similar size to ORF2) is silenced in a HUSH-dependent manner, further solidifying their model. The northern-blot controls are very nice and further rule out splicing. As above, I am curious about the 9-kb band... do the authors know the nature of this sequence?

We thank the reviewer for their comments and as stated above, the 9kb hybridizing band likely corresponds to the mRNA product from read-through transcription terminating at the second polyadenylation site (PAS) within the piggyBac constructs.

- In response to Major point 4: I am completely satisfied with the response provided and agree that the new data indicate that integration is not strictly required for HUSH-repression. Additionally, the addition of browser tracks and northern blot data helps to clarify lack of expression from promoterless constructs, as expected.

- In response to Major point 5:

-- Northern blot data: Again, I congratulate the authors as these important controls exclude splicing.

-- Use of self-splicing intron: I acknowledge the effort of authors in testing a self-splicing intron to further demonstrate that splicing doesn't prevent HUSH-repression of L1 ORF2. I also agree that the efficiency of splicing in this context is not high enough to interpret these data without biases and agree that these data should not be incorporated in the manuscript. I am completely satisfied with the response provided.

We thank the reviewer for the positive comments above.

- In response to Major point 6 (length and composition of introns): Again, I acknowledge the effort of the authors in addressing this point with the addition of new data.

- In response to Major point 7 (exploring additional pseudogenes): The analysis of new pseudogenes (>2500) and findings reported clearly solidify this part of the study. I congratulate the authors for their efforts and findings.

We are glad the reviewer finds our analysis strengthens our manuscript.

- In response to Major point 8 (role of RNA-binding proteins (RBP) on HUSH-repression): First, and for clarification, I never suggested that MATR3 (or PTB1P) would influence L1 retrotransposition on an engineered assay that is known to NOT recapitulate L1 transcription from genomic loci (L1-GFP reporter assay). Thus, I don't understand the point made by authors. That said, and although I don't have access to these data, the fact that neither gene

was found in the CRISPR-Cas9 screen for negative regulators of L1pb expression might suggest that these genes are likely not influencing HUSH-repression.

However, and as originally stated in my review, a fundamental question that remains to be addressed is how HUSH can distinguish between intronless and intron-containing sequences, and how young L1s are preferentially targeted by HUSH. I understand that these are not trivial questions.

- In response to Major point 9 (intron-less genes): The addition of new genome-wide analyses and RIP-seq data solidify that HUSH preferentially regulate intronless genes, which was the point I was making. Thus, and while not functionally tested, these data, together with the remaining data included in the study, is enough to support the model proposed by authors.

We thank the reviewer for acknowledging that our data support the model.

Regarding the newly added data, below I add several minor points that should be explored by authors:

A) New RIP experiments: These experiments provide a nice link between transcription and HUSH repression and are highly relevant to support conclusions of the study. However, there are some aspects that deserve further analyses.

A.1. On one hand, Ext. Data Fig. 5 clearly shows an enrichment on L1, and in an age-dependent manner: the younger the L1, the more RNAs bound by periphilin. However, L1Hs elements, the younger and only active class in the human genome, are not the elements bound by periphilin at the highest level. This is surprising considering that what it is reported here is a new restriction mechanism to active TEs. In fact, L1PA2 and L1PA3 elements, currently inactive, are the ones showing the highest level of periphilin binding. However, data in cultured cells have further demonstrated that histone modifications, and not DNA methylation, are the main factor influencing expression of young L1s. As a result, only a handful of full-length L1 copies are expressed in a given cell line (see Philipe et al., eLife, 2016 for a recent study). By exploring periphilin binding to expressed L1s using subfamilies, the authors might be masking preferential binding to expressed young L1s, which are the likely target of HUSH. Thus, the authors should use recent pipelines that allow the identification of specific expressed L1s copies in the periphilin dataset. L1ME, from the Fenyo lab, is a recent tool that allows one to explore expression at a loci-specific manner. By reanalysing the periphilin dataset using L1ME, the authors would have a better resolution of which L1 copies are bound by periphilin, which could influence mechanistic aspect of HUSH repression.

The reviewer is correct that a percentage of L1Hs bound by Periphilin is lower than the percentage of Periphilin-bound L1PA2-L1PA3 (please see the table 1 for exact numbers).

	in WT cells	in SETDB1 KO (mix) cells
L1Hs	7.9%	9.3%
L1PA2	10.9%	14.0%

L1PA3	10.3%	12.7%
-------	-------	-------

Table 1: Fraction of Periphilin-bound genomic L1s.

It is important to underline that L1s/TEs need to be **transcriptionally active** to be targeted by HUSH but not necessarily active in terms of their ability to ‘jump’ i.e. retrotransposition competent. Retrotransposition competence requires intact L1s/TEs open reading frames and generation of functional L1/TE proteins, but these are not features recognized by HUSH, that can only identify its targets at the chromatin level. Therefore we agree with the reviewer (on the data presented in Extended Data Fig. 5E) that there is indeed a correlation between the Periphilin-binding and the L1 age (given a lower percentage of genomic L1Hs bound by Periphilin is observed, in comparison to L1PA2-L1PA3), but this correlation is not perfect.

This is most likely to relate to the ability to uniquely map RIPseq reads to L1Hs, as L1Hs is **the least sequence-divergent** L1 family (as discussed by Liu et al. Nature 2018 in Extended data Figure 5A). In fact, L1PA2-L1PA3 can be uniquely mapped with higher confidence than L1Hs. The number of reads for L1PA2-L1PA3 is therefore expected to be higher, and will facilitate peak calling, resulting in a higher number of instances identified as HUSH-targets. Periphilin-bound L1Hs are thus likely to have been underestimated in comparison to L1PA2-L1PA3 and we now highlight this limitation in the legend of **Extended Data Fig.5E**. While there are a number of different bioinformatics approaches to try to alleviate this problem, they each have their pros and cons and we do not think this issue will necessarily be alleviated by using a different pipeline for the analysis as the ability to uniquely map reads is ultimately dependent on the length of sequencing reads.

The relationship between Periphilin L1 RNA binding and L1 age is not central to our study and was presented to further validate the specificity of Periphilin binding to target RNA by illustrating that Periphilin-L1 RNA binding reflects the pattern of HUSH-mediated H3K9me3-deposition over L1 elements. This is exactly the case. In fact, in previous work by the Wysocka lab, **L1Hs also shows slightly lower enrichment in MORC2/HUSH ChIPseq than L1PA2, exactly as we observe in our RIPseq experiments** (Fig. 3D – most right hand panel) from Liu et al. Nature 2018).

A.2. TcMar-Tigger DNA Transposon: Are these sequences more A-T-rich than other TEs? What is the average length of TcMar-Tigger DNA transposons explored? Clearly, a closer look at these transposons can provide important clues to definitively address how HUSH might recognise TEs that would be repressed.

This is an excellent point and we have indeed started investigating HUSH-repressed Tiggers more closely. Our preliminary data confirm that HUSH-repressed Tiggers are AT/A-rich (~61% overall AT content) and that Periphilin preferentially binds to full-length Tigger1 and Tigger2, which validates our model. These analyses need to be further expanded by investigating HUSH-repressed Tiggers in the context of other TEs and the entire TcMar-Tigger family and will be presented independently.

nature portfolio